# PROGRESSIVE DISTILLATION INDUCES AN IMPLICIT CURRICULUM

**Abhishek Panigrahi**[*,†]  **Bingbin Liu**[*α]  **Sadhika Malladi**[†]  **Andrej Risteski**[α]  **Surbhi Goel** [β]
[†] Princeton University       [α] Carnegie Mellon University       [β] University of Pennsylvania
{ap34,smalladi}@cs.princeton.edu, {bingbinl,aristesk}@cs.cmu.edu,
surbhig@cis.upenn.edu

## ABSTRACT

Knowledge distillation leverages a teacher model to improve the training of a student model. A persistent challenge is that a better teacher does not always yield a better student, to which a common mitigation is to use additional supervision from several "intermediate" teachers. One empirically validated variant of this principle is *progressive distillation*, where the student learns from successive intermediate checkpoints of the teacher. Using sparse parity as a sandbox, we identify an *implicit curriculum* as one mechanism through which progressive distillation accelerates the student's learning. This curriculum is available only through the intermediate checkpoints but not the final converged one, and imparts both empirical acceleration and a provable sample complexity benefit to the student. We then extend our investigation to Transformers trained on probabilistic context-free grammars (PCFGs) and real-world pre-training datasets (Wikipedia and Books). Through probing the teacher model, we identify an analogous implicit curriculum where the model progressively learns features that capture longer context. Our theoretical and empirical findings on sparse parity, complemented by empirical observations on more complex tasks, highlight the benefit of progressive distillation via implicit curriculum across setups. Code is available here.

## 1   INTRODUCTION

As the cost of training state-of-the-art models grows rapidly (Hoffmann et al., 2022), there is increased interest in using knowledge distillation (Hinton et al., 2015) to leverage existing capable models to train new models more efficiently and effectively. Knowledge distillation is an effective technique to train smaller vision (Jia et al., 2021; Touvron et al., 2021; Yu et al., 2022; Lin et al., 2023) and language models (Sanh et al., 2019; Gunasekar et al., 2023; Touvron et al., 2023; Reid et al., 2024) that permit faster inference with comparable performance. However, one curiously persistent phenomenon is that a better teacher does not always yield a stronger student. Prior works (Mirzadeh et al., 2019; Jin et al., 2019; Jafari et al., 2021; Harutyunyan et al., 2022; Anil et al., 2018) hypothesized that this is due to a capability gap between the teacher and the student. As such, they proposed *progressive distillation*, where the student is incrementally supervised by increasingly capable teachers. This technique has yielded strong empirical performance. One recent example is the training of Gemini-1.5 Flash from Gemini-1.5 Pro (Reid et al., 2024; Team et al., 2024): Gemini-1.5 Flash achieves 95% of Gemini-1.5 Pro's performance on average and outperforms Gemini-1.0 Pro on 41 out of 50 text-based long-context benchmarks, while being substantially smaller. However, little is understood about progressive distillation in terms of the optimization or generalization benefits, compared to directly learning from the data or the final teacher checkpoint (i.e., *one-shot distillation*).

Most prior work hypothesizes that progressive distillation enables better generalization (Mirzadeh et al., 2019; Jafari et al., 2021; Harutyunyan et al., 2022). In contrast, we identify a novel mechanism by which progressive distillation helps a student by *accelerating its optimization* (Figure 1). We define optimization acceleration as achieving improved performance with fewer training steps or samples. In this paper, we use fresh training samples in each training step; hence we use training steps and samples interchangeably to measure the optimization speed.

---
[*]Equal Contribution

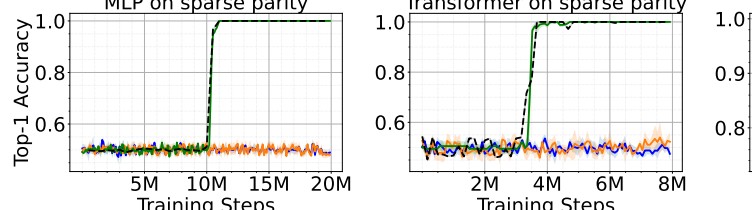

Figure 1: **Progressive distillation accelerates training.** *Left*: MLP on $(100, 6)$-sparse parity (Definition 3.1), with width-50k teachers and width-100 students. Progressive distillation checkpoints are at 100k-step intervals, and one-shot checkpoint uses the final (20M-step) checkpoint. *Middle*: Transformer on $(100, 6)$-sparse parity, with 32-head teachers and 4-head students. Progressive distillation checkpoints are at 10k-step intervals, and the one-shot checkpoint is at 250k steps. *Right*: Transformers on PCFG (Section 4), with 32-head teachers and 8-head students using BERT-style masked prediction. Progressive distillation uses 8 intermediate checkpoints.

We study two tasks where learning the right features is believed to be important and show that the intermediate checkpoints provide signal towards these features. The first is learning sparse parity (Definition 3.1), which is a commonly studied setting to understand the feature learning dynamics of neural networks. The second is learning probabilistic context-free grammars (PCFGs), which we use as a sandbox for capturing certain aspects of language modeling. Theory and extensive experiments in these settings support the following claims.

1. **Progressive distillation accelerates student learning.** Our experiments in multiple settings demonstrate that progressive distillation accelerates training compared to standard one-shot distillation and learning from the data directly (Figure 1). More specifically, for sparse parity, progressive distillation can train a smaller MLP (or Transformer) at the same speed as a larger MLP (or Transformer). For PCFGs, progressive distillation improves the accuracy of a smaller BERT model (Devlin et al., 2018) at masked prediction. Finally, we verify our findings on more realistic setups of training BERT on Wikipedia and Books dataset.

2. **An implicit curriculum drives faster learning.** We demonstrate theoretically and empirically that acceleration comes from an *implicit curriculum* of easy-to-learn subtasks provided by intermediate teacher checkpoints, which is not available from the final teacher checkpoint. For sparse parity, the easy-to-learn subtasks provide supervision for the coordinates which constitute the support of the sparse parity (Section 3). As a consequence, we show progressive distillation provably improves the sample complexity for sparse parity over one-shot distillation or learning directly from data (Theorem 3.2). For PCFGs, the implicit curriculum is defined in terms of learning features that increasingly capture larger $n$-gram contexts. Our results also provide guidance on how to select the intermediate teachers used during progressive distillation.

**Related works**[1]. One persistent surprise in knowledge distillation is that stronger teachers do not always lead to stronger students. Prior works have speculated that an overly large "teacher-student gap" is the cause, and accordingly proposed to bridge this gap by introducing supervision of intermediate difficulty (Mirzadeh et al., 2019; Cho & Hariharan, 2019; Harutyunyan et al., 2022; Jafari et al., 2021). Mirzadeh et al. (2019) used multi-step distillation involving models of intermediate sizes, and Shi et al. (2021) proposed to directly inject teacher supervision into the student's trajectory using an approximation of mirror descent. Most related to our work, Harutyunyan et al. (2022) analyzed distillation for extremely wide networks and found it helpful to learn from the intermediate checkpoints of the teacher, a strategy also adopted by Jin et al. (2019). They speculated that this is because neural networks learn progressively complex functions during training (Kalimeris et al., 2019). In contrast to their focus on the generalization ability of the student, we study the optimization dynamics of distillation.

It is worth noting that there is also a rich body of work on understanding standard (one-shot) distillation, mostly regarding regularization effects. In particular, Menon et al. (2021) shows that learning from the teacher leads to a tighter generalization bound when the teacher is closer to the Bayes distribution over the class labels. However, such Bayes perspective cannot explain the training

---

[1]We defer a detailed discussion of related work to Appendix A.1.

acceleration in the feature learning tasks considered in this work, whose the Bayes distributions are delta masses and hence are the same as the one-hot labels themselves. Our results fill this gap by providing an orthogonal view of implicit curriculum.

The benefit of curriculum on parity has also been explored in Abbe et al. (2024a;b), where the curriculum gradually increases the input Hamming weight. The difference though is that such curricula are defined by *explicitly* altering the distribution over the inputs, whereas our curriculum shows up *implicitly* in the teacher supervision. Moreover, our implicit curriculum emphasizes that a properly chosen intermediate checkpoint, while having a worse accuracy than the final checkpoint, can lead to a better-performing student. This can be seen as a plausible mechanism for *weak-to-strong generalization* (Burns et al., 2023).

**Outline.** Section 2 describes the distillation strategies. Section 3 introduces the implicit curriculum with a case study on sparse parity, presenting both empirical evidence and a provable benefit in sample complexity. Section 4 continues the empirical investigations on PCFG, and extends the observations to BERT's training on Wikipedia and Books dataset. Finally, Section 5 discusses open directions.

## 2 PRELIMINARIES

We now outline the distillation strategies considered in this paper and their empirical instantiation. For ease of exposition, we discuss one-dimensional label classification tasks here and generalize to sequence-to-sequence functions in Section 4. Denote the teacher and student models operating on input domain $\mathcal{X}$ as $f_{\mathcal{T}} : \mathcal{X} \to \mathbb{R}^C$ and $f_{\mathcal{S}} : \mathcal{X} \to \mathbb{R}^C$, respectively. The outputs of a model $f$ are logits that are transformed into a probability distribution over $C$ classes using a softmax function with temperature $\tau$, denoted as $p(x; \tau) := \text{softmax}(f(x)/\tau)$. We will use $p_{\mathcal{T}}$, $p_{\mathcal{S}}$ to denote the probability distributions of the teacher and the student, and will omit the subscript to denote a generic model. When $\tau = 1$, we omit $\tau$ from the notation for brevity. Following Zheng & Yang (2024), we set $\tau = 1$ for the student and vary the temperature of the teacher.

We compare two loss functions: $\ell$, where the student $f_{\mathcal{S}}$ learns only from ground-truth labels , and $\ell_{\text{DL}}$ , where the student $f_{\mathcal{S}}$ is supervised only with the logits of some teacher $f_{\mathcal{T}}$.[2]

$$\ell(x, y; f_{\mathcal{S}}) = \text{KL}(\boldsymbol{e}_y \| p_{\mathcal{S}}(x)), \tag{1}$$

$$\ell_{\text{DL}}(x; f_{\mathcal{S}}, f_{\mathcal{T}}) = \text{KL}(p_{\mathcal{T}}(x; \tau) \| p_{\mathcal{S}}(x)), \tag{2}$$

where $\boldsymbol{e}_y$ is a one-hot vector whose $y^{th}$ entry is 1. We consider two strategies for choosing the teacher. The first is *one-shot distillation*, where the student learns from a fixed $f_{\mathcal{T}}$ throughout the training, and the teacher is chosen as the final converged checkpoint. The second is *progressive distillation*, where the student learns from multiple intermediate checkpoints of the teacher's training run:

**Definition 2.1** (($C_{\mathcal{T}}, \mathcal{D}$)-progressive distillation)**.** Given a set of teacher checkpoints $C_{\mathcal{T}} = \{f_{\mathcal{T}_i}\}$ and a set of training durations $\mathcal{D}$, the student is trained with the logits of teacher checkpoint $f_{\mathcal{T}_i}$ for training length $\mathcal{D}_i$ with $i \in [|C_{\mathcal{T}}|] := \{1, \cdots, |C_{\mathcal{T}}|\}$.

To simplify the presentation, the main paper tests a specific type of progressive distillation schemes, where $C_{\mathcal{T}}$ contains $N$ equally-spaced checkpoints and the student is trained on each one for $T$ steps:

**Definition 2.2** (($N, T$)-progressive distillation)**.** $C_{\mathcal{T}}$ contains $N - 1$ equally-spaced intermediate teacher checkpoints and the final teacher checkpoint. The student is trained with each checkpoint for $T$ training steps. After $NT$ steps, the student is trained with the final teacher checkpoint.

To study the effect of each teacher checkpoint, we will also consider an extreme version of progressive distillation with $N = 2$, where the student uses one intermediate teacher checkpoint.

**Choice of temperature.** We set $\tau = 10^{-4}$ for sparse parity and PCFG experiments (Section 4) where the vocabulary size is smaller than 5, and $\tau = 10^{-20}$ for natural language experiments (Section 4.2) whose vocabulary size is 30k.[3] Using such a small temperature makes the teacher's outputs close to one-hot labels. This removes potential regularization effects due to the softness of the

---

[2]We note that prior papers generally use a combination of these objectives, but we use supervision from one source in order to isolate its effects on distillation.

[3]Figure 14 provides a comparison in temperature choices for sparse parity learning.

labels (Yuan et al., 2020) which would otherwise be a confounding factor. Moreover, the supervision with nearly one-hot labels is more representative of the setting where the student learns directly from the teacher's *generations* instead of the logits. This method, often described as generating synthetic data in the language modeling setting, has generally yielded small yet highly performant students (Gunasekar et al., 2023; Liu et al., 2024). For one-shot distillation, we report the best-performing temperature among $\tau = 1, 10^{-4}$ in the main paper and defer other results to Appendix D.8.

## 3    THE IMPLICIT CURRICULUM: A CASE STUDY WITH SPARSE PARITY

To elucidate the mechanism by which distillation accelerates training, we first focus on the well-studied task of learning sparse parity.[4] Sparse parity is a commonly used sandbox for understanding neural network optimization in the presence of feature learning (Barak et al., 2022; Bhattamishra et al., 2022; Morwani et al., 2023; Edelman et al., 2023; Abbe et al., 2024b).

**Definition 3.1** ($(d, k)$-sparse parity task). Let $\mathbf{S} \subset [d]$ denote a fixed set of coordinates, with $|\mathbf{S}| = k$ and $k < d$. Then, the sparse parity task is defined for any input $\mathbf{x} \in \{\pm 1\}^d$, whose label is computed as $y = 1$ if $\prod_{i \in \mathbf{S}} \mathbf{x}_i > 0$ and 2 otherwise.

We train the teacher and student models using 2-label classification, where $f_\mathcal{T}$ and $f_\mathcal{S}$ return logits in $\mathbb{R}^2$. The teacher and the student have the same number of layers but different sizes. We vary the model width for MLP, and vary the number of attention heads for Transformer, with a fixed per-head dimension. These choices not only affect the parameter counts, but also govern the learning speed[5].

**Why can larger models learn faster?** A natural way to learn sparse parity with gradient descent involves first identifying the support $\mathbf{S}$ and subsequently computing the product of variables in the support (i.e., $\prod_{i \in \mathbf{S}} \mathbf{x}_i$). Empirically, the two stages of learning manifest as a long plateau period in the model's accuracy, followed by a sharp phase transition (Figure 1, left and middle). The search for the support is what makes learning problem difficult, as it depends on the input dimension $d$ rather than the support size (Abbe et al., 2023; Barak et al., 2022). The benefit of increasing the width or the number of heads comes from providing more "parallel search queries." For MLP, prior work has shown that increasing the width accelerates training (Edelman et al., 2023), which we also observe in Figure 7 (left) in appendix. For Transformers though, we find that increasing the number of attention heads is the most effective for improving the convergence speed, as opposed to increasing the per-head dimension or the MLP width. A detailed comparison is provided in Appendix C.2 (Figure 10). Given this finding, we will vary the number of attention heads between the teacher and the student, while keeping the per-head dimension fixed. The number of heads hence directly controls the parameter count. This choice also aligns with the practice in open-sourced models such as the Llama series (Touvron et al., 2023).

In the following, we first empirically verify that carefully chosen intermediate teacher checkpoints constitute an *implicit curriculum* for the student to learn from. Then, we show that this curriculum provably improves the speed of learning in the student by improving its training sample efficiency.

### 3.1    ACCELERATING LEARNING WITH THE IMPLICIT DEGREE CURRICULUM

The difficulty of the search problem suggests that we can accelerate student learning by providing direct supervision for what the support is (Abbe et al., 2023). We show that supplying the intermediate signal from a bigger teacher model accelerates the search process for the smaller model, as described by the following set of results.[6]

**(R1) Intermediate teacher checkpoints constitute an implicit degree curriculum.** We provide empirical evidence that the supervision from intermediate teacher checkpoints serves as an implicit curriculum supplying strong signals for certain degree-1 monomials, which require fewer samples to learn. In Figure 2, we report the correlation between degree-1 monomials and the prediction of the teacher logits at various checkpoints. The correlation for each monomial $x_j, j \in [d]$ is computed as $|\mathbb{E}_{\boldsymbol{x},y}([p_\mathcal{T}(\mathbf{x})]_1 \cdot x_j)|$ at each checkpoint $f_\mathcal{T}$. Here $[p_\mathcal{T}(\mathbf{x})]_1$ refers to the first output dimension of

---

[4]We also experiment with a hierarchical generalization of sparse parity, which is deferred to Appendix C.3.

[5]In terms of the number of samples or the number of training steps, which coincide in our experiments as we use freshly sampled batches.

[6]We will mark our results with **(Ri)** throughout the paper for easy reference.

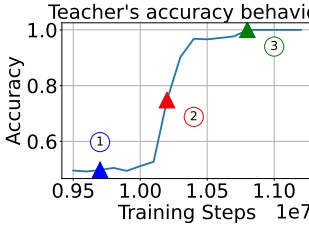 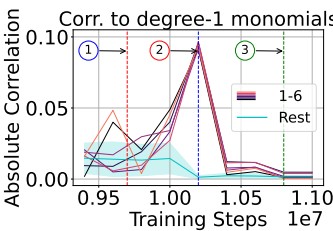 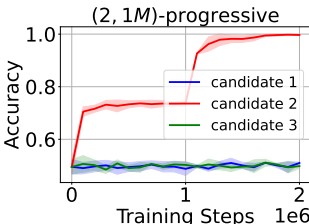

Figure 2: **Implicit curriculum for** $(100, 6)$**-sparse parity.** We compare **3 candidate intermediate checkpoints**, labeled as ①, ②, ③, corresponding to 9.7M, 10.2M, and 10.8M steps, or the beginning, middle, and end of the teacher's phase transition. *Left*: Teacher's accuracy throughout training. *Middle*: During the phase transition, $f_{\mathcal{T}}$ is much more strongly correlated with in-support variables $(x_1, \cdots, x_6$ in this case) than with off-support variables. *Right*: Only candidate ② (i.e., during phase transition) enables $(2, 1M)$-progressive distillation to reach $100\%$ accuracy. We use width-50k teachers and width-100 students; Figure 8 shows similar results for width-1000 students.

$f_{\mathcal{T}}$, which corresponds to $p(y = 1) = p(\prod_{i \in \mathbf{S}} \mathbf{x}_i > 0) = 1 - p(y = 2)$ (recall Definition 3.1). We take the absolute value as we are only concerned with the magnitude of the correlation. Importantly, these strong correlations emerge when the teacher learns the sparse parity task (i.e., during the phase transition) but diminish with continued training.

Note that the monomials need not be strictly degree-1. While our theory (Section 3.2) will only focus on degree-1 monomials for the sake of mathematical analysis, low-degree polynomials can still provide acceleration, which we also observe in practice (see Figure 9 in the Appendix for such an example). This transient low-degree supervision, available only through intermediate teacher checkpoints, may explain the superior performance of progressive distillation over one-shot distillation (Figure 1). We will confirm the provable sample complexity benefit of this implicit low degree curriculum in Section 3.2. The importance of the implicit curriculum is further strengthened by the superior performance of $(2, T)$-progressive distillation:

**(R2) Progressive distillation with a single intermediate checkpoint can outperform one-shot distillation.** We consider the extreme version of progressive distillation where only a single intermediate checkpoint is used (in addition to the final checkpoint). Figure 2 shows the result for $(2, 1M)$-progressive distillation. We consider 3 candidates for the intermediate teacher checkpoint, occurring respectively at the beginning, middle or the end of the teacher's phase transition. Our result demonstrates that the checkpoint selection is crucial, where only the checkpoint during the phase transition is useful in accelerating training.[7] This provides further evidence that the implicit degree curriculum is the key to faster training via progressive distillation.

More complex tasks may require more intermediate checkpoints, which we discuss in more depth in Appendix C.3. Nevertheless, we find that progressive distillation can be run efficiently and effectively across tasks, and a small number of intermediate teacher checkpoints often suffice to accelerate training provided that the checkpoints are properly selected.

## 3.2 THE LOW-DEGREE CURRICULUM REDUCES SAMPLE COMPLEXITY

We now formalize the benefits of progressive distillation for $(d, k)$-sparse parity in terms of sample complexity. For the sake of mathematical analysis, we take the student $f_{\mathcal{S}}$ and the teacher $f_{\mathcal{T}}$ models to be 1-hidden-layer MLPs with ReLU activations and scalar outputs. Further, the labels $y$ are given as $\pm 1$, where 1 (or $-1$) corresponds to the class dimension 1 (or 2) in Definition 3.1. Following previous works (Barak et al., 2022; Abbe et al., 2023; Edelman et al., 2023), we analyze a simplified two-stage training procedure and train the model using the hinge loss: $L_\alpha(\mathbf{x}, y; f_{\mathcal{S}}, f_{\mathcal{T}}) = \alpha \max(0, 1 - f_{\mathcal{S}}(\mathbf{x})y) + (1 - \alpha) \max(0, 1 - f_{\mathcal{S}}(\mathbf{x})f_{\mathcal{T}}(\mathbf{x}))$.

Let's first recall the hardness of learning sparse parity.[8] For simplicity, we consider the case of MLPs of width $\tilde{\mathcal{O}}(2^k)$ trained using online SGD. When learning from data alone, statistical query

---
[7]We show similar results for width-1000 students (Figure 8) and transformers (Figure 13).
[8]A more detailed discussion is provided in Appendix A.1.

(SQ, Kearns (1998)) lower bound shows that learning the support for a $(d, k)$-sparse parity requires $\Omega(d^{k-1})$ samples (Abbe et al., 2023; Edelman et al., 2023). We will show that although this lower bound also applies to one-shot distillation from a strong teacher, it can be circumvented when learning from the implicit low-degree curriculum identified in the previous section.

Specifically, we compare the sample complexity of one-shot distillation and $(2, T)$-progressive distillation (Section 2). Both strategies use a well-trained final checkpoint with an error of $\mathcal{O}(\epsilon)$ error for an arbitrarily small $\epsilon > 0$. Progressive distillation additionally uses the teacher's intermediate checkpoint after its first phase of training, where we can provably show its predictions to have correlations at least $\Omega(1/k)$ to the monomials $x_i, \forall i \in \mathbf{S}$. That is, progressive distillation first learns from the intermediate checkpoint and then switches to the final checkpoint, whereas one-shot distillation learns directly from the final checkpoint.

**(R3) Progressive distillation reduces sample complexity.** We formally demonstrate the sample complexity benefit of progressive distillation.

**Theorem 3.2** (Informal version of Theorem B.1). *Consider learning $(d, k)$-sparse parity with a student model of size $\tilde{m} = \tilde{\Theta}(2^k)$, where $\tilde{\cdot}$ hides polylog factors in $d, k$. Suppose the teacher has a loss $\mathcal{O}(\epsilon)$ for some small $\epsilon > 0$. Then, the total sample complexity needed for the student to reach $\epsilon$-loss using progressive distillation with 2 checkpoints is $\tilde{\Theta}(2^k d^2 \epsilon^{-2} + k^3)$. However, one-shot distillation requires at least $\Omega(d^{k-1}, \epsilon^{-2})$ samples.*

*Proof sketch.* We track the training behavior of the teacher model during its two-phase training. We show that at the end of the first phase, the teacher's predictions will have $\Omega(1/k)$ correlations to degree-1 monomials $x_i, \forall i \in \mathbf{S}$. In contrast, the correlations are smaller for degree-1 monomials $x_i, \forall i \notin \mathbf{S}$. Hence, the teacher's predictions can be written as $\sum_{i \in \mathbf{S}} c_i x_i + \sum_{i \notin \mathbf{S}} c_i x_i$, plus additional higher degree odd polynomials which can be controlled, with $|c_i| \geq \Omega(1/k)$ for $i \in \mathbf{S}$, and $|c_i| = o(1/kd)$, if $i \notin \mathbf{S}$. When training on the predictions from this intermediate teacher checkpoints, the correlation gap between in- and off-support degree-1 monomials will be reflected in the gradients of the student's weights. Namely, there is a $\Omega(1/k)$ gap between the support and non-support coordinates in the weight gradients. This gap allows the coordinates $i \in \mathbf{S}$ in the student's weights to grow quickly with only $\mathcal{O}(k^2 \log(\tilde{m}))$ samples.

On the other hand, for a teacher that has loss $\mathcal{O}(\epsilon)$, a similar argument can show that the separation gap between the correlations of the teacher's predictions to degree-1 monomials on support and outside support can be at most $\mathcal{O}(\epsilon)$. So, harnessing this gap will require a sample size of at least $\Omega(\epsilon^{-2})$ by concentration inequalities. Learning directly from the labels will require $\Omega(d^{k-1})$ samples from the SQ lower bound as discussed above. This gives the sample complexity differences between one-shot and progressive distillation. The full proof is provided in Appendix B. $\square$

*Remark.* One gap between our theory and experiments is that our analysis applies to large-batch SGD with small gradient noise, whereas the experiments use online SGD with batch size 1. Bridging this gap, such as by adapting the analyses in Abbe et al. (2023) on Gaussian data, is an interesting future direction.

# 4    IMPLICIT CURRICULUM WITH PCFGS AND NATURAL LANGUAGE

In this section, we empirically show that an implicit curriculum emerges generally, both when learning on probabilistic context-free grammars (PCFGs) and when performing natural language modeling tasks on the Wikipedia and Books datasets. We focus on BERT models (Devlin et al., 2018)[9], and discuss experiments on GPT-2 (Radford et al., 2019) in Appendix E.

**The masked prediction task.** Our experiments will be based on BERT models trained to perform masked prediction, which requires filling in masked-out tokens in an input sequence and excels at feature learning in natural languages (Hewitt & Manning, 2019; Tenney et al., 2019; Li et al., 2022).

**Definition 4.1** (Masked prediction task with mask rate $p$). Let $v$ denote the vocabulary that contains a special token $[mask]$, and let $h$ denote an arbitrary sequence length. Given a sequence $\mathbf{x} \in v^h$,

---

[9]See Appendix D.5.1 for a primer on BERT.

Figure 3: An example of a PCFG tree T($\mathbf{x}$) that generates $\mathbf{x}$ ="The cat ran away". "The cat" is an example of level-2 span, and "cat" is as a boundary token for the spans of both the level-1 non-terminal `Noun` and the level-2 non-terminal `Noun Phrase`.

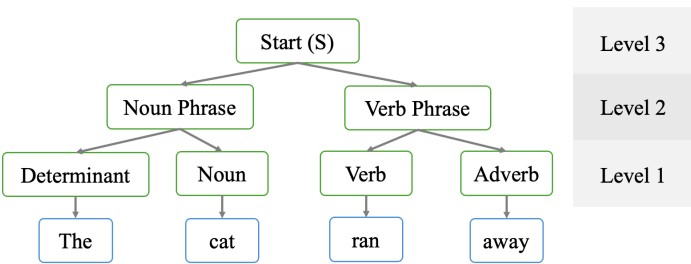

sample a set of masked positions $\mathcal{M} \in [h]$ following $P(i \in \mathcal{M}) = p, \forall i \in [h]$. Create a masked input $\mathbf{x}_{\backslash \mathcal{M}}$ from $\mathbf{x}$ by replacing tokens at positions in $\mathcal{M}$ with $[mask]$, a random token from $\mathcal{X}$, or kept unchanged with probabilities $80\%, 10\%, 10\%$ respectively. Then, the masked prediction objective is the cross-entropy of the model's predictions at positions $i \in \mathcal{M}$ on input $\mathbf{x}_{\backslash \mathcal{M}}$.

Since we are performing sequence-to-sequence modeling, we need to generalize the definition of the teacher $f_{\mathcal{T}}$ and student $f_{\mathcal{S}}$ from Section 2 accordingly, denoted as $f_{\mathcal{T}} : v^h \to \mathbb{R}^{h \times C}$ and $f_{\mathcal{S}} : v^h \to \mathbb{R}^{h \times C}$. We will use $p_{\mathcal{T}}^{(i)}(\mathbf{x}; \tau) := \text{softmax}([f_{\mathcal{T}}(\mathbf{x})]_i / \tau)$ to denote the teacher's output distribution on the $i_{th}$ position; similarly for $p_{\mathcal{S}}^{(i)}$. As before, we omit $\tau$ when $\tau = 1$. We use the following loss functions for the masked prediction task (Definition 4.1):

$$\ell(\mathbf{x}; f_{\mathcal{S}}) = \mathbb{E}_{\mathcal{M}} \frac{1}{|\mathcal{M}|} \sum_{i \in \mathcal{M}} \text{KL}(\boldsymbol{e}_{x_i} \| p_{\mathcal{S}}^{(i)}(\mathbf{x}_{\backslash \mathcal{M}})), \tag{3}$$

$$\ell_{\text{DL}}(\mathbf{x}; f_{\mathcal{S}}, f_{\mathcal{T}}) = \mathbb{E}_{\mathcal{M}} \frac{1}{|\mathcal{M}|} \sum_{i \in \mathcal{M}} \text{KL}(p_{\mathcal{T}}^{(i)}(\mathbf{x}_{\backslash \mathcal{M}}; \tau) \| p_{\mathcal{S}}^{(i)}(\mathbf{x}_{\backslash \mathcal{M}})), \tag{4}$$

where $\boldsymbol{e}_y$ is a one-hot vector whose $y_{th}$ entry is 1.

We train BERT models with $\ell, \ell_{\text{DL}}$ and report the average top-1 accuracy on the masked tokens. As discussed in Section 3.1, the teacher and student have the same depth (4 layers) but differ in the number of attention heads, with 32 heads for the teacher and 8 heads for the student. Each attention head has dimension 8, so the teacher has width 256 and the student has width 64. All hyperparameter details are in Appendix D.6.

## 4.1 $n$-GRAM CURRICULUM IN PCFGS

We first consider probabilistic context free grammars (PCFGs), which are commonly used to emulate the structure of natural language and thus provide mechanistic insights into language models (Zhao et al., 2023; Allen-Zhu & Li, 2023a). A PCFG generates sentences following a tree structure; Figure 3 shows an example for the sentence "*The cat ran away*." More precisely, a PCFG $\mathcal{G} = (\mathcal{N}, \mathcal{R}, \mathcal{P}, v)$ is defined by a set of non-terminals $\mathcal{N}$, rules $\mathcal{R}$ over the non-terminals, a probability distribution $\mathcal{P}$ over $\mathcal{R}$, and a vocabulary (terminals) $v$. A sentence $\mathbf{x}$ is associated with a generation tree T($\mathbf{x}$), whose intermediate nodes are non-terminals in $\mathcal{N}$, leaf nodes are terminals in $v$, and edges are defined by rules sampled from $\mathcal{R}$ according to $\mathcal{P}$. A formal definition of PCFG is provided in Appendix D.1. Our choices of PCFGs are taken from Allen-Zhu & Li (2023a), where all leaves in the same tree have the same distance to the root. Experiments in the main paper are based on the PCFG `cfg3b` generated by depth-7 trees, and results on other PCFGs are deferred to Appendix D.4.

### 4.1.1 PROGRESS MEASURES OF IMPLICIT CURRICULUM

Unlike our experiments on parity, what constitutes as feature is less straightforward for PCFG. We will use three progress measures to quantify the implicit curriculum for masked language modeling on PCFGs, based on $n$-gram statistics and non-terminal prediction.

Measures that use $n$-gram statistics will measure the dependence of the model's predictions on tokens in the neighboring contexts, defined as follows:

**Definition 4.2** ($n$-gram neighboring context). For a $h$-length sentence $\boldsymbol{x} \in v^h$ and for $i \in [h]$, we define the $n$-gram neighboring context around the $i^{th}$ token as the set of tokens at positions

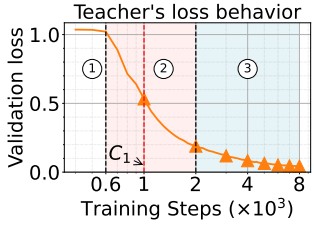 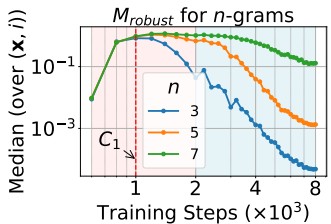 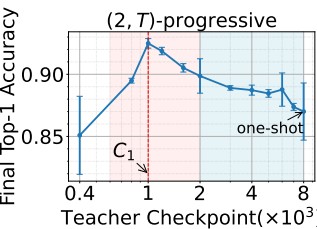

Figure 4: BERT on the PCFG `cfg3b`. *Left*: A 32-head teacher's loss exhibits three distinct phases: ① an initial phase with little change, ② a middle phase with a rapid drop, and ③ a final plateauing phase until the end of training. The triangles mark the selected checkpoints for progressive distillation, with the first teacher checkpoint (denoted by $C_1$) located at the middle of phase ②. *Middle*: $M_{\text{robust}}$ across training, which peaks at $C_1$. The model gets more robust to shorter $n$-gram perturbation as training progresses. The median is taken over the input sequences. *Right*: A 8-head student's final accuracy with $(2, T)$-progressive distillation after 4000 total training steps. The $x$-axis marks the choice of the first teacher checkpoint. $T$ is grid-searched over $\{500, 1000, 2000\}$. The best performance is obtained by choosing $C_1$. Although results in the plots are for a single training run of the teacher, similar behaviors occur robustly across random seeds.

within $(n-1)/2$ distance from $i$, denote as $n\text{-gram}(i) := \{j : \max(i - \lceil (n-1)/2 \rceil, 0) \le j \le \min(i + \lfloor (n-1)/2 \rfloor, h)\}$.

In the example of Figure 3, for the word "cat", its 3-gram neighboring context consists of words "The" and "ran", and its 5-gram neighboring context additionally includes the word "away." The choice of $n$-grams is inspired by results in Zhao et al. (2023), which show that a BERT model can solve masked prediction by implementing a dynamic programming algorithm that builds hierarchically on increasingly larger $n$-gram neighboring context spans (Definition 4.2). A model that primarily uses short $n$-gram neighboring context will be largely affected if the tokens within the context are perturbed during evaluation. This motivates us to consider two $n$-gram based measures.

**Measure 1: Robustness to removing $n$-gram context.** Our first progress measure of feature learning checks how the model's prediction changes when the $n$-gram context is present or absent. For each masked position $i$, we measure the total variation (TV) distance between the probability distributions when masking out only the current token, and when masking out all the tokens in $n\text{-gram}(i)$, i.e. the neighboring $n$-gram context centered at $i$. Recall that $\mathbf{x}_{\backslash \mathcal{M}}$ denotes a masked version of $\mathbf{x}$ with masked set $\mathcal{M}$ (Definition 4.1), and that $p^{(i)}$ denotes a model's output probability distribution at the $i_{th}$ position. Then, our first measure is defined as

$$M_{\text{robust}}(f, \mathbf{x}, i, n) = \text{TV}(p^{(i)}(\mathbf{x}_{\backslash \{i\}}), p^{(i)}(\mathbf{x}_{\backslash n\text{-gram}(i)})). \tag{5}$$

We report median of $M_{\text{robust}}(f, \mathbf{x}, i, n)$ over randomly sampled $\mathbf{x}$ and $i$ [10]. A larger $M_{\text{robust}}(f, \mathbf{x}, i, n)$ indicates the model heavily depends on neighboring $n$-gram context tokens for the masked prediction.

**Measure 2: Closeness between full and $n$-gram predictions.** Our second progress measure examines the change in predictions when the model is given the full sequence versus only a local $n$-gram window:

$$M_{\text{close}}(f, \mathbf{x}, i, n) = \text{TV}(p^{(i)}(\mathbf{x}_{\backslash \{i\}}), p^{(i)}(\mathbf{x}_{n\text{-gram}(i)\backslash \{i\}})), \tag{6}$$

where $\mathbf{x}_{n\text{-gram}(i)\backslash \{i\}}$ denotes the $n$-gram context centered at position $i$, minus the position $i$ itself. We report median of $M_{\text{close}}(f, \mathbf{x}, i, n)$ over randomly sampled $\mathbf{x}$ and $i$. A large $M_{\text{close}}(f, \mathbf{x}, i, n)$ indicates that the model utilizes contexts outside a $n$-gram window in its predictions.

**Measure 3: Non-terminal prediction.** Finally, we also measure how well the model outputs encode the features of the underlying PCFG by checking the accuracy at predicting non-terminals (Allen-Zhu & Li, 2023b). The predictions are given by a linear classifier on top of the output embeddings.

**Definition 4.3** (PCFG non-terminal prediction task). Define the *span* of a non-terminal $n$ as the set of terminals within the subtree rooted at $n$, denoted by span($n$). The (right) *boundary* of span($n$) refers

---

[10]Our observations stay the same for other percentiles.

to the rightmost position within span($n$). We say a non-terminal is of *level $i$* if it is at distance $i$ from the root. Then, *level-$i$ non-terminal prediction task* aims to predict $n^{(i)}$ at boundary of span($n^{(i)}$).

As an example, in Figure 3, the *level-2 non-terminal prediction task* aims to predict the non-terminals `Noun Phrase` and `Verb Phrase` at words "cat" and "away" respectively. More details are provided in Appendix D.3.

### 4.1.2 EMPIRICAL VERIFICATION OF THE $n$-GRAM CURRICULUM

Similar to Section 3.1, we will start with examining the training dynamics of the teacher model. We observe a phase transition period akin to that of sparse parity, during which we identify an inflection point concerning $M_{\text{robust}}$ and $M_{\text{close}}$. This inflection point proves to be a crucial intermediate checkpoint. We then demonstrate that progressive distillation improves feature learning in the student model, substantiated by the three measures defined in Section 4.1.1.

For training dynamics, we observe 3 distinct phases of training in the teacher's loss (Figure 4 left): 1) an initial phase where the loss doesn't change much for the first $5\%$ of training; 2) a rapid loss drop phase in the next $\approx 20\%$ of training; and 3) a final phase of slow loss drop till end of training. In particular, the rapid loss drop phase is reminiscent of the phase transition in sparse parity (Section 3). Moreover, we identify an inflection point (marked by $C_1$) during the second phase: before the inflection point, the robust loss $M_{\text{robust}}$ increases (Figure 4 middle), and the loss $M_{\text{close}}$ stays high (Figure 22 left); after the inflection point, both $M_{\text{robust}}$ and $M_{\text{close}}$ start to drop rapidly, suggesting that the model learns to utilize longer contexts as opposed to short neighboring $n$-grams.

**(R4) The inflection point is best for $(2, T)$-progressive distillation.** We study the importance of each teacher checkpoint by comparing the performance of $(2, T)$-progressive distillation, where the student learns from a single intermediate checkpoint in addition to the final checkpoint. The value of $T$ is grid-searched (more details in Appendix D.6). For the choice of the intermediate checkpoint, Figure 4 shows that the best intermediate checkpoint is the one at the inflection point (at 1000 training steps), which we denote as $C_1$. Note that at the inflection point, the teacher has the highest reliance on shorter $n$-grams (e.g. for $n = 3$), which are analogous to the low-degree monomials in Section 3 and serves as intermediate tasks that are likely easier to learn. Hence, $C_1$ being the optimal checkpoint choice further strengthens our hypothesis that an implicit curriculum is the key to the acceleration enabled by progressive distillation.

Following (**R4**), we will choose the checkpoints for progressive distillation at training steps that are multiples of that of $C_1$, i.e. at steps $\{i \times 10^3\}_{i=1}^8$. As shown in Figure 1 (right), progressive distillation helps the student learn faster than both one-shot distillation and cross entropy training. Furthermore, progressive distillation leads to improved feature learning.

**(R5) Progressive distillation improves feature learning on PCFG.** Progressive distillation improves over one-shot or no distillation over all 3 measures mentioned in Section 4.1.1. As shown in Figure 5, progressive distillation makes the student better utilize long contexts rather than local $n$-gram windows, evidenced by a lower $M_{\text{robust}}$ and $M_{\text{close}}$. The student can also better predict the non-terminals, suggesting a better structural learning of the underlying PCFG.

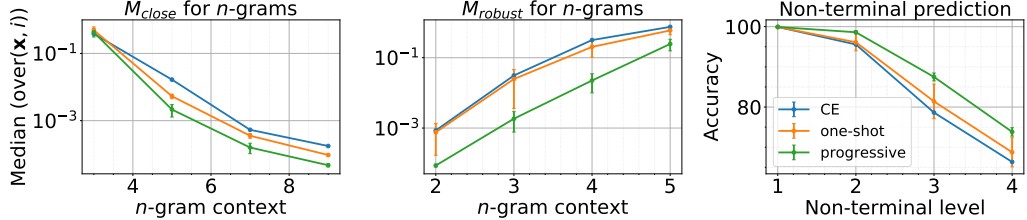

Figure 5: Comparisons on a 8-attention head BERT model. (Left) $M_{\text{close}}$ for different $n$-grams. Progressive distillation has a lower $M_{\text{close}}$ with longer $n$-gram context. (Middle) $M_{\text{robust}}$ for different $n$-grams. Progressive distillation has a lower $M_{\text{robust}}$ for all $n$-gram contexts. (Right) Probe performance to predict the non-terminals (NTs) (Definition 4.3). Progressive distilled student performs better when probed for higher level non-terminals in its contextual embeddings.

### 4.2 BEYOND SYNTHETIC SETUPS: IMPLICIT CURRICULUM IN NATURAL LANGUAGES

We conduct experiments on BERT training (Devlin et al., 2018) on Wikipedia and Books (details in Appendix F). The teacher and student both have 12 layers, with 12 and 4 attention heads per-layer respectively. Each attention head is of dimension 64, corresponding to a width-768 teacher and a width-256 student. Similar to PCFG, the teacher's loss exhibits 3 distinct phases (Figure 6 left), with an inflection point marking the change in $M_{\text{robust}}$ (Figure 6 middle). The inflection point can hence provide an implicit curriculum towards easier-to-learn local $n$-grams. Finally, progressive distillation helps the student achieve better accuracy at masked language prediction (Figure 6 right). In Table 3 (appendix), we further show that progressive distilled models achieve an average 0.9 performance improvement on downstream tasks after fine-tuning.

*Connections to related works.* Our results align with those of Chen et al. (2023), who observed a phase transition in loss when training BERT on real-world language data corresponding to the model learning syntax rules of language. Comparable findings were also reported in a concurrent work on matrix completion (Gopalani et al., 2024). For auto-regressive models, prior work has discussed the emergence of $n$-gram induction heads which indicate phases in which the model learns to perform in-context learning (Akyürek et al., 2024; Quirke et al., 2023; Olsson et al., 2022). We observe similar behavior on PCFGs and Wikipedia datasets, quantify the phase change using $n$-gram context dependence, and further leverage these transitions to accelerate training of a smaller student model.

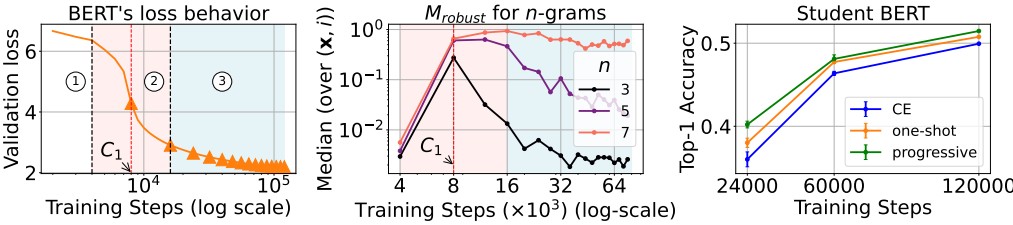

Figure 6: BERT on Wikipedia and Books. Left to right: (a) Similar to our experiments on PCFG (Figure 4), we observe three distinct phases in the loss behavior of 12-head teacher. The rapid loss drop phase signifies a transition phase for the model. The triangles mark the selected checkpoints for progressive distillation, with the first teacher checkpoint roughly picked in the middle of the second phase ($C_1$). (b) We observe $M_{\text{robust}}$ peaks at $C_1$, and the model gets more robust to shorter $n$-gram context masking, as training progresses. (c) A 4-head student achieves better top-1 accuracy on masked prediction objective with progressive distillation.

## 5 DISCUSSIONS

We have shown that progressive distillation can improve the student's feature learning via an implicit curriculum provided by the intermediate checkpoints. We discuss limitations and potential future directions below, and provide preliminary results for some of them in the appendix (see Appendix A).

**Impact of temperature.** The teacher temperature $\tau$ is an important hyperparameter in knowledge distillation, where varying $\tau$ can sometimes lead to a greater performance gain than changing the distillation method (Touvron et al., 2021; Harutyunyan et al., 2022). Our results are consistent with these prior findings. However, our experiments use limited temperature choices, i.e. the default ($\tau = 1.0$) and low temperature ($\tau = 10^{-4}$ or $10^{-20}$). A more precise understanding of temperature, especially its impact on optimization, is an interesting direction for future work.

**Distillation via generations.** Another related distillation setting is training smaller (language) models using the generations of larger models, which has been shown to greatly improve various abilities (Liu et al., 2024; Yue et al., 2023; Yu et al., 2023; Luo et al., 2023; Chaudhary, 2023; Taori et al., 2023; Zheng et al., 2023). There are two differences between our experiments and these generation-based approaches. First, the supervision in our experiments are distributions (over classes or the vocabulary), while generations are samples from distributions. Our experiments with a low or zero temperature provide positive evidence towards bridging this gap, but the precise effect remains to be explored. More importantly, given an input, there is a unique supervision in our settings, whereas there could be multiple generations given by multiple steps of unrolling of the teacher. Extending our framework to these generative setting will be an important direction for future work.

ACKNOWLEDGEMENT

AP and SM acknowledge funding from NSF, ONR, Simons Foundation, and DARPA. AR and BL are supported in part by NSF awards IIS-2211907, CCF-2238523, IIS-2403275, an Amazon Research Award, a Google Research Scholar Award, and an OpenAI Superalignment Fast Grant. BL also acknowledges the support of the Tang Family Presidential Fellowship. SG was supported in part by a Microsoft Accelerating Foundation Models Grant and an OpenAI Superalignment Fast Grant.

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

# Appendix

## Table of Contents

## A   OVERVIEW OF THE APPENDIX

The appendix provides omitted proofs and additional empirical explorations, which we outline below.

**Omitted proofs**   We will start with the proof of Theorem 3.2 in Appendix B. The main idea is to show that the teacher can develop stronger correlation to in-support variables than to off-support variables, which can then be utilized by the students to reduce sample complexity.

**Additional empirical results on sparse parity**   We present more experiments with MLP (Appendix C.1) and Transformers (Appendix C.2), as well as results on learning a hierarchical extension of sparse parity (Appendix C.3). For Transformer experiments, we study how scaling along different dimensions of the architecture, such as MLP width and number of attention heads, affects the search of support for sparse parity. We discuss the effect of temperature in Figure 14. For the hierarchical extension of sparse parity, we show that the implicit curriculum occurs in different phases, which suggests a natural choice for number of intermediate checkpoints used in progressive distillation.

**Masked prediction on PCFGs**   In Appendix D.5, we provide a formal definition of probablistic context-free grammar (PCFG) and introduce the PCFGs that we use from Allen-Zhu & Li (2023b). We then provide details of our experimental setup and conduct extensive ablation studies on training a BERT model using the masked prediction task with PCFG data. We experiment with variants of progressive distillation and confirm that they lead to improved performance on PCFGs, as measured

by accuracy and the three progress measures introduced in Section 4.1.1. Furthermore, we investigate the effect of temperature, masking rate, and PCFG variation in Appendix D.8.

**Next-token prediction on PCFGs** In Appendix E, we conduct next-token prediction experiments using GPT-2 models on PCFG "cfg3f", i.e. the most complex PCFG in Allen-Zhu & Li (2023a). We characterize conditions under which progressive distillation provides significant gains.

## A.1 ADDITIONAL RELATED WORKS

**Understanding knowledge distillation** There have been many works dedicated to understanding the effectiveness of knowledge distillation (Hinton et al., 2015; Mobahi et al., 2020; Menon et al., 2021; Dao et al., 2021; Nagarajan et al., 2024). For classification tasks, which are the focus of most knowledge distillation works, one intuitive explanation is that the teacher output provides a distribution over the class labels, which is more informative than the one-hot data labels. Menon et al. (2021) formalizes this intuition and shows that a teacher that provides the Bayes class probabilities leads to a tighter generalization gap. Motivated by their result and the observation that a high-accuracy teacher can be poorly calibrated, Ren et al. (2022) proposes to supervise the student using a moving average of the teacher across the training trajectory. While Ren et al. (2022) uses information of trajectory, their student learns from a fixed target throughout training, which is a major difference from progressive distillation. The teacher supervision also provides regularization benefits, such as controlling the bias-variance tradeoff (Zhou et al., 2020), encouraging sparsity (Mobahi et al., 2020), or as a form of label smoothing (Yuan et al., 2020).

**Learning sparse parity** There are well established hardness results for learning sparse parity. When given access to labels only, learning $(d, k)$-sparse parity with gradients from finite samples is an example of learning with statistical queries (SQ) (Kearns, 1998), for which a $\Omega(d^k)$ SQ computational lower bound applies (Edelman et al., 2023). When learning with a fully-connected network (MLP), these parallel queries correspond to a combination of model width (i.e. neurons) and training steps, [11] and hence the SQ lower bound implies a fundamental trade-off between the width, the number of training steps, and the number of samples (Edelman et al., 2023). In particular, given the same number of training steps, narrower models require more samples to learn parity.

**Feature learning** In this work, we use feature learning to refer to a learning process that recovers a low-dimensional "feature" which helps reduce sample complexity. Sparse parity is a task that can benefit from feature learning, where the feature is the support. For the special case of $k = 2$, Glasgow (2024) shows that feature learning using a jointly-optimized 2-layer neural network can reduce the sample complexity from $\Theta(d^2)$ (corresponding to learning with NTK (Wei et al., 2019; Ghorbani et al., 2019)) to $O(d\mathrm{poly}\log d)$. Sparse parity is an example of a single-/multi-index function, where the label is determined by a 1-dimensional/low-dimensional projection of the data. These functions have also been studied on Gaussian inputs (Nichani et al., 2022; Abbe et al., 2022; 2023; Damian et al., 2024a;b) and have known separation between neural networks (Abbe et al., 2022; 2023) and non-feature-learning kernel methods (Hsu, 2021).

**Benefit of width in optimization** Prior work has shown that width plays an important role in the optimization difficulty, where wider networks are more optimized easily. Du & Hu (2019) shows that sufficient width is necessary for the optimization on deep linear networks. Multiple works show that overparameterization leads to favorable optimization landscape, such as fewer sub-optimal local minima (Soudry & Hoffer, 2017; Soltanolkotabi et al., 2018) or guaranteed convergence at the limit (Chizat & Bach, 2018; 2020). Wider models also exhibit faster decaying loss empirically (Yang et al., 2022; Bordelon et al., 2024a). Most related to our focus on learning sparse parity, Edelman et al. (2023) relates the width to the number of parallel statistical queries (SQs). Combined with sparse parity's SQ lower bound, their result implies a trade-off where a larger width requires fewer optimization steps. Our work also acknowledges the benefit of width in optimization, but takes a different perspective by demonstrating that a smaller student can inherit the optimization benefit when learning from a higher-width teacher. Moreover, we consider the number of attention heads as another

---

[11]More precisely, it is a combination of width and steps, as well as the batch size which affects the precision of the stochastic gradient. We omit the impact of batch size here since we keep the batch size unchanged in the experiments.

---

**Algorithm 1** 2-stage training

---

**Require:** Stage lengths: $T_1, T_2$, learning rates $\eta_1, \eta_2$, batch size $B_1, B_2$, weight decay $\lambda_1, \lambda_2$.

    **for** $t \in [0, T_1]$ and all $i \in [m]$ **do**

        Sample $B_1$-samples $\{(\mathbf{x}^{(j)}, y^{(j)})\}_{j=1}^{B_1}$.

        Update the weights $\boldsymbol{w}_i$ as $\boldsymbol{w}_i^{(t)} \leftarrow \boldsymbol{w}_i^{(t-1)} - \eta_1 \mathbb{E}_{(\mathbf{x},y) \in \{(\mathbf{x}^{(j)}, y^{(j)})\}_{j=1}^{B_1}} \nabla_{\boldsymbol{w}_i} \left( L_{\theta^{(t)}}(\mathbf{x}, y) + \lambda_1 \|\boldsymbol{w}_i\|^2 \right)$.

    **end for**

    **for** $t \in [0, T_2]$ and all $i \in [m]$ **do**

        Sample $B_2$-samples $\{(\mathbf{x}^{(j)}, y^{(j)})\}_{j=1}^{B_2}$.

        Update the outer layer weights $a_i$ as

$$a_i^{(t+T_1)} \leftarrow a_i^{(t+T_1-1)} - \eta_2 \mathbb{E}_{(\mathbf{x},y) \in \{(\mathbf{x}^{(i)}, y^{(i)})\}_{j=1}^{B_2}} \nabla_{a_i} \left( L_{\theta^{(t+T_1-1)}}(\mathbf{x}, y) + \lambda_2 a_i^2 \right).$$

    **end for**

---

scaling dimension for Transformers, where the intuition is similar to having more "paths" (Dong et al., 2021). There have been results on studying the limiting output distribution as the number of attention heads goes to infinity (Hron et al., 2020; Bordelon et al., 2024b), though to our knowledge, there are no quantitative descriptions for finite number of heads.

# B    Proofs of results in Section 3.2

We provide the formal version of Theorem 3.2 in this section.

Recall that the teacher model is defined as

$$f_{\mathcal{T}}(\mathbf{x}) = \sum_{i=1}^{m} a_i \sigma \left( \langle \boldsymbol{w}_i, \mathbf{x} \rangle + b_i \right).$$

The student model is similarly defined as

$$f_{\mathcal{S}}(\mathbf{x}) = \sum_{i=1}^{\tilde{m}} \tilde{a}_i \sigma \left( \langle \tilde{\boldsymbol{w}}_i, \mathbf{x} \rangle + \tilde{b}_i \right).$$

**Setup**    We assume the data points are sampled at random from $\mathcal{U}(\{\pm 1\}^d)$. Without loss of generality, let the target $k$-sparse parity function be $y = x_1 x_2 \cdots x_k$. *Symmetric initialization:* Following (Barak et al., 2022), we use the following symmetric initialization: for each $1 \le i \le m/2$,

$$\boldsymbol{w}_i \sim \mathcal{U}(\{\pm 1\}^d), \quad b_i \sim \mathcal{U}(\{-1 + k^{-1}, \cdots, 1 - k^{-1}\}), \quad a_i \sim \mathcal{U}(\{\pm 1/m\}),$$
$$\boldsymbol{w}_{i+m/2} = \boldsymbol{w}_i, \quad b_{i+m/2} = b_i, \quad a_{i+m/2} = -a_i.$$

*Two-stage training:* Following prior work (Barak et al., 2022; Abbe et al., 2023; 2024b), we adopt a two-stage batch gradient descent training, where we first train the first-layer weights $\{\boldsymbol{w}_1, \cdots, \boldsymbol{w}_m\}$, keeping the output weights $\{a_i\}_{i=1}^{m}$ fixed. In the second stage of training, we fit the output weights $\{a_i\}_{i=1}^{m}$ while keeping others fixed. We keep the biases $\{b_i\}_{i=1}^{m}$ fixed throughout training. Similar strategy for training the student model as well. The teacher is trained with hinge loss, given by $\ell(\mathbf{x}, y) = \max(0, 1 - f_{\mathcal{T}}(\mathbf{x})y)$. The student is trained with $\ell_{\mathrm{DL}}(\mathbf{x}, y; f_{\mathcal{S}}, f_{\mathcal{T}}) = \max(0, 1 - f_{\mathcal{S}}(\mathbf{x}) f_{\mathcal{T}}(\mathbf{x}))$.

The training process is summarized in Algorithm 1.

**Sample complexity benefits with progressive distillation for the student**    Our result is that progressive distillation provably reduces the sample complexity compared to (one-shot) distillation or no distillation. The key is to establish a separation between the correlations with in-support and off-support variables, which happens with high probability as formalized in Corollary B.6. Under such event, we show:

---

**Corollary B.6: conditions satisfied by the teacher after first phase**

W.h.p. the output of the teacher after the first phase satisfies the following condition for all $i$.

$$\left| \mathbb{E}_{\mathbf{x},y} f_{\mathcal{T}}^{(1)}(\mathbf{x}) \cdot \mathrm{Maj}(\mathbf{x}) x_i \right| \geq \Omega(k^{-1}), \quad \text{if } i \in [k],$$

$$\left| \mathbb{E}_{\mathbf{x},y} f_{\mathcal{T}}^{(1)}(\mathbf{x}) \cdot \mathrm{Maj}(\mathbf{x}) x_i \right| \leq o(k^{-1}), \quad \text{if } i \notin [k].$$

---

**Theorem B.1** (Sample complexity benefits with progressive distillation). *Suppose the teacher model has been trained with 2-stage training in Algorithm 1, which satisfies the conditions in Corollary B.6 at the end of first stage and achieves loss $\mathcal{O}(d^{-c})$ for some constant $c \geq 1$ at the end of the second stage. Suppose we train a student model $f_{\mathcal{S}}$ of size $\tilde{m} = \tilde{\Theta}(2^k k)$ using the following two strategies:*

1. ***Progressive distillation:*** *Train for the first $T_1 = 1$ steps w.r.t. the teacher's logits at $T_1$ checkpoint. Then, train with the final teacher checkpoint in the second stage.*

2. ***Distillation:*** *Train with the final teacher checkpoint throughout training.*

*Then,*

1. *Under progressive distillation, the total sample complexity to reach a loss of $\epsilon$ with probability $1 - \delta$ is*
$$\Theta(k^2 \log(d\tilde{m}/\delta) + 2^k d^2 k^4 \epsilon^{-2} \log(k/\delta)).$$

2. *The necessary sample complexity under distillation is at least $\Omega(d^{\min(2c, k-1)})$.*

The proof consists of two parts: 1) showing that the teacher develops strong correlation with the in-support variables after the first stage of training (Lemma B.2, Corollary B.6), and 2) showing that given the support, the second phase of training converges quickly (Corollary B.8). These two helper lemmas are proven in Appendix B.1.1 (first stage) and Appendix B.1.2 (second stage). The proof of Theorem B.1 is given in Appendix B.2.

**Notations** Before stating the proofs, we provide a list of necessary notations.

- At any training step $t$, $f_{\mathcal{T}}^{(t)}$ will refer to the teacher's output at that step. Its parameters are referred to as $\theta^{(t)} = \{a_i^{(t)}, \boldsymbol{w}_i^{(t)}, b_i^{(t)}\}_{i=1}^m$. The loss for $f_{\mathcal{T}}^{(t)}$ is denoted by $L(f_{\mathcal{T}}^{(t)})$ or $L_{\theta^{(t)}}$. Notations for the student $f_{\mathcal{S}}$ are defined similarly.
- Given a set $\tilde{S}$, $\chi_{\tilde{S}}$ denotes the Fourier function on $\tilde{S}$, where $\chi_{\tilde{S}}(\mathbf{x}) = \prod_{i \in \tilde{S}} x_i$. We are particularly interested in $\tilde{S} = \mathbf{S}$, i.e. the support of the sparse parity.
- $\mathrm{Maj} : \{\pm 1\}^d \to \pm 1$ represents the majority function. On any $\mathbf{x}$, $\mathrm{Maj}$ returns the sign of $\sum_{i=1}^d \mathbf{x}_i$. $\zeta_i$ for $i \geq 1$ represents its $i$th fourier coefficient, i.e. $\zeta_i = \mathbb{E}_{\mathbf{x},y} \mathrm{Maj}(\mathbf{x}) \chi_S(\mathbf{x})$ for any $S \in \{0, 1\}^d$ with $|S| = i$. $\zeta_i = 0$ when $i$ is even, and $\zeta_i = \Theta(i^{-1/3}/\binom{d}{i})$ when $i$ is odd (O'Donnell, 2014).
- $\tau_g$ denotes the error tolerance in the gradient estimate due to mini-batch gradient estimation: let $g$ be the population gradient and $\hat{g}$ be the estimated gradient with a few examples, $\tau_g$ is defined such that $\|\hat{g} - g\|_\infty \leq \tau_g$. A $\tau_g$-error gradient estimate can be obtained using a batch size of $\tilde{\Omega}(1/\tau_g^2)$.

## B.1 ANALYSIS FOR THE TEACHER

### B.1.1 FIRST STAGE ANALYSIS FOR THE TEACHER

First, we show that with an appropriate learning rate, the magnitude of the weights $w_{ij}$ on coordinates $i \in \mathbf{S}$ increases to $\frac{1}{2k}$, while the coordinates $i \notin \mathbf{S}$ stay $\mathcal{O}\left(\frac{1}{kd}\right)$ small.

**Lemma B.2** (Single step gradient descent, adapted from Claims 1, 2 in Barak et al. (2022)). *Fix $\tau_g, \delta > 0$. Set $T_1$ as 1. Suppose the batch size $B_1 \geq \Omega(\tau_g^{-2} \log(md/\delta))$. For learning rate $\eta_1 = \frac{m}{k|\zeta_{k-1}|}$ and $\lambda_1 = 1$, the following conditions hold true for all neurons $i \in [m]$ at the end of first stage of training w.p. at least $1 - \delta$.*

1. $\left| w_{ij}^{(1)} - \frac{\operatorname{sign}(a_i^{(0)} \zeta_{k-1}) \operatorname{sign}(\chi_{[k]\setminus\{j\}}(\boldsymbol{w}_i^{(0)}))}{2k} \right| \le \frac{\tau_g}{|\zeta_{k-1}|}, \text{for all } j \in [k].$

2. $\left| w_{ij}^{(1)} - \frac{\zeta_{k+1}}{|\zeta_{k-1}|} \frac{\operatorname{sign}(a_i^{(0)}) \operatorname{sign}(\chi_{[k]\cup\{j\}}(\boldsymbol{w}_i^{(0)}))}{2k} \right| \le \frac{\tau_g}{|k\zeta_{k-1}|}, \text{for all } j > k.$

*Proof.* The proof follows that of (Barak et al., 2022), which we outline here for completeness. The proof has two major components: First, the magnitude of the population gradient at initialization reveals the support of the sparse parity. Second, the batch gradient and the population gradient can be made sufficiently close given a sufficiently large batch size. We will explain each step below.

**Claim B.3.** *At initialization, the population gradient of the weight vector in neuron $i$ is given by* $\mathbb{E}_{\mathbf{x},y} \nabla_{w_{ij}} \ell(\mathbf{x}, y; f_{\mathcal{T}}^{(0)}) = -\mathbb{E}_{\mathbf{x},y} \nabla_{w_{ij}} f_{\mathcal{T}}^{(0)}(\mathbf{x}) y$, *which can be split across the coordinates as*

$$\mathbb{E}_{\mathbf{x},y} \nabla_{w_{ij}} f_{\mathcal{T}}^{(0)}(\mathbf{x}) y = -\frac{1}{2} a_i^{(0)} \zeta_{k-1} \chi_{[k]\setminus\{j\}}(\boldsymbol{w}^{(0)}), \quad \text{for all } j \in \mathbf{S}$$

$$\mathbb{E}_{\mathbf{x},y} \nabla_{w_{ij}} f_{\mathcal{T}}^{(0)}(\mathbf{x}) y = -\frac{1}{2} a_i^{(0)} \zeta_{k+1} \chi_{[k]\cup\{j\}}(\boldsymbol{w}^{(0)}), \quad \text{for all } j \notin \mathbf{S}$$

Thus, the gradient of the weight coordinates $w_{ij}$ for any neuron $i$ and $j \in \mathbf{S}$ has magnitude $|\zeta_{k-1}|$, while the gradients of the weight coordinates $w_{ij}$ for any neuron $i$ and $j \notin \mathbf{S}$ has magnitude $|\zeta_{k+1}|$. The gap between the gradient in support and out of support is given by $|\zeta_{k-1}| - |\zeta_{k+1}| \ge 0.03((d-1)^{-(k-1)/2})$ (Lemma 2 in Barak et al. (2022)).

The second component involves applying a hoeffding's inequality to show the gap between sample and population gradient.

**Claim B.4.** *Fix $\delta, \tau_g > 0$. For all $i, j$, for a randomly sampled batch of size $B_1$, $\{(\mathbf{x}_k, y_k)\}_{k=1}^{B_1}$, with probability at least $1 - \delta$,*

$$\left| \mathbb{E}_{\mathbf{x},y \sim \mathcal{U}(\{\pm\}^d)} \nabla_{w_{ij}} f_{\mathcal{T}}^{(0)}(\mathbf{x}) - \mathbb{E}_{\{(\mathbf{x}_k, y_k)\}_{k=1}^{B_1}} \nabla_{w_{ij}} f_{\mathcal{T}}^{(0)}(\mathbf{x}) \right| \le \tau_g,$$

*provided $B_1 \ge \Omega(\tau_g^{-2} \log(md/\delta))$.*

Because we want the noise $\tau_g$ to be smaller than the magnitude of the true gradients for the coordinates in the support $\mathbf{S}$, we want $\tau_g$ to be smaller than $|\zeta_{k-1}|$. We set this to get favorable condition for second phase of training (see Lemma B.7). $\square$

On the other hand, we show that after the first phase, the output of the network has positive correlations to the individual variables in the support of the label function, and thus the checkpoint after the first phase can be used to speed up training of future models.

**Lemma B.5** (Correlation with in-support variables). *Under the event that the conditions in Lemma B.2 are satisfied by each neuron, which occurs with probability at least $1 - \delta$ w.r.t. the randomness of initialization and sampling, the output of the model after the first phase satisfies the following conditions:*

1. $\mathbb{E}_{\mathbf{x},y} f_{\mathcal{T}}^{(1)}(\mathbf{x}) x_i \ge \frac{1}{8k} + \mathcal{O}(\tau_g d |\zeta_{k-1}|^{-1}) + \mathcal{O}(m^{-1/2})$ *for all $i \in \mathbf{S}$.*

2. $\mathbb{E}_{\mathbf{x},y} f_{\mathcal{T}}^{(1)}(\mathbf{x}) x_i \le \mathcal{O}((kd)^{-1})$ *for all $i \notin \mathbf{S}$.*

3. $\mathbb{E}_{\mathbf{x},y} f_{\mathcal{T}}^{(1)}(\mathbf{x}) \chi_S(\mathbf{x}) \le \mathcal{O}(\tau_g d |\zeta_{k-1}|^{-1})$ *for all $S$ with even $|S|$.*

4. $\left\| f_{\mathcal{T}}^{(1)} \right\|_2^2 = \mathbb{E}_{\mathbf{x},y}[f_{\mathcal{T}}^{(1)}(\mathbf{x})]^2 \le \mathcal{O}(d/k)$.

*Proof.* Consider a neuron $i \in [m/2]$ and its symmetric counterpart $i + m/2$. W.L.O.G., we assume $\operatorname{sign}(w_{ij}^{(0)}) = \operatorname{sign}(a_i^{(0)} \zeta_{k-1})$ for all $j \in [k]$, and $\operatorname{sign}(a_i^{(0)}) = 1$. Recall that $k$ is assumed to be even,

hence $\text{sign}(\chi_{[k]}(\boldsymbol{w}_i^{(0)})) = 1$. Then, the condition in Lemma B.2 can be simplified as

$$w_{ij}^{(1)} = \frac{1}{2k} + v_{ij}, \quad w_{i+m/2,j}^{(1)} = -\frac{1}{2k} - v_{ij}, \text{ for all } j \in [k],$$

$$w_{ij}^{(1)} = \frac{1}{2k}\frac{\zeta_{k+1}}{|\zeta_{k-1}|}\text{sign}(w_{ij}^{(0)}) + v_{ij}, \quad w_{i+m/2,j}^{(1)} = -\frac{1}{2k}\frac{\zeta_{k+1}}{|\zeta_{k-1}|}\text{sign}(w_{ij}^{(0)}) + v_{ij}, \text{ for all } j \geq k,$$

where $v_{ij}$ satisfies the following conditions.

$$|v_{ij}| \leq \frac{\tau_g}{|\zeta_{k-1}|}, \text{ for all } j \in [k],$$

$$|v_{ij}| \leq \frac{\tau_g}{|k\zeta_{k-1}|}, \text{ for all } j \geq k.$$

Then, the sum of the output of the neurons $i$ and $i + m/2$ on an input $\mathbf{x}$ (ignoring the magnitude of $a_i$) is given by

$$(f_{\mathcal{T}}^{(1)})_i(\mathbf{x}) = \sigma\left(\frac{1}{2k}\sum_{j=1}^{k}x_j + \frac{1}{2k}\frac{\zeta_{k+1}}{|\zeta_{k-1}|}\sum_{j=k+1}^{d}\text{sign}(w_{ij}^{(0)})x_j + \langle\boldsymbol{v}_i,\mathbf{x}\rangle + b_i\right)$$

$$- \sigma\left(-\frac{1}{2k}\sum_{j=1}^{k}x_j - \frac{1}{2k}\frac{\zeta_{k+1}}{|\zeta_{k-1}|}\sum_{j=k+1}^{d}\text{sign}(w_{ij}^{(0)})x_j + \langle\boldsymbol{v}_i,\mathbf{x}\rangle + b_i\right),$$

and

$$f_{\mathcal{T}}^{(1)}(\mathbf{x}) = \sum_{i=1}^{m/2}a_i(f_{\mathcal{T}}^{(1)})_i(\mathbf{x}) = \frac{1}{m}\sum_{i=1}^{m/2}(f_{\mathcal{T}}^{(1)})_i(\mathbf{x}).$$

**1. In-support correlations:** We are interested in the correlation of this function to a variable $x_u$ for $u \in \mathbf{S}$. We argue for $u = 1$, as the similar argument applies for others. Thus, we are interested in

$$\mathbb{E}_{\mathbf{x},y}(f_{\mathcal{T}}^{(1)})_i(\mathbf{x})x_1 = \mathbb{E}_{\mathbf{x},y}\sigma\left(\frac{1}{2k}\sum_{j=1}^{k}x_j + \frac{1}{2k}\frac{\zeta_{k+1}}{|\zeta_{k-1}|}\sum_{j=k+1}^{d}\text{sign}(w_{ij}^{(0)})x_j + \langle\boldsymbol{v}_i,\mathbf{x}\rangle + b_i\right)x_1$$

$$- \sigma\left(-\frac{1}{2k}\sum_{j=1}^{k}x_j - \frac{1}{2k}\frac{\zeta_{k+1}}{|\zeta_{k-1}|}\sum_{j=k+1}^{d}\text{sign}(w_{ij}^{(0)})x_j + \langle\boldsymbol{v}_i,\mathbf{x}\rangle + b_i\right)x_1.$$

$$(7)$$

We focus on the first term; argument for the second term is similar. First of all, we can ignore $\langle\boldsymbol{v}_i,\mathbf{x}\rangle$ incurring an error of $\mathcal{O}(\tau_g d\,|\zeta_{k-1}|^{-1})$.

$$\mathbb{E}_{\mathbf{x},y}\sigma\left(\frac{1}{2k}\sum_{j=1}^{k}x_j + \frac{1}{2k}\frac{\zeta_{k+1}}{|\zeta_{k-1}|}\sum_{j=k+1}^{d}\text{sign}(w_{ij}^{(0)})x_j + b_i\right)x_1$$

$$= \mathbb{E}_{\mathbf{x},y:x_1=+1}\sigma\left(\frac{1}{2k} + \frac{1}{2k}\sum_{j=2}^{k}x_j + \frac{1}{2k}\frac{\zeta_{k+1}}{|\zeta_{k-1}|}\sum_{j=k+1}^{d}\text{sign}(w_{ij}^{(0)})x_j + b_i\right)$$

$$- \mathbb{E}_{\mathbf{x},y:x_1=-1}\sigma\left(-\frac{1}{2k} + \frac{1}{2k}\sum_{j=2}^{k}x_j + \frac{1}{2k}\frac{\zeta_{k+1}}{|\zeta_{k-1}|}\sum_{j=k+1}^{d}\text{sign}(w_{ij}^{(0)})x_j + b_i\right)$$

$$\geq \frac{1}{2k}\mathbb{E}_{\mathbf{x},y}\mathbb{I}\left(\frac{1}{2k}\sum_{j=2}^{k}x_j + \frac{1}{2k}\frac{\zeta_{k+1}}{|\zeta_{k-1}|}\sum_{j=k+1}^{d}\text{sign}(w_{ij}^{(0)})x_j + b_i \geq 0\right).$$

The final step follows from the observation that the argument of $\sigma$ in the first term is $\frac{1}{k}$ higher than the argument of $\sigma$ in the second term. This implies that when the first term is non-zero, it's at least $\frac{1}{2k}$ higher than the second term. Hence, we lower bound by considering one scenario where the first term is non-zero.

Continuing, we can further split the indicator function into cases when each term in the argument of the indicator function is positive.

$$
\mathbb{E}_{\mathbf{x},y}\sigma\left(\frac{1}{2k}\sum_{j=1}^{k}x_j + \frac{1}{2k}\frac{\zeta_{k+1}}{|\zeta_{k-1}|}\sum_{j=k+1}^{d}\mathrm{sign}(w_{ij}^{(0)})x_j + b_i\right)x_1
$$

$$
\geq \frac{1}{2k}\mathbb{E}_{\mathbf{x},y}\mathbb{I}\left(\frac{1}{2k}\sum_{j=2}^{k}x_j + \frac{1}{2k}\frac{\zeta_{k+1}}{|\zeta_{k-1}|}\sum_{j=k+1}^{d}\mathrm{sign}(w_{ij}^{(0)})x_j + b_i \geq 0\right)
$$

$$
\geq \frac{1}{2k}\mathbb{E}_{\mathbf{x},y}\mathbb{I}\left(\sum_{j=2}^{k}x_j \geq 0\right)\mathbb{I}\left(\sum_{j=k+1}^{d}x_j \geq 0\right)\mathbb{I}\left(b_i \geq 0\right)
$$

$$
\geq \frac{1}{8k}\mathbb{I}\left(b_i \geq 0\right).
$$

From Equation (7), we then have

$$
\mathbb{E}_{\mathbf{x},y}(f_{\mathcal{T}}^{(1)})_i(\mathbf{x})x_1 \geq \frac{1}{4k}\mathbb{I}\left(b_i \geq 0\right) + \mathcal{O}(\tau_g d\,|\zeta_{k-1}|^{-1}).
$$

As $b_i$ has been kept at random initialization and thus is a random variable selected from the set $\{-1+\frac{1}{k}, \cdots, 1-\frac{1}{k}\}$, with probability $\frac{1}{2}$, $\mathbb{I}(b_i \geq 0)$. This implies, w.p. atleast $1/2$ w.r.t. a neuron's bias initialization, $\mathbb{E}_{\mathbf{x},y}(f_{\mathcal{T}}^{(1)})_i(\mathbf{x})x_1 \geq \frac{1}{4k} + \mathcal{O}(\tau_g d\,|\zeta_{k-1}|^{-1})$. The final bound comes from the fact that $\mathbb{E}_{\mathbf{x},y}f_{\mathcal{T}}(\mathbf{x})x_1 = \mathbb{E}_{\mathbf{x},y}\frac{1}{m}\sum_{i=1}^{m}(f_{\mathcal{T}}^{(1)})_i(\mathbf{x})x_1 \geq \frac{1}{8k} + \mathcal{O}(\tau_g d\,|\zeta_{k-1}|^{-1}) + \mathcal{O}(m^{-1/2})$, where the error term is bounded using Hoeffding's inequality.

**2. Out-of-support correlations:** Similar to the Equation (7), we have for $u \notin \mathbf{S}$,

$$
\mathbb{E}_{\mathbf{x},y}(f_{\mathcal{T}}^{(1)})_i(\mathbf{x})x_u = \mathbb{E}_{\mathbf{x},y}\sigma\left(\frac{1}{2k}\sum_{j=1}^{k}x_j + \frac{1}{2k}\frac{\zeta_{k+1}}{|\zeta_{k-1}|}\sum_{j=k+1}^{d}\mathrm{sign}(w_{ij}^{(0)})x_j + \langle \boldsymbol{v}_i, \mathbf{x}\rangle + b_i\right)x_u
$$

$$
- \sigma\left(-\frac{1}{2k}\sum_{j=1}^{k}x_j - \frac{1}{2k}\frac{\zeta_{k+1}}{|\zeta_{k-1}|}\sum_{j=k+1}^{d}\mathrm{sign}(w_{ij}^{(0)})x_j + \langle \boldsymbol{v}_i, \mathbf{x}\rangle + b_i\right)x_u.
$$

$$\tag{8}$$

However, we observe that the influence of $x_u$ in each of the terms is bounded by $\frac{1}{k}\frac{\zeta_{k+1}}{|\zeta_{k-1}|}$. Consider the first term; the argument for the second term is similar. We can again ignore $\langle \boldsymbol{v}_i, \mathbf{x}\rangle$ incurring an error of $\mathcal{O}(\tau_g d\,|\zeta_{k-1}|^{-1})$.

$$\mathbb{E}_{\mathbf{x},y}\sigma\left(\frac{1}{2k}\sum_{j=1}^{k}x_j + \frac{1}{2k}\frac{\zeta_{k+1}}{|\zeta_{k-1}|}\sum_{j=k+1}^{d}\text{sign}(w_{ij}^{(0)})x_j + b_i\right)x_u$$

$$= \mathbb{E}_{\mathbf{x},y:x_u=+1}\sigma\left(\frac{1}{2k}\frac{\zeta_{k+1}}{|\zeta_{k-1}|}\text{sign}(w_{iu}^{(0)}) + \frac{1}{2k}\sum_{j=1}^{k}x_j + \frac{1}{2k}\frac{\zeta_{k+1}}{|\zeta_{k-1}|}\sum_{j=k+1\to d;j\neq u}\text{sign}(w_{ij}^{(0)})x_j + b_i\right)$$

$$- \mathbb{E}_{\mathbf{x},y:x_u=-1}\sigma\left(-\frac{1}{2k}\frac{\zeta_{k+1}}{|\zeta_{k-1}|}\text{sign}(w_{iu}^{(0)}) + \frac{1}{2k}\sum_{j=1}^{k}x_j + \frac{1}{2k}\frac{\zeta_{k+1}}{|\zeta_{k-1}|}\sum_{j=k+1\to d;j\neq u}\text{sign}(w_{ij}^{(0)})x_j + b_i\right)$$

$$= \mathbb{E}_{\mathbf{x},y}\frac{C(\mathbf{x})}{k}\frac{\zeta_{k+1}}{|\zeta_{k-1}|}\text{sign}(w_{iu}^{(0)})\mathbb{I}\left(\frac{1}{2k}\sum_{j=1}^{k}x_j + \frac{1}{2k}\frac{\zeta_{k+1}}{|\zeta_{k-1}|}\sum_{j=k+1\to d;j\neq u}\text{sign}(w_{ij}^{(0)})x_j + b_i \geq 0\right),$$

where $C(\mathbf{x}) \in \{1,2\}$ denotes a function that depends on $\mathbf{x}$. The final step follows from a first order taylor expansion of $\sigma$. The magnitude can hence be bounded by $\frac{1}{k}\frac{|\zeta_{k+1}|}{|\zeta_{k-1}|}$. This can be bounded by $\frac{1}{kd}$ (section 5.3, O'Donnell (2014)). The final bound comes from the fact that $\mathbb{E}_{\mathbf{x},y}f_{\mathcal{T}}(\mathbf{x})x_u = \mathbb{E}_{\mathbf{x},y}\frac{1}{m}\sum_{i=1}^{m}(f_{\mathcal{T}}^{(1)})_i(\mathbf{x})x_u \leq \mathcal{O}((kd)^{-1})$.

**3. Correlations to support of an even size:** The function $(f_{\mathcal{T}}^{(1)})_i$ is given by

$$(f_{\mathcal{T}}^{(1)})_i(\mathbf{x}) = \sigma\left(\frac{1}{2k}\sum_{j=1}^{k}x_j + \frac{1}{2k}\frac{\zeta_{k+1}}{|\zeta_{k-1}|}\sum_{j=k+1}^{d}\text{sign}(w_{ij}^{(0)})x_j + \langle\boldsymbol{v}_i,\mathbf{x}\rangle + b_i\right)$$

$$- \sigma\left(-\frac{1}{2k}\sum_{j=1}^{k}x_j - \frac{1}{2k}\frac{\zeta_{k+1}}{|\zeta_{k-1}|}\sum_{j=k+1}^{d}\text{sign}(w_{ij}^{(0)})x_j + \langle\boldsymbol{v}_i,\mathbf{x}\rangle + b_i\right)$$

$$= \sigma\left(\frac{1}{2k}\sum_{j=1}^{k}x_j + \frac{1}{2k}\frac{\zeta_{k+1}}{|\zeta_{k-1}|}\sum_{j=k+1}^{d}\text{sign}(w_{ij}^{(0)})x_j + b_i\right)$$

$$- \sigma\left(-\frac{1}{2k}\sum_{j=1}^{k}x_j - \frac{1}{2k}\frac{\zeta_{k+1}}{|\zeta_{k-1}|}\sum_{j=k+1}^{d}\text{sign}(w_{ij}^{(0)})x_j + b_i\right) + \mathcal{O}(\tau_g d|\zeta_{k-1}|^{-1})$$

$$:= g(\mathbf{x}) + \mathcal{O}(\tau_g d|\zeta_{k-1}|^{-1}).$$

One can observe that $g(\mathbf{x})$ is a symmetric function and so an odd function. Thus, $\mathbb{E}_{\mathbf{x},y}g(\mathbf{x})\chi_S(\mathbf{x}) = 0$ (exercise 1.8, O'Donnell (2014)) and so, $\mathbb{E}_{\mathbf{x},y}(f_{\mathcal{T}}^{(1)})_i(\mathbf{x})\chi_S(\mathbf{x}) = \mathcal{O}(\tau_g d|\zeta_{k-1}|^{-1})$.

**4. Output norm:** Focusing on function $(f_{f_{\mathcal{T}}}^{(1)})_i$:

$$\left\|(f_{\mathcal{T}}^{(1)})_i\right\|_2^2 = \mathbb{E}_{\mathbf{x},y}(f_{\mathcal{T}}^{(1)})_i(\mathbf{x})^2$$

$$= \mathbb{E}_{\mathbf{x},y}\left(\sigma(\langle w_{ij}^{(1)},\mathbf{x}\rangle + b_i) - \sigma(\langle w_{i+m/2,j}^{(1)},\mathbf{x}\rangle + b_i)\right)^2$$

$$\leq \mathbb{E}_{\mathbf{x},y}\min\left(\left\|\boldsymbol{w}_i^{(1)}\right\|_2^2 + b_i^2, \left\|\boldsymbol{w}_{i+m/2}^{(1)}\right\|_2^2 + b_i^2\right)\|\mathbf{x}\|_2^2 = \mathcal{O}\left(\frac{1}{k}\right)\cdot d.$$

The intermediate step uses Cauchy-Schwartz inequality, and the final step uses the values of $w_{ij}^{(1)}, w_{i+m/2,j}^{(1)}$. As $f_{\mathcal{T}}^{(1)}(\mathbf{x}) = \frac{1}{m}\sum_{i=1}^{m/2}(f_{\mathcal{T}}^{(1)})_i(\mathbf{x})$, we have $\left\|(f_{\mathcal{T}}^{(1)})\right\|_2^2 \leq \frac{2}{m}\sum_{i=1}^{m/2}\left\|(f_{\mathcal{T}}^{(1)})_i\right\|_2^2 = \mathcal{O}\left(\frac{d}{k}\right)$. $\qquad\square$

**Corollary B.6.** *Under the event that the conditions in Lemma B.2 are satisfied by each neuron, which occurs with probability at least $1 - \delta$ w.r.t. the randomness of initialization and sampling, the output of the model after the first phase can be given as:*

$$f_{\mathcal{T}}^{(1)}(\mathbf{x}) = \sum_{j=1}^{k} c_j x_j + \sum_{j=k+1}^{d} c_j x_j + \sum_{S \subseteq [d]: |S|\%2=1, |S|\geq 3} c_S \chi_S(\mathbf{x}) + \sum_{S \subseteq [d]: |S|\%2=0} c_S \chi_S(\mathbf{x}),$$

*where*

$$\begin{aligned}
|c_j| &\geq \Omega(k^{-1}), && \text{for all } 1 \leq j \leq k, \\
|c_j| &\leq \mathcal{O}((kd)^{-1}), && \text{for all } j > k, \\
|c_S| &\leq \mathcal{O}(\tau_g d \,|\zeta_{k-1}|^{-1}), && \text{for all } S \subseteq [d] \text{ with } |S|\%2=0, \\
|c_S| &\leq \mathcal{O}(d/k), && \text{for all } S \subseteq [d] \text{ with } |S|\%2=1.
\end{aligned}$$

*As such, the following correlations hold true for all $i$.*

$$\mathbb{E}_{\mathbf{x},y} f_{\mathcal{T}}^{(1)}(\mathbf{x}) \cdot Maj(\mathbf{x}) x_i = \frac{1}{2} c_i + \mathcal{O}(\tau_g d^{5/3} \,|\zeta_{k-1}|^{-1}).$$

*If batch size $B_1$ is set $\geq \Omega(k^2 d^{10/3} \zeta_{k-1}^{-2})$, such that $\tau_g \leq \mathcal{O}(k^{-1} d^{-5/3} \,|\zeta_{k-1}|)$, then the following holds for all $i$.*

$$\begin{aligned}
\left| \mathbb{E}_{\mathbf{x},y} f_{\mathcal{T}}^{(1)}(\mathbf{x}) \cdot Maj(\mathbf{x}) x_i \right| &\geq \Omega(k^{-1}), && \text{if } i \in [k], \\
\left| \mathbb{E}_{\mathbf{x},y} f_{\mathcal{T}}^{(1)}(\mathbf{x}) \cdot Maj(\mathbf{x}) x_i \right| &\leq o(k^{-1}), && \text{if } i \notin [k],
\end{aligned}$$

*Proof.* The form of $f_{\mathcal{T}}^{(1)}$ follows from the fourier coefficient analysis in Lemma B.5.

Now, we can use the formulation to derive

$$\begin{aligned}
&\mathbb{E}_{\mathbf{x},y} f_{\mathcal{T}}^{(1)}(\mathbf{x}) \cdot \text{Maj}(\mathbf{x}) x_i \\
=& \mathbb{E}_{\mathbf{x},y} \sum_{j=1}^{d} c_j x_j \cdot \text{Maj}(\mathbf{x}) \cdot x_i + \mathbb{E}_{\mathbf{x},y} \sum_{S \subseteq [d]: |S|\%2=1, |S|\geq 3} c_S \text{Maj}(\mathbf{x}) \chi_S(\mathbf{x}) \cdot x_i \\
&+ \mathbb{E}_{\mathbf{x},y} \sum_{S \subseteq [d]: |S|\%2=0} c_S \chi_S(\mathbf{x}) \cdot \text{Maj}(\mathbf{x}) x_i \\
=& \mathbb{E}_{\mathbf{x},y} \sum_{j=1}^{d} c_j x_j \cdot \text{Maj}(\mathbf{x}) \cdot x_i + \mathbb{E}_{\mathbf{x},y} \sum_{S \subseteq [d]: |S|\%2=0} c_S \text{Maj}(\mathbf{x}) \chi_S(\mathbf{x}) \cdot x_j \\
=& c_i \mathbb{E}_{\mathbf{x},y} \text{Maj}(\mathbf{x}) + \mathbb{E}_{\mathbf{x},y} \sum_{j, j \neq k} c_j \text{Maj}(\mathbf{x}) x_j x_i + \mathbb{E}_{\mathbf{x},y} \sum_{S \subseteq [d]: |S|\%2=0} c_S \text{Maj}(\mathbf{x}) \chi_S(\mathbf{x}) \cdot x_i \\
=& \frac{1}{2} c_i + \mathbb{E}_{\mathbf{x},y} \sum_{S \subseteq [d]: |S|\%2=0} c_S \text{Maj}(\mathbf{x}) \chi_S(\mathbf{x}) \cdot x_i.
\end{aligned}$$

The second step removes $\mathbb{E}_{\mathbf{x},y} \sum_{S \subseteq [d]: |S|\%2=0} c_S \chi_S(\mathbf{x}) \cdot \text{Maj}(\mathbf{x}) x_i$ because $\text{Maj}(\mathbf{x})$ is an odd function, and so $\mathbb{E}_{\mathbf{x},y} \text{Maj}(\mathbf{x}) \chi_S(\mathbf{x}) x_i$ will be $0$ for odd sized $S$. Similar argument holds for removing $\mathbb{E}_{\mathbf{x},y} \sum_{j, j \neq i} c_j \text{Maj}(\mathbf{x}) x_j x_i$ in the final step. We finish the proof by bounding $\mathbb{E}_{\mathbf{x},y} \sum_{S \subseteq [d]: |S|\%2=0} c_S \text{Maj}(\mathbf{x}) \chi_S(\mathbf{x}) \cdot x_i$.

As $|c_S| \leq \mathcal{O}(\tau_g d \, |\zeta_{k-1}|^{-1})$ for all $S$ with $|S|\%2 = 0$, we can bound it as

$$\left| \mathbb{E}_{\mathbf{x},y} \sum_{S \subseteq [d]:|S|\%2=0} c_S \mathrm{Maj}(\mathbf{x}) \chi_S(\mathbf{x}) \cdot x_i \right|$$

$$\leq \mathcal{O}(\tau_g d \, |\zeta_{k-1}|^{-1}) \cdot \left( \sum_{S \subseteq [d]:|S|\%2=0} |\mathbb{E}_{\mathbf{x},y} \mathrm{Maj}(\mathbf{x}) \chi_S(\mathbf{x}) x_i| \right)$$

$$\leq \mathcal{O}(\tau_g d \, |\zeta_{k-1}|^{-1}) \cdot \left( \sum_{S \subseteq [d]} |\mathbb{E}_{\mathbf{x},y} \mathrm{Maj}(\mathbf{x}) \chi_S(\mathbf{x})| \right)$$

$$\leq \mathcal{O}(\tau_g d \, |\zeta_{k-1}|^{-1}) \cdot \left( \sum_{S \subseteq [d]} |\mathbb{E}_{\mathbf{x},y} \mathrm{Maj}(\mathbf{x}) \chi_S(\mathbf{x})| \right)$$

$$= \mathcal{O}(\tau_g d \, |\zeta_{k-1}|^{-1}) \cdot \sum_{S \subseteq [d]} \Theta \left( \frac{|S|^{-1/3}}{\binom{d}{|S|}} \right)$$

$$= \mathcal{O}(\tau_g d^{5/3} \, |\zeta_{k-1}|^{-1}).$$

Here the pre-final step follows from the bounds on the Fourier coefficients of Maj outlined in Appendix B. Finally, we set $B_1 \geq \Omega(\tau_g^{-2})$ is set such that $\tau_g \leq \mathcal{O}(k^{-1} d^{-5/3} \zeta_{k-1})$. This makes $\mathcal{O}(\tau_g d^{5/3} \, |\zeta_{k-1}|^{-1}) = o(1/k)$. Hence, with appropriate batch size $B_1$,

$$\mathbb{E}_{\mathbf{x},y} f_{\mathcal{T}}^{(1)}(\mathbf{x}) \cdot \mathrm{Maj}(\mathbf{x}) x_i = \frac{1}{2} c_i + o(1/k).$$

The proof follows from the magnitude of $c_i$ derived above. $\qquad \square$

### B.1.2 SECOND STAGE ANALYSIS FOR THE TEACHER

**Lemma B.7** (Second stage Training, cf. Theorem 4 in (Barak et al., 2022)). *Fix $\epsilon, \delta > 0$. Suppose $m \geq \Omega(2^k k \log(k/\delta))$, $d \geq \Omega\left(k^4 \log(kd/\epsilon)\right)$. Furthermore, suppose $B_1 \geq \Omega(|\zeta_{k-1}|^2 k^2 \log(kd/\epsilon))$ s.t. the weights satisfy the conditions in Lemma B.2 with $\tau_g = \mathcal{O}(|\zeta_{k-1}| k^{-1})$ after the first phase. Then after $T_2 = \Omega(md^2 k^3/\epsilon^2)$ steps of training with batch size $B_2 = 1$, learning rate $\eta_2 = 4k^{1.5}/(d\sqrt{m(T_2 - 1)})$ and decay $\lambda_2 = 0$, we have with expectation over the randomness of the initialization and the sampling of the batches:*

$$\min_{t \in [T_2]} \mathbb{E}\left[ L_{\theta^{(t)}}(\mathbf{x}, y) \right] \leq \epsilon.$$

*Thus, the minimal sample complexity to reach a loss of $\epsilon$ is given by*

$$T_1 \times B_1 + T_2 \times B_2 = \Theta(|\zeta_{k-1}|^2 k^2 \log(kd/\epsilon)) + \Theta(md^2 k^3/\epsilon^2)$$
$$= \Theta(d^{k-1} k^2 \log(dk/\epsilon) + 2^k d^2 k^4 \epsilon^{-2} \log(k/\delta)).$$

**Corollary B.8.** *Under the conditions outlined in Lemma B.7, after $T_2$ steps of training in the second phase, if $t^\dagger$ denote the time step at which the model achieves the minimum loss, i.e. $t^\dagger := \arg\min_{t \in [T_2]} \mathbb{E}\left[ L_{\theta^{(t)}}(\mathbf{x}, y) \right]$, then*

$$\mathbb{E}\left[ f_{\mathcal{T}}^{(t^\dagger)}(\mathbf{x}) x_i \right] \leq \epsilon, \text{ for all } i \in [d].$$

The proof follows from the fact that if the correlation along $y = \prod_{i \in \mathbf{S}} x_i$ is large ($\geq 1 - \epsilon$ as hinge loss is below $\epsilon$), the correlations along other Fourier basis functions will be small. Hence, depending on how saturated the model is, the signal along the support elements are small.

We will use a slightly modified version of Lemma B.7 with higher sample complexity in the first phase, to ensure the stronger conditions of Corollary B.6 hold true as well. This will be necessary to get improved signal to teach a smaller student.[12]

---

[12] We haven't optimized the error bounds in Corollary B.6. Our sample complexity bounds are likely loose in Corollary B.9

**Corollary B.9** (Modified Version of Lemma B.7). *Fix $\epsilon, \delta > 0$. Suppose $m \geq \Omega(2^k k \log(k/\delta))$, $d \geq \Omega\left(k^4 \log(kd/\epsilon)\right)$. Furthermore, suppose $B_1 \geq \Omega(|\zeta_{k-1}|^2 k^2 d^{10/3} \log(kd/\epsilon))$ s.t. the weights satisfy the conditions in Corollary B.6 with $\tau_g = \mathcal{O}(|\zeta_{k-1}| k^{-1} d^{-5/3})$ after the first phase. Then after $T_2 = \Omega(md^2 k^3/\epsilon^2)$ steps of training with batch size $B_2 = 1$, learning rate $\eta_2 = 4k^{1.5}/(d\sqrt{m(T_2 - 1)})$ and decay $\lambda_2 = 0$, we have with expectation over the randomness of the initialization and the sampling of the batches:*

$$\min_{t \in [T_2]} \mathbb{E}\left[L_{\theta^{(t)}}(\mathbf{x}, y)\right] \leq \epsilon.$$

*Thus, the minimal sample complexity to reach a loss of $\epsilon$ is given by*

$$
\begin{aligned}
T_1 \times B_1 + T_2 \times B_2 &= \Theta(|\zeta_{k-1}|^2 d^{10/3} k^2 \log(kd/\epsilon)) + \Theta(md^2 k^3/\epsilon^2) \\
&= \Theta(d^{k+7/3} k^2 \log(dk/\epsilon) + 2^k d^2 k^4 \epsilon^{-2} \log(k/\delta)).
\end{aligned}
$$

### B.2 ANALYSIS FOR THE STUDENT

*Proof of Theorem B.1.* We will first prove the sample complexity upper bound for progressive distillation, followed by a sample complexity lower bound for distillation.

**Sample complexity for Progressive distillation:** Under progressive distillation, the label is given by $f_{\mathcal{T}}^{(T_1)}$ for the first $T_1$ steps. We will follow similar steps as Lemma B.2, where the label is replaced by $f_{\mathcal{T}}^{(T_1)}$. Claim B.3 changes, while Claim B.4 stays the same. We will showcase the change in Claim B.3 here.

At initialization, the population gradient of the weight vector in neuron $i$ at coordinate $j$ is given by

$$
\begin{aligned}
&\mathbb{E}_{\mathbf{x},y} \nabla_{\tilde{w}_{ij}^{(0)}} \ell_{\mathrm{DL}}(\mathbf{x}, y; f_{\mathcal{S}}^{(0)}, f_{\mathcal{T}}) \\
&= -\mathbb{E}_{\mathbf{x},y} \nabla_{\tilde{w}_{ij}^{(0)}} f_{\mathcal{S}}^{(0)}(\mathbf{x}) f_{\mathcal{T}}^{(T_1)}(\mathbf{x}) \\
&= -a_i \mathbb{E}_{\mathbf{x},y} \mathbb{I}\left[\langle \tilde{\boldsymbol{w}}_i^{(0)}, \mathbf{x}\rangle + \tilde{b}_i \geq 0\right] f_{\mathcal{T}}^{(T_1)}(\mathbf{x}) x_j \\
&= -a_i \mathbb{E}_{\mathbf{x},y} \left(\frac{1}{2} + \frac{1}{2}\mathrm{Maj}(\tilde{\boldsymbol{w}}_i^{(0)}, \mathbf{x})\right) f_{\mathcal{T}}^{(T_1)}(\mathbf{x}) x_j \\
&= -a_i \frac{1}{2} \mathbb{E}_{\mathbf{x},y} f_{\mathcal{T}}^{(T_1)}(\mathbf{x}) x_j - a_i \frac{1}{2} \mathbb{E}_{\mathbf{x},y} \mathrm{Maj}(\tilde{\boldsymbol{w}}_i^{(0)}, \mathbf{x}) f_{\mathcal{T}}^{(T_1)}(\mathbf{x}) x_j,
\end{aligned}
$$

where the relation between $\mathbb{I}\left[\langle \boldsymbol{w}_i^{(0)}, \mathbf{x}\rangle + \tilde{b}_i \geq 0\right]$ and $\mathrm{Maj}(\tilde{\boldsymbol{w}}_i^{(0)}, \mathbf{x})$ follows because of $\left|\tilde{b}_i\right| < 1$ at initialization. From Corollary B.6,

$$
\left|\mathbb{E}_{\mathbf{x},y} \nabla_{\tilde{w}_{ij}^{(0)}} \ell_{\mathrm{DL}}(\mathbf{x}, y; f_{\mathcal{S}}^{(0)}, f_{\mathcal{T}})\right| \geq \Omega(k^{-1}), \quad \text{if } j \in [k],
$$

$$
\left|\mathbb{E}_{\mathbf{x},y} \nabla_{\tilde{w}_{ij}^{(0)}} \ell_{\mathrm{DL}}(\mathbf{x}, y; f_{\mathcal{S}}^{(0)}, f_{\mathcal{T}})\right| \leq o(k^{-1}), \quad \text{if } j \notin [k].
$$

Thus, a fourier gap exists between the population gradients on in-support and out-of-support coordinates in the gradients. We can then apply Claim B.4 to show that a finite batch size of $B_1 \geq \Omega(k^2 \log(d\tilde{m}/\delta))$ is sufficient to maintain this gap between the coordinates in support and out of support. Thus, the change in the necessary sample complexity comes from the reduced sample complexity in the first phase. The proof for the second phase training is exactly equal to the proof for the teacher in Theorem B.1.

**Sample complexity for Distillation:** On the other hand, for the teacher checkpoint with loss $\mathcal{O}(d^{-c})$, the correlation to the monomial terms in the support is bounded by $\mathcal{O}(d^{(-c)})$ (by Corollary B.8). If we want to learn from the correlations to the support, we need the number of samples to be at least $\Omega(d^{2c})$ as the gradient noise needs to be lower than $\mathcal{O}(d^{-c})$ (by Claim B.4). To learn the support from the true label, we need the number of samples to be at least $\Omega(d^{k-1})$, by the following result:

**Lemma B.10** (Width-optimization trade-off, cf. Proposition 3 in (Edelman et al., 2023)). *For $\delta > 0$, gradient noise $\tau_g > 0$, and model width $m > 0$, if $T \leq \frac{1}{2}\binom{d}{k}\frac{\delta \tau_g^2}{m}$, then there exists a $(d, k)$-sparse parity such that w.p. at least $1 - \delta$ over the randomness of initialization and samples, the loss is lower bounded as $L(f_{\mathcal{T}}^{(t)}) \geq 1 - \tau_g$ for all $t \in \{1 \cdots T\}$.*

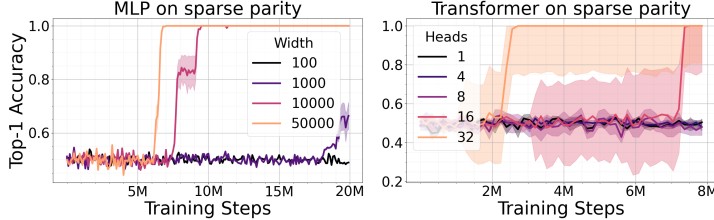

Figure 7: **Larger models learn sparse parity faster.** A larger model has more width (MLP, *left*) or more attention heads (Transformers, *right*). The results are for $(100, 6)$-parity, aggregated over 5 runs for each setup.

This result implies that for a fixed batch size (and hence a fixed $\tau_g$), we either require a bigger width, or more number of gradient steps (which translates to sample complexity since we are using fresh samples each batch). Hence, for the model to learn the support from a combination of the two components, it needs a sample complexity at least $\Omega(d^{\min(2c, k-1)}/\tilde{m})$. □

## C   RESULTS ON SPARSE PARITY AND ITS GENERALIZATION

### C.1   ADDITIONAL RESULTS ON SPARSE PARITY WITH MLP

We take both the teacher and student models to be 1-hidden-layer MLPs with ReLU activations. The teacher has a hidden width of $5 \times 10^4$, and the students are of widths $10^2$ or $10^3$. All models are trained using SGD with batch size 1 for $20M$ steps on sparse parity data with $n = 100$ and $k = 6$ (Definition 3.1). The support is set to be the first 6 coordinates of the input vector without loss of generality. The learning rate is searched over $\{10^{-2}, 5 \times 10^{-3}, 10^{-3}\}$. Evaluation is based on a held-out set consisting of 4096 examples, and we report the average across 3 different training seeds. For one-shot distillation, we use the teacher checkpoint at the end of training (20M checkpoint), at which point the teacher has fully saturated. For progressive distillation, we use $N = 200$ equally spaced teacher checkpoints that are 0.1M steps apart.

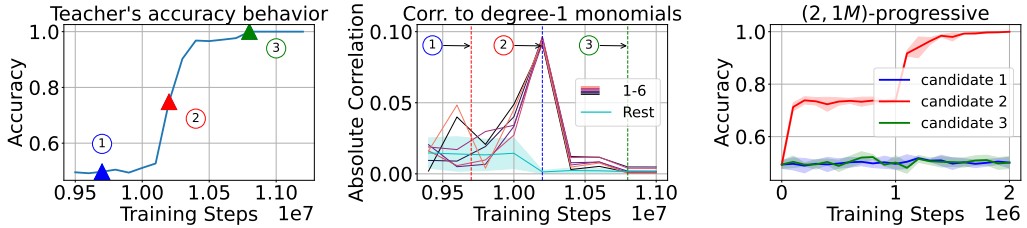

Figure 8: Repeated experiments from Figure 2 for a student of width 1000.

### C.2   LEARNING WITH TRANSFORMERS: PARALLEL SEARCH WITH ATTENTION HEADS

The benefit of progressive distillation and the implicit curriculum is not specific to MLP. This section presents similar results with Transformers (Vaswani et al., 2017). The $d$-dimensional input vector is now treated as a length-$d$ sequence, and the label is predicted using the last token's output. We fix the support $\mathbf{S}$ to be the first 6 coordinates of the sequence. Note that unlike MLP, Transformer's learning is not permutation-invariant to the location of $\mathbf{S}$ due to the causal mask. Nevertheless, given the same $\mathbf{S}$, the comparison on learning speed is still meaningful.

For Transformers, the parallel queries come from both the MLP width and also the *number of attention heads*. To illustrate this, consider the following two solutions (which we formalize in Appendix C.2.1) to sparse parity: The first solution uses attention to locate the support and then uses MLP to compute the product of the in-support variables. The second solution copies over all variables to the final position, whose MLP is then responsible for both identifying the support and computing the product.

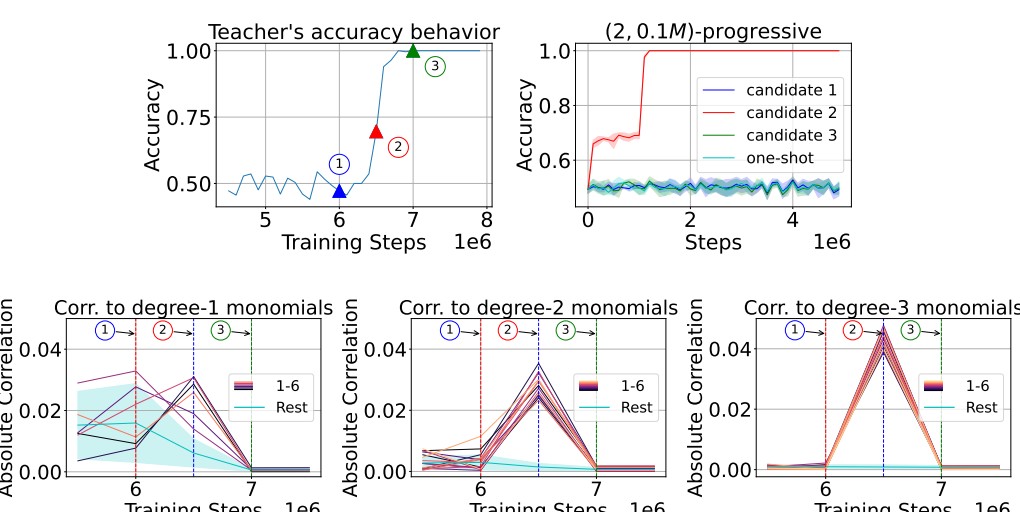

Figure 9: Repeated experiments from Figure 2 but for a different teacher. For $(2, 0.1M)$ progressive distillation, the checkpoint that lies in the middle of the second phase accelerates training the most. In Figure 2, we used a teacher that was trained with learning rate $5 \times 10^{-3}$. The correlation plot for the teacher to degree-1 monomials had a clear gap for degree-1 monomials in-support and out-of-support at the middle of the second phase (indicated by candidate 2). However, for a teacher that is trained with a higher learning rate $10^{-2}$, we didn't find such a clean gap in correlations for degree-1 monomials. On the other hand, correlations to degree-2 and degree-3 monomials showed a clean gap between in-support and off-support variables at the middle of the phase transition. Hence, the student needn't learn only from degree-1 monomials to get training acceleration, any low degree monomials suffice to teach the student about the support. Rest for degree-2 monomials refers to all monomials of the form $x_i x_j$ where atleast one of $i, j \notin \mathbf{S}$. Similar definition for degree-3 monomials.

The second solution is less interesting as it reduces to an MLP, so we focus on the first solution in the following, which utilizes the attention mechanism unique to Transformers.

**(R6) More attention heads helps with the search for support** Our experiments are based on 2-layer Transformers [13] with 8 dimensions per attention head. As shown in Figure 7 (right), increasing the number of heads makes learning faster. There are clear phase transitions similar to the MLP case.

**Ablation with other ways to vary the model size** Most Transformer experiments in this work keep the per-head dimension to be fixed and vary the number of attention heads between the teacher and the student. The MLP input dimension is the sum of the attention head dimensions, so a student with fewer heads will have a smaller MLP than the teacher, which is preferable in terms of efficiency. Fixing the per-head dimension is a widely adopted setup in practice, such as in the Llama series (Touvron et al., 2023). We now additionally consider two other ways to vary the model size. In particular, we vary the number of heads, while 1) fixing the hidden dimension (i.e. the total dimension of all heads concatenated) to be 256, or 2) fixing the dimension of each head to be 256 and averaging the output from each head, in which case the hidden dimension is also 256. These two setups are less common in practice but nevertheless serves as complementary evidence: the performance difference comes solely from the number of attention heads, as the MLP dimension is kept the same. As shown in Figure 10 (b,c), increasing the number of attention also increases the training speed in these two setups.

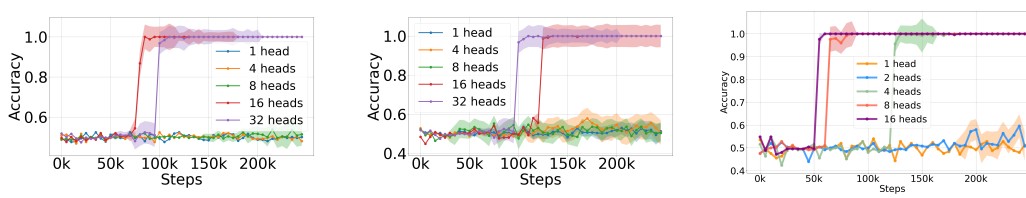

(a) Per-head dimension = 8          (b) Hidden dimension = 256          (c) Dimension 256, averaging heads

Figure 10: Increasing the number of attention heads speeds up training. Each plot compares the accuracy throughout training for 2-layer models with various heads, while fixing: (a) the per-head dimension to 8; (b) the MLP hidden dimension to 256; (c) both the per-head and MLP hidden dimension to 256, by averaging (rather than concatenating) the heads. We report runs with the learning rate that has the highest mean accuracy and break tie with training speeds. The shadows show the variances of the runs.

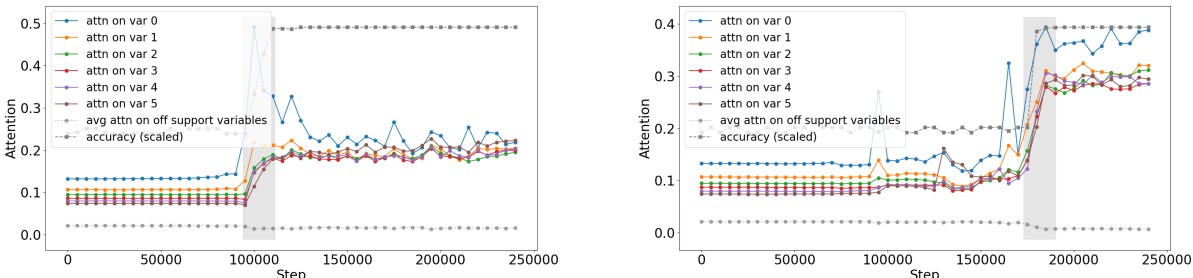

Figure 11: **In-support attention growth co-occurs with accuracy increase** Attention on individual coordinates on or off the support of the sparse parity, taking the median of 1024 random binary input sequences. The shade highlights the teacher's phase transition period. The model accuracy is marked by the gray dashed line, with scale adjusted for better display. The two subfigures show the same type of results but with different randomness seeds.

**Ablation with 2-shot distillation** We repeat the 2-shot distillation ablation for MLP. We first confirm that the low-degree curriculum described in Section 3.1 is also observed in Transformers. As

---

[13]We use 2 layers since 1-layer Transformers are hard to train empirically, despite being representationally sufficient to solve sparse parity.

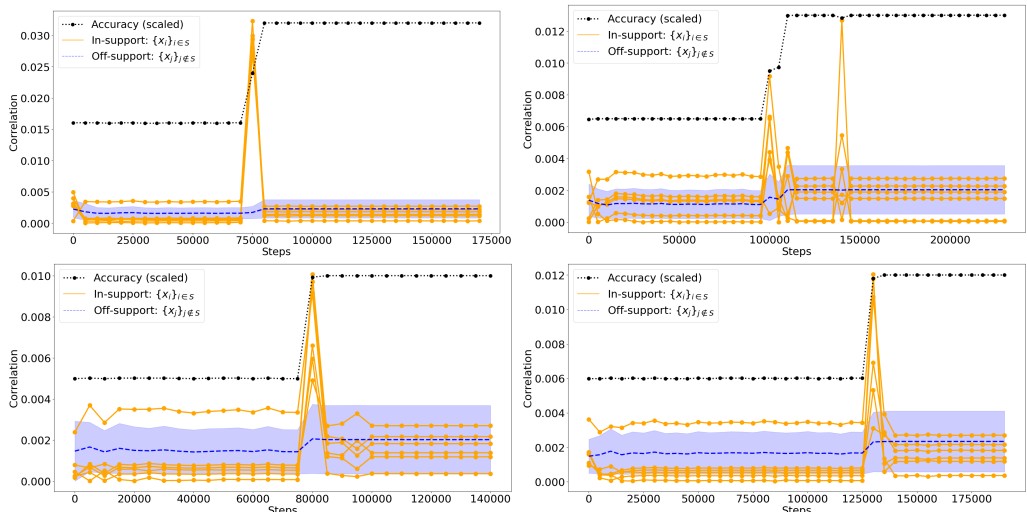

Figure 12: **Low-degree curriculum in Transformers on (100, 6)-sparse parity.** The $x$-axis shows the training steps, and $y$-axis shows the 2-layer 32-head teacher's correlation with in-support (orange lines) vs off-support (blue lines, aggregated into mean and standard deviation) degree-1 monomials. The black dotted lines mark the accuracy, scaled for better display. The correlation values are calculated using 100k randomly drawn sequences. The 4 subplots correspond to models trained using 4 random seeds.

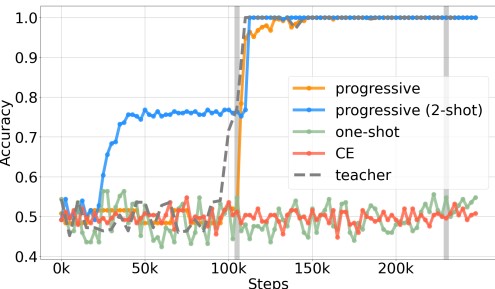

Figure 13: **2-shot progressive distillation with Transformers**: Compared to cross-entropy training or one-shot distillation, transformers learn faster with progressive distillation, where the intermediate checkpoints are taken either at regular 10k intervals ("progressive"), or during the phase transition ("progressive (2-shot)"). The two vertical lines show the teacher training steps at which the two checkpoints for 2-shot distillation are chosen. We set the teacher temperature to be $\tau = 10^{-4}$ for progressive distillation, and $\tau = 1$ for one-shot distillation.

shown in Figure 12, the 2-layer 32-head teacher model exhibits significantly higher correlation with the in-support monomials (i.e. $\{x_i\}_{i \in \mathbf{S}}$ than with off-support monomials during the phase transition. [14] Then, we show in Figure 13 that using as few as 1 intermediate checkpoint suffices to significantly speeds up the training of the student.

**Ablation with various temperatures** As mentioned in Section 2, our progressive distillation results use a low temperature in order to remove potential favorable regularization effects from soft labels. We chose a temperature of $\tau = 10^{-4}$ for sparse parity, where the output dimension is 2. In Figure 14, we empirically confirm that setting the temperature to be below 0.01 is sufficient to get results that are qualitatively similar to using $\tau = 0$ (i.e. taking the argmax). Note that using a higher temperature such as $\tau = 1$ can make learning slower despite potentially having more regularization effects from softer labels. We leave understanding the exact effect of temperature to future work.

---

[14]Note that the upper right subplot in Figure 12 has a second correlation spike with the in-support variables. However, supervising with this second checkpoint does not provide acceleration. This suggests that there might be mechanisms other than the low-degree curriculum at play.

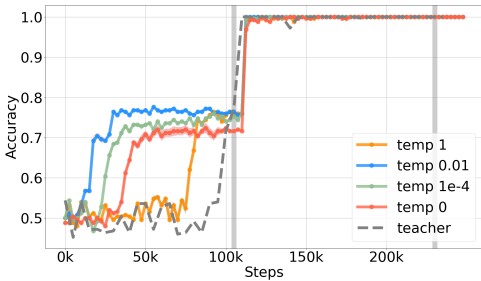

Figure 14: **The benefit of progressive distillation holds with hard labels**, as shown by comparing 2-shot progressive distillation with different temperatures. The two gray vertical lines mark the training steps at which the teacher checkpoints are taken.

### C.2.1 TWO TRANSFORMER SOLUTIONS FOR SPARSE PARITY (PROPOSITION C.1 AND PROPOSITION C.3)

We consider a simplified version of a Transformer block, without the residual connection or the layernorm:

$$f_{\text{block}} = f_{\text{mlp}}^{(L)} \circ f_{\text{attn}},$$

where

$$f_{\text{attn}}(X; W_Q, W_K, W_V) := \text{CausalAttn}(XW_Q W_K^\top X^\top)XW_V,$$

with $W_Q, W_K, W_V$ being the query, key, value matrices, and $f_{\text{mlp}}^{(L)}(x; \{W_l, b_l\}_{l \in [L]})$ is a $L$-layer MLP that recursively apply $f_{\text{mlp}}^{(l+1)}(x) = \sigma(W_{l+1} f_{\text{mlp}}^{(l)}(x) + b_{l+1})$ position-wise. $\sigma$ is the relu function for $l \in [L-1]$, and is the identity function for $l = L$.

**Proposition C.1** (Attention support selection). *$(d, k)$-sparse parity can be solved by a 1-layer Transformer with a 2-layer MLP, whose attention weights satisfy $\alpha_{i,d} \propto \exp(c\mathbb{1}[i \in \mathbf{S}])$ for some large constant $c > 0$. The MLP has hidden dimension $4(k+1)$, $L_\infty$ norm bounded by $4k(k+1)$.*

*Proof.* The idea is that the attention selects the $k$ in-support variables, and the MLP computes the product of these variables.

To select the in-support variables, we want the attention weight $\alpha_{i,d} \propto \exp(c\mathbb{1}[i \in \mathbf{S}])$, for some large constant $c$. This can be achieved by having the projection matrices $W_Q, W_K$ focus only on the position and ignore the tokens. In particular, let $z$ denote the input sequence, and let the embedding of a token be $\boldsymbol{x}_i = \boldsymbol{v}_{z_i} + p_i$, where $\{\boldsymbol{v}_0, \boldsymbol{v}_1\}$ are embeddings for the binary token 0 or 1, and $p_i$ is the position encoding for position $i$. Take $\{\boldsymbol{v}_0, \boldsymbol{v}_1\}, \{p_i\}_i$ such that $\boldsymbol{v}_{z_i} \perp p_i$. Let $c > 0$ be a large enough constant. Choose $W_Q, W_K$ such that for any $i \in [d]$, $\boldsymbol{x}_i^\top W_Q^\top W_K \boldsymbol{x}_d = p_i^\top W_Q^\top W_K p_d = c \cdot \mathbb{1}[i \in \mathbf{S}]$. This ensures that $\alpha_{i,d} \propto \exp(c\mathbb{1}[i \in \mathbf{S}])$.

Then, the role of attention is to average over the in-support tokens. For simplicity of exposition, let's take $c \to \infty$ for now (i.e. using saturated attention (Merrill et al., 2022)), so that $\alpha_{i,n} \to \frac{\mathbb{1}[i \in \mathbf{S}]}{k}$; that is, the attention weights at the last position average over the in-support variables. Take $W_V$ to be a vector, such that $W_V$ ignores the positional information and the input token 0, and only preserves the input token 1, i.e. $W_V p_i = 0, \forall i \in [d]$, $W_V \boldsymbol{v}_0 = 0$, and $W_V \boldsymbol{v}_1 = 1$.

Next, the MLP needs to compute the parity function over the $k$ in-support variables. The input to the MLP is hence proportional to $(\sum_{i \in \mathbf{S}} \mathbb{I}[z_i = 1])\boldsymbol{v}_1$, and the size of the set of inputs is $k + 1$. To determine the size of the MLP, we use the following lemma:

**Lemma C.2** (1D discrete function interpolation with an MLP (Lemma 1 in Liu et al. (2022))). *Let $\mathcal{X}$ be a finite subset of $\mathbb{R}$, such that $|x| \leq B_x$ for all $x \in \mathcal{X}$, and $|x - x'| \geq \Delta$ for all $x \neq x' \in \mathcal{X}$. Let $f : \mathcal{X} \to \mathbb{R}^d$ be such that $\|f(x)\|_\infty \leq B_y$ for all $x \in \mathcal{X}$. Then, there is a 2-layer ReLU network for which*

$$f_{\text{mlp}}(x + \xi; \theta_{\text{mlp}}) = f(x) \qquad \forall x \in \mathcal{X}, \quad |\xi| \leq \Delta/4.$$

*The inner dimension is $d' = 4|\mathcal{X}|$, and the weights satisfy*

$$\|W_1\|_\infty \le \frac{4}{\Delta}, \quad \|b_1\|_\infty \le \frac{4B_x}{\Delta} + 2, \quad \|W_2\|_\infty \le B_y, \quad b_2 = 0.$$

Setting $B_x = 1$, $B_y = 1$, and $\Delta = \frac{1}{|\mathbf{S}|}$, the parity function over these $k + 1$ input values can be approximated by a 2-layer MLP with inner dimension $4(k + 1)$, with norm bounded by $4k(k + 1)$.

As a concrete example, one way to satisfy the requirements above is to set the attention weights to $\boldsymbol{v}_1 = W_V = \boldsymbol{e}_1 := [1, 0, 0, 0]$, $\boldsymbol{v}_0 = \boldsymbol{e}_2 := [0, 1, 0, 0]$. Set $p_i = p_n = \boldsymbol{e}_3$ for $i \in \mathbf{S}$, and $p_i = \boldsymbol{e}_4$ for $i \notin \mathbf{S}$. Set $W_Q = W_K = c \begin{bmatrix} 0 & 0 & 1 & 0 \\ 0 & 0 & 0 & 0 \end{bmatrix}$ for some sufficiently large $c > 0$.

$\square$

**Proposition C.3** (No attention selection). *There exists a 1-layer Transformer with 3-layer MLP that computes $k$-sparse parity, whose attention weights satisfy $\alpha_{i,d} = \frac{1}{d}$. Consequently, the MLP computes the sparse parity function given the full set of variables.*

*Proof.* The idea is for the uniform attention to copy all tokens to the last position. However, unlike in Proposition C.1, the attention needs to copy the tokens into a length-$d$ embedding vector, as we need to preserve the position information in this embedding vector. We need to generalize Lemma C.2 accordingly to handle multi-dimensional inputs:

**Lemma C.4** (General discrete function interpolation with an MLP; (Lemma 2 in Liu et al. (2022))). *Let $\mathcal{X}$ be a finite subset of $\mathbb{R}^{d_{\mathrm{in}}}$, such that $\|x\|_\infty \le B_x$ for all $x \in \mathcal{X}$, and $\|x - x'\|_\infty \ge \Delta$ for all $x \ne x' \in \mathcal{X}$. Let $f : \mathcal{X} \to \mathbb{R}^{d_{\mathrm{out}}}$ be such that $\|f(x)\|_\infty \le B_y$ for all $x \in \mathcal{X}$. Then, there is a 3-layer ReLU network for which*

$$f_{\mathrm{mlp}}(x + \xi; \theta_{\mathrm{mlp}}) = f(x) \qquad \forall x \in \mathcal{X}, \quad |\xi| \le \Delta/4.$$

*Letting $\mathcal{X}_i$ denote the set of unique values in coordinate $i$, the inner MLP dimensions are as follows:*

$$d_1 = 4 \sum_{i \in [d_{\mathrm{in}}]} |\mathcal{X}_i|, \quad d_2 = |\mathcal{X}|.$$

*The weights satisfy*

$$\|W_1\|_\infty \le \frac{4}{\Delta}, \quad \|b_1\|_\infty \le \frac{4B_x}{\Delta} + 2, \quad \|W_2\|_\infty \le 1, \quad \|b_2\|_\infty \le d_{\mathrm{in}}, \quad \|W_3\|_\infty \le B_y, \quad b_3 = 0.$$

Then, the MLP at the last position computes the sparse parity over the $k$ coordinates while ignoring the others. Hence the effective input set is $|\mathcal{X}| = 2^k$. Setting $B_x = 1$, $B_y = 1$, and $\Delta = 1$, there exists a 3-layer MLP with width $2^k$ and norm bound $2^{k+2}$ by Lemma C.4. $\square$

**Preliminary interpretability analysis: Transformer does utilize attention in practice** We observe that the model focuses attention on relevant tokens and that the amount of attention weights put on the support is tightly correlated with the accuracy, which suggests that the model indeed utilizes the attention mechanism in learning sparse parity.

Specifically, Figure 11 shows the results on 2-layer 16-head GPT-2 models. The attention weights are for the final position, whose logits are used for computing the binary parity label for the entire sequence. We track the attention weights along *length-2 paths* from the first and the second layer. For example, for a single-head model, let $\mathbf{a}_i^{(l)} \in \Delta^{d-1}$ denote the $l_{th}$-layer attention vector at the $i_{th}$ position; then, the on-support attention for a given sample is computed as $\langle \mathbf{a}_d^{(2)}, \mathbf{v}_{\mathcal{T}}^{(1)} \rangle$, where $[\mathbf{v}_{\mathcal{T}}^{(1)}]_i := \sum_{j \in \mathcal{T}} \mathbf{a}_i^{(1)}[j]$ is the total amount of first-layer attention weights that the $i_{th}$ position puts on the support $\mathcal{T}$. For multi-head models, $\mathbf{a}_i^{(l)} \in \Delta^{d-1}$ is defined as the sum of attention vectors from all heads, and the rest is computed similarly.

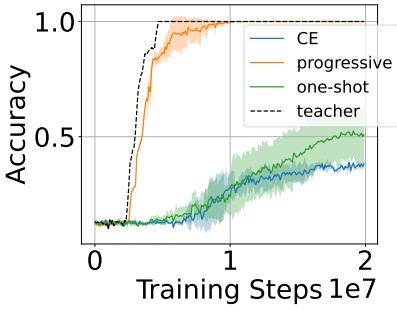 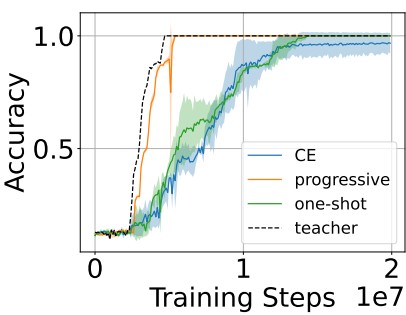

(a) Width-100 student.  (b) Width-1000 student.

Figure 15: 8-way classification using a hierarchical decision tree of depth 3, with each node represented by 5-sparse parity. Progressive distillation helps student learn faster from a width-50k teacher, compared to one-shot distillation from the final checkpoint.

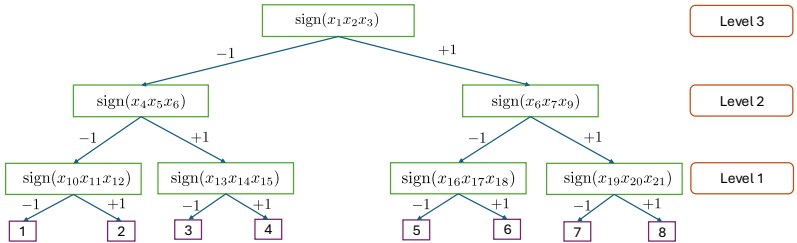

Figure 16: An illustration of hierarchical data generation, for a 3-level tree with 3 variables per feature. A feature corresponds to a tree node, each marked by a rectangle. The product of the binary variables in a feature determines which child to take: the left child is chosen if the product evaluates to $-1$, and the right child is chosen if the product is $+1$. The final label for an example is decided based on the tree leaf reached.

## C.3 A HIERARCHICAL GENERALIZATION OF SPARSE PARITY

This section considers an extension of sparse parity, where the labels are given by a decision tree. Sparse parity can be considered as a special case with tree depth 1.

**Definition:** The input $\mathbf{x}$ is a boolean vector picked uniformly at random from the $d$-dimensional hypercube $\{\pm 1\}^d$, and the label $y \in [K]$ where $K := 2^D$ for some fixed $D \in \mathbb{N}$. The underlying labeling function for $y$ follows a binary decision tree of depth $D$, whose leaves correspond to class labels. The branching at a node depends on a sparse parity problem. An example visualization is provided in Figure 16.

More formally, the nodes in the decision tree are represented by a set of sparse parity problems $\mathbb{S} = \{\mathcal{T}_1, \mathcal{T}_2, \cdots, \mathcal{T}_{K-1}\}$, where $\mathcal{T}_j$ is determined by product of a subset of size $k$ variables selected from the dimensions of the input $\mathbf{x}$ (e.g. $x_1 x_2 \cdots x_5$ for $k = 5$). An input $\mathbf{x}$ belongs to the class $i \in [K]$ iff

$$[\prod_{j=1}^{D} \mathbb{I}\left[c(i,j)\mathcal{T}_{v_j^{(i)}}(\mathbf{x}) > 0\right] > 0, \quad \text{where}$$

$$c(i,j) = \begin{cases} 1, & \text{if } i \geq 2^{D-j} \\ -1, & \text{otherwise} \end{cases}$$

Here, $v_1^{(i)}, \cdots v_D^{(i)}$ denote the features in $\mathbb{S}$ that lie on the path joining the root of the decision tree to the leaf representing the label $i$. An example is given in Figure 16.

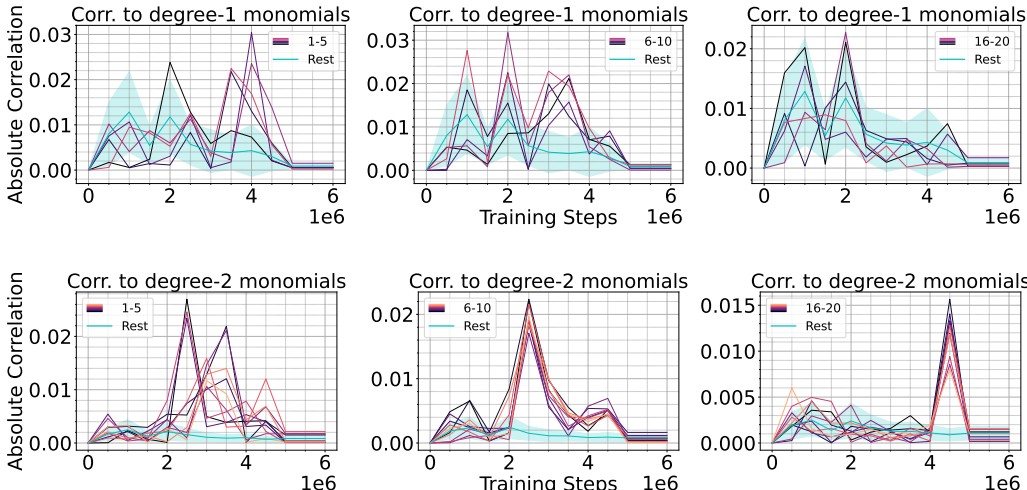

Figure 17: Setting: 8-way classification using a hierarchical decision tree of depth 3, with each node represented by 5-sparse parity. The relevant features for class $y = 1$ are $x_1 \cdots x_5, x_6 \cdots x_{10}, x_{16} \cdots x_{20}$ at tree levels $3, 2$, and $1$ respectively (Figure 16). The irrelevant features are $x_{36}, \cdots, x_{100}$. Here we plot the magnitude of correlation to degree-1 monomials $\mathbb{E}_{\mathbf{x},y}[p_{\mathcal{T}}(\mathbf{x})]_1 x_i$ for each $i$ in the relevant feature groups for class 0. Because the degree-1 monomials show noisy correlations, we also report the magnitude of correlation to degree-2 monomials $\mathbb{E}_{\mathbf{x},y}[p_{\mathcal{T}}(\mathbf{x})]_1 x_i x_j$ for each $i, j$ in the relevant feature groups for class 1. For degree-2 monomials, rest refers to correlation to monomials of the form $x_i x_j$ where atleast one variable is outside support variables ($x_36, \cdots, x_100$). The correlations to degree-1 (or 2) monomials on the relevant features spike at different training steps.

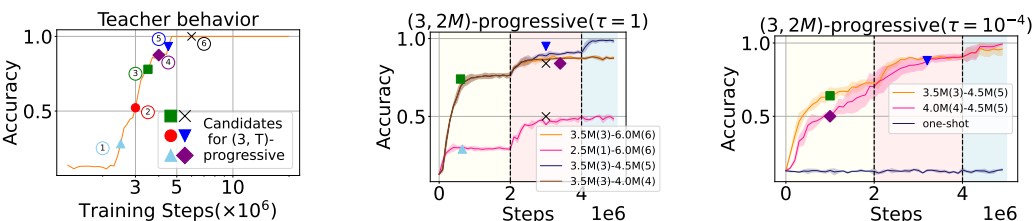

Figure 18: Setting: 8-way classification using a hierarchical decision tree of depth 3, with each node represented by 5-sparse parity. $(3, 2M)$-progressive distillation from 3 checkpoints on a 1000 width student; 2 intermediate teacher checkpoints are used each for $2M$ steps, and then the final checkpoint is used till end of training. **Observations:** (a) Teacher shows a phase transition in accuracy during training. 6 candidate checkpoints for $(3, 2M)$-progressive distillation have been marked, out of which 2 are selected in each setting. The checkpoint at $6M$ lies outside the phase transition of the teacher. (b): We show the behavior of a few representative settings. Two main observations: (1) Selecting only a single checkpoint during the phase transition of the teacher is sub-optimal, as shown by plots that contain $6M$ checkpoint as an intermediate checkpoint, (2) 2 checkpoints during the stage transition suffice to train the student to $100\%$ accuracy, however the performance can heavily depend on their selection. Figure 17 shows that the teacher learns the low-level features at $4.5M$ checkpoint, making it crucial for distillation. (c): Even with extremely low temperature, the benefit of the phase transition checkpoint persists, suggesting that the monomial curriculum, not regularization, is the key to the success of progressive distillation.

**Experiment Setup:** In this section, we focus on 8-way classification, where the data is generated by a tree of depth 3. Each feature in $\mathbb{S}$ is given by a product of 5 variables. We keep the variables distinct in each feature, i.e., $\mathcal{T}_1 = x_1 x_2 \cdots x_5, \mathcal{T}_2 = x_6 x_7 \cdots x_{10}$ and so on.

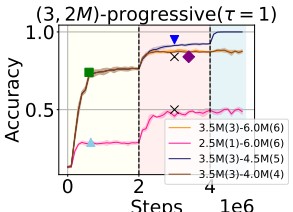 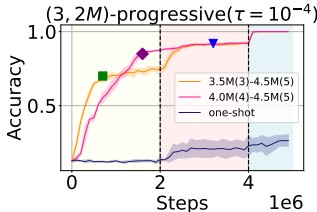

Figure 19: Same experiments as Figure 18 for a width-100 student.

**Experiments and Observations:** We conduct similar experiments as our sparse parity experiments. In Figure 15, we show that progressive distillation helps train a smaller student as fast as the teacher, and even reach 100% accuracy.

**Low-degree curriculum:** We show the correlations of the teacher's logits for a particular label and its relevant features in Figure 17. We observe similar spikes in the degree-1 monomials involving the support of the features. However, because there are multiple features defining a label class, with features at level 1 being shared among multiple labels, we see a difference in the time-frames at which the spikes appear in the degree-1 monomials of the features. As such, a single teacher checkpoint won't give information of entire support to a student to learn from.

**Effectiveness of $(3, T)$-progressive distillation:** We consider progressive distillation with 3 checkpoints, where the student only uses 2 intermediate teacher checkpoint in addition to the final one. We show in Figure 18 that there exists a $(3, 2M)$-progressive distillation that can help train a student successfully. Furthermore, we demonstrate that these two intermediate checkpoints must be positioned within the phase transition to achieve 100% accuracy in training the student. This supports the hypothesis that a low-degree curriculum is crucial for progressive distillation since the correlations with degree-1 monomials are high only during the phase transition period. Additionally, we find that a distillation strategy with only a single intermediate checkpoint and the final checkpoint is insufficient for the student to achieve 100% accuracy, which aligns with our observation that degree-1 monomials for different features emerge at different steps. However, we also note that even within the phase transition, the optimal selection of the two checkpoints can significantly impact the student model's performance.

## D  EXTENSIVE STUDY ON PCFGS

### D.1  A FORMAL DESCRIPTION OF PCFGS

We study progressive distillation using probabilistic context free grammar (PCFG). Compared to sparse parity and hierarchical data, PCFG is a more realistic proxy for natural languages and has been commonly used as a sandbox for mechanistically understanding the training of language models (Zhao et al., 2023; Allen-Zhu & Li, 2023b). A PCFG consists of a set of non-terminals (NTs) and grammar rules involving the non-terminals that specify the generation process of a sentence. For example, for the sentence *The cat ran away*, the grammatical structure dictates words *the*, *cat*, *ran*, *away* as `determinant`, `noun`, `verb`, and `adverb`. *ran* and *away* together represent a `verb phrase`, and *the*, *cat* together represent a `noun phrase` (see Figure 3). For a language model to generate grammatically correct sentences, it needs to learn the underlying grammatical rules.

A probabilistic context-free grammar (PCFG) is defined as a 4-tuple $\mathcal{G} = (\mathcal{N}, v, \mathcal{R}, \mathcal{P})$, where

- $\mathcal{N}$ is the set of non-terminals, which can be considered as internal nodes of a parse tree. There is a special non-terminal $S$, known as the start symbol.
- $[v]$ is the set of all possible words, corresponding to parse tree leaves.
- $\mathcal{R}$ denotes a set of rules. For all $A, B, C \in \mathcal{N}$, there is a rule $A \to BC$ in $\mathcal{R}$. Furthermore, there are rules $A \to w$ for all $A \in \mathcal{N}, w \in [v]$.
- $\mathcal{P}$ specifies the probability of each rule to be used in the generation process. For a rule $r \in \mathcal{R}$, if $\mathcal{P}[r] = 0$, then the rule is an invalid rule under the generation process. Furthermore, for each non-terminal $A \in \mathcal{N}$, on all rules $r \in \mathcal{R}$ of the form $A \to \cdot$, $\sum_{r \in \mathcal{R}: r = A \to \cdot} \mathcal{P}(r) = 1$. We denote $\mathcal{R}(A)$ as the set of all non-zero rules from $A$.

A concrete example of PCFGs is to model grammars of natural languages (Jurafsky, 2000). In this case, language tokens form the vocabulary of PCFG, while parts of speech such as nouns, verbs or noun phrases, verb phrases form the non-terminals. Rules like noun phrases being composed of a determinant and a noun form the core of such PCFG, while the probability of each rule is determined by their occurrences across sentences in the language.

**Data generation from PCFG**   Given a PCFG $\mathcal{G} = (\mathcal{N}, v, \mathcal{R}, \mathcal{P})$, a string is generated in a recursive fashion as follows: we start with $s_1 = \text{ROOT}$ at step 1, and maintain a string $s_t \in ([v] \cup \mathcal{N})^*$ at step $t$. At step $t$, if all characters in $s_t$ belong to $[v]$, the generation process terminates, and $s_t$ is the resulting string. Otherwise, for each character $A \in s_t$, if $A \in \mathcal{N}$, we sample a rule $r \in \mathcal{R}$ of the form $A \to \cdot$ with probability $\mathcal{P}(r)$ and replace $A$ by characters given by $r(A)$.

**Tracking $n$-grams**   As outlined in Section 4, we track the behavior of trained models by measuring the behavior of their output on the neighboring $n$-gram context. In the context of PCFGs and masked language modeling for BERT, Zhao et al. (2023) theoretically demonstrate that one of the optimal algorithms for predicting masked tokens is a dynamic programming algorithm based on the inside-outside algorithm (textbook reference: Jurafsky (2000)). This algorithm computes "inside probabilities" for spans of tokens of various lengths, representing pairwise token dependencies within those spans. For example, in the setting of Figure 3, the inside probability for the span "The cat" indicates the likelihood that these two tokens co-occur. The dynamic programming approach calculates these inside probabilities hierarchically, with smaller spans forming the basis for larger spans. The model's performance ultimately depends on how accurately it represents span probabilities across different lengths. For instance, if the token "cat" is masked in the sentence "The cat ran away", the success of the model depends on the representation of the likelihood of the spans "The cat", "cat ran", "The cat ran", and "The cat ran away". We denote the neighboring tokens in the $n$-gram window span of a token as its $n$-gram context.

### D.2  VARIANTS OF PROGRESSIVE DISTILLATION

**Comparisons at different lengths** We follow common practices for training self-attention models for both one-shot distillation and progressive distillation. We use Adam optimizer (Kingma & Ba, 2014), 512 batch size training (to imitate large batch training), and a cosine learning rate schedule

(Loshchilov & Hutter, 2016) which is generally used to train large language models. As cosine learning rate depends on the total training horizon, in order to show that progressive distillation converges faster than one-shot distillation, we compare the two algorithms by varying the number of training samples for the student. That is, we train the teacher model with $4 \times 10^6$ training samples (equal to 8000 steps), and compare the two algorithms for a student model at $\{1, 2, 4, 8\} \times 10^6$ training samples (equal to $\{2000, 4000, 8000, 16000\}$ steps).

**Progressive Distillation choices** Because we are considering comparisons at different training lengths for the student, we have to consider a more general version of progressive distillation introduced in Definition 2.2. In Definition 2.2, progressive distillation is defined by two parameters, (a) number of teacher checkpoints ($N$) for supervision, and (b) training steps per checkpoint. We define our selection criteria for the $N$ checkpoints later. However, after selecting the $N$ checkpoints, we have the following two variants of progressive distillation.

1. $N$-shot Equal-split distillation: Here, we simply split the entire student's training length into $N$ equal intervals, where the student is supervised by the *ith* teacher checkpoint in interval $i \in [N]$.
2. $N$-shot $\kappa T_0$-Equal-split distillation: Here $\kappa \in (0, 1]$, and $T_0$ refers to the total training length of the teacher. The idea is to decide the allocation on the basis of the training length of the teacher, instead of the training length for the student. We train the student under the supervision of each checkpoint for $\kappa T_0 / N$ training steps. Teacher checkpoints that fail to fit into the student's supervision schedule are ignored (corresponding to a large $\kappa$), and the final checkpoint is kept till the end of training if the student is trained for longer than $\kappa T_0$. We can view $\frac{1}{\kappa}$ as the amount of "speed up"; for instance, we recover one-shot distillation with $\kappa \to 0$. Our experiments (Appendix D.2) suggest that $\kappa = 1/2$ is a reasonable rule of thumb that can help the student learn faster than the teacher at any given training length.

In the main paper, in Figures 1, 5 and 6, we have reported performance on PCFG and Wikipedia for $N$-shot $T_0$-Equal-split distillation as progressive distillation. We conduct more ablation studies on $\kappa$ in Appendix D.2. We keep the exploration of optimal strategies of progressive distillation to future work.

*Selection criteria for $N$ teacher checkpoints:* While there are multiple ways in which one can pick the reference checkpoints to train the student model, we use a simple strategy which is sufficient to demonstrate the benefit of progressive distillation. Similar to our observation of transition phase for parity in Section 3, we search for transition phases in the loss behavior of the teacher and select the first teacher checkpoint roughly in the middle of the transition phase. The rest are picked at multiples of this initial checkpoint.

### D.3 DETAILS ON NON-TERMINAL PREDICTION WITH MULTI-HEAD LINEAR PROBING

Following Allen-Zhu & Li (2023b), we train a position-based linear attention on the model's embeddings to predict the non-terminals at each level of underlying PCFG. We consider a set of linear functions $f_r : \mathbb{R}^d \to \mathbb{R}^{|\mathcal{N}|}$, where $r \in [H]$ and $H$ is the number of "heads" in the linear attention model. If $e_1, \cdots, e_L$ denote the model's output embeddings for a sequence $x_1, \cdots, x_L$, then the prediction of the model at each index $i \in [L]$ is given by

$$G_i(x) = \sum_{r \in [H], k \in [L]} w_{r, i \to k} f_r(e_k),$$

$$w_{r, i \to k} = \frac{exp(\langle P_{i, r}, P_{k, r} \rangle)}{\sum_{k' \in [L]} exp(\langle P_{i, r}, P_{k', r} \rangle)},$$

for trainable parameters $P_{i, r} \in \mathbb{R}^d$. We train the parameters with logistic regression on 51200 examples and test on a validation set of 1024 examples.

### D.4 DETAILS ON THE SYNTHETIC PCFGS

We use 5 synthetic PCFGs considered by Allen-Zhu & Li (2023a) (please see Figure 20 for the rules involved in the PCFGs). These 5 PCFGs differ in difficulty, based on the number of rules

per non-terminal and the ambiguities in the rules per non-terminal. Under a PCFG, each string is generated by generation trees of depth 7. We give differences in the PCFGs, as outlined by Allen-Zhu & Li (2023a) below.

- In cfg3b, the PCFG is constructed such that the degree $|\mathcal{R}(A)| = 2$ for every non-terminal $A$. In any generation rule, consecutive pairs of symbols on the generated symbols are distinct. The $25\%, 50\%, 75\%,$ and $95\%$ percentile string lengths generated by the PCFG are $251, 278, 308, 342$ respectively.

- In cfg3i, $|\mathcal{R}(A)| = 2$ for every non-terminal $A$. However, the consecutive pairs of symbols needn't be distinct in generation rules. he $25\%, 50\%, 75\%,$ and $95\%$ percentile string lengths generated by the PCFG are $276, 307, 340, 386$ respectively.

- In cfg3h, $|\mathcal{R}(A)| \in \{2, 3\}$ for every non-terminal $A$. he $25\%, 50\%, 75\%, 95\%$ percentile string lengths generated by the PCFG are $202, 238, 270, 300$ respectively.

- In cfg3g, $|\mathcal{R}(A)| = 3$ for every non-terminal $A$. he $25\%, 50\%, 75\%, 95\%$ percentile string lengths generated by the PCFG are $212, 258, 294, 341$ respectively.

- In cfg3f, $|\mathcal{R}(A)| \in \{3, 4\}$ for every non-terminal $A$. he $25\%, 50\%, 75\%, 95\%$ percentile string lengths generated by the PCFG are $191, 247, 302, 364$ respectively.

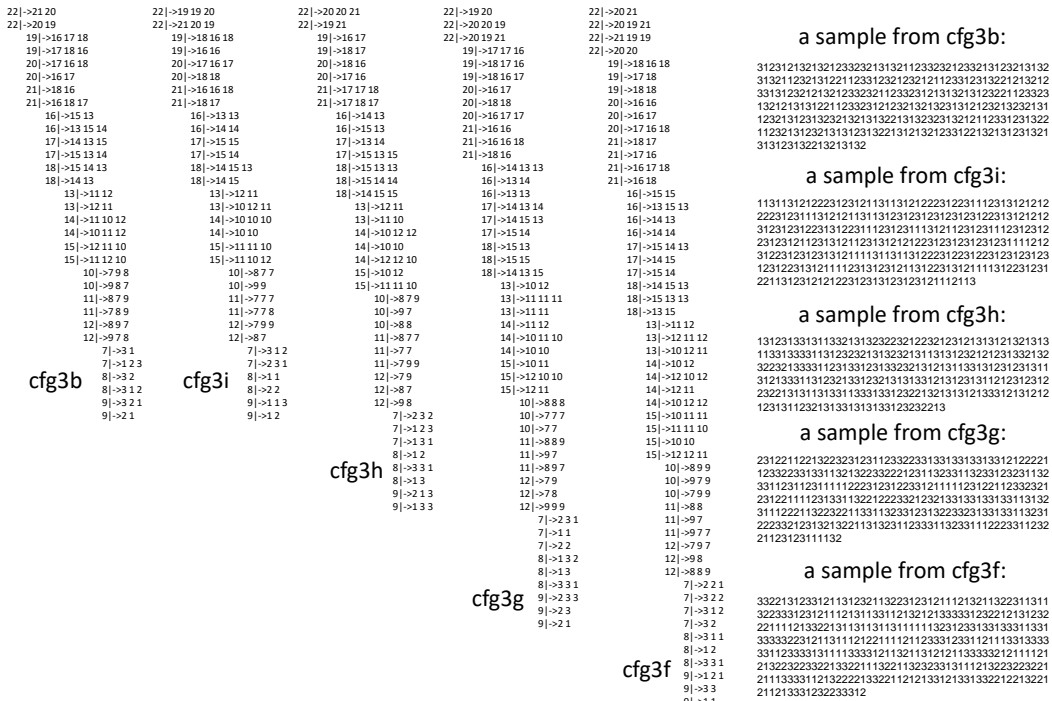

Figure 20: The synthetic PCFGs considered from Allen-Zhu & Li (2023b). Vocabulary is $\{1, 2, 3\}$ in each setting. More details on the differences between the PCFGs are in Appendix D.4.

## D.5 EXTENSIVE EXPERIMENTS ON BERT

We first give some details on the architecture of BERT and its pre-training loss function.

### D.5.1 A PRIMER ON BERT

BERT (Devlin et al., 2018) is an encoder-only transformer that is trained with masked language modeling (MLM) (Figure 21). In encoder-only architecture, the contextual information are shared across the tokens using bidirectional self-attention layers. During pre-training, the model is trained with MLM loss, that perturbs certain fraction of the tokens in the input at random and the model is

trained to predict the original tokens at positions of the perturbed tokens. The pre-training recipe follows a 80-10-10 principle, where tokens at $80\%$ of the perturbed positions are replaced by a special $\langle mask \rangle$ token, while tokens at $10\%$ of the perturbed positions are replaced by random tokens from the vocabulary, while remaining positions are filled with the original tokens themselves. We stick to this principle, while creating data for training from different PCFGs.

**Model architecture considered:** We train depth-4 BERT models with $\{8, 16, 32\}$ attention heads, each of which operates on 8 dimensions, using a $30\%$ masking rate. The head dimension is fixed to 8, with the corresponding width of the 4 models being $\{64, 128, 256\}$ respectively.

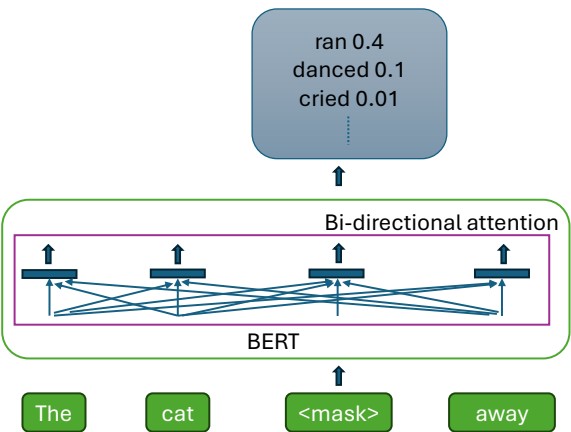

Figure 21: An informal representation of BERT (Devlin et al., 2018). The model uses bidirectional attention layers to share contextual information across the tokens. During pre-training, few of input tokens are replaced by special $< mask >$ tokens, and the model is trained to predict the masked tokens.

### D.5.2 DATA GENERATIONS

**Data for masked language modeling:** We generate $8 \times 10^6$ random sequences for each PCFG. We follow Devlin et al. (2018) to create masked input sequences and output labels, i.e. for each sampled sequence we mask $p\%$ of tokens for input and the labels are given by the tokens in the masked positions of the original sequence. We also follow the 80-10-10 principle, where for input, the tokens in $80\%$ of the masked positions are represented by a special mask token $[mask]$, while $10\%$ of the masked positions are represented by a randomly sampled token from the vocabulary and the remaining $10\%$ are represented by tokens from the original sequence.

### D.6 HYPERPARAMETER DETAILS

We use a batch size of 512 in each setting. We use Adam (Kingma & Ba, 2014) optimizer with 0 weight decay, $\beta_1, \beta_2 = (0.9, 0.95)$. We use cosine decay learning rate. We extensively tune the learning rate in the grid $\{10^{-2}, 7.5 \times 10^{-3}, 5 \times 10^{-3}, 2.5 \times 10^{-3}, 10^{-3}\}$ in each setting. We train the teacher on $4 \times 10^6$ training samples (equal to $8 \times 10^3$ steps).

**Distillation experiments at different training horizons:** To thoroughly compare the sample complexity requirements of one-shot and progressive distillation, we evaluate both algorithms using a smaller student model across various training sample sizes. The smaller student is trained with $\{1, 2, 4, 8\} \times 10^6$ training samples (equal to $\{2 \times 10^3, 4 \times 10^3, 8 \times 10^3, 16 \times 10^3\}$ training steps) and the performance is compared in each horizon. For example, Figure 1 (right) plot contains 4 distinct points for each method which represents the performance of the smaller model under the 4 different training steps (sample sizes).

**Training split for $(2, T)$-progressive distillation for PCFGs:** We report the performance in Figure 4 for 4000 training steps. We find the best training time split $T$ between the intermediate checkpoint and the final checkpoint in the grid $\{500, 1000, 15000, 2000\}$, i.e. the student is trained

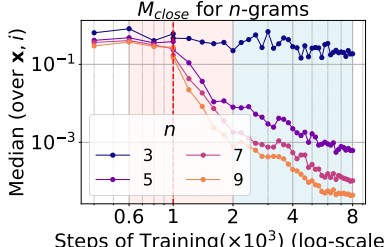 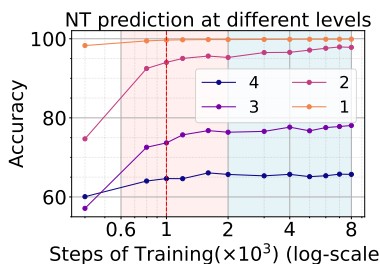

Figure 22: We conduct additional probing experiments on the teacher's ($4$ layer, $32$ attention head BERT) logits during training to indicate curriculum learning. (left) TV distance between model's predictions with full context and context with only $n$-gram tokens ($M_{\text{close}}$). We observe that the teacher's logits get closer to higher $n$-gram context predictions, and the inflection appears at the middle of the second phase (our first selected checkpoint for progressive distillation) (right) Performance of linear classifier probe on teacher's intermediate checkpoints to predict the non-terminals at different levels of the PCFG generation tree. We observe that the probe's performance is $> 95\%$ of the final probe performance by the middle of the second phase, indicating the model has almost learned the underlying PCFG features by this time.

with the logits of the first intermediate teacher checkpoint till step $T$ and then the teacher is switched to the final teacher checkpoint.

*Low-temperature distillation:* We focus on distillation with a small temperature of $\tau = 10^{-4}$ (in Equation (1)), for the following reasons. First, as discussed in Section 3, it removes any potential regularization effects induced by soft labels. Moreover, using such a small temperature corresponds to training with the top-1 predictions of the teacher model, which is more memory-efficient compared to training with the full teacher logits, especially when the vocabulary size is large.

### D.7 ADDITIONAL CURRICULUM PROBING ON THE TEACHER'S CHECKPOINTS

In this section, we study the performance of different progressive distillation variants and compare them to one-shot distillation. As per our experiments in Figure 4, we use the $8$ teacher checkpoints selected for supervision. In Figure 24, we compare one-shot distillation to the two variants of progressive distillation, i.e. $8$-shot Equal-split and $8$-shot $\frac{T_0}{2}$-Equal-split distillation. We observe that both variants of progressive distillation help the student learn faster than one-shot distillation, and the gap diminishes as the students are trained for longer. The optimal strategy for progressive distillation depends on the training budget for the student. For training steps lower than the teacher's $T_0$ budget, $\frac{T_0}{2}$-Equal-split distillation slightly performs better than $T_0$-Equal-split distillation, which changes as we train longer. **To keep things simple, we focus on $\frac{T_0}{2}$-Equal-split progressive distillation in all of our subsequent experiments.**

### D.8 ABLATIONS WITH HYPERPARAMETERS

**Ablation with temperature** Here, we compare progressive distillation and one-shot distillation at temperature $1$ and temperature $10^{-4}$ (representing hard label supervision) (Figure 25). We observe that progressive distillation at temperature $10^{-4}$ performs better than one-shot distillation at both temperatures. However, progressive distillation at temperature $1$ can perform worse than one-shot distillation for a stronger student. We keep explorations on the effect of temperature on the algorithms as future work.

**Ablation with mask rate** In Figure 26, we compare progressive distillation with one-shot distillation at different masking rates. We observe that at all masking rates, progressive distillation performs better than one-shot distillation.

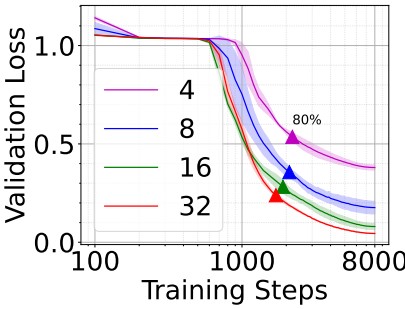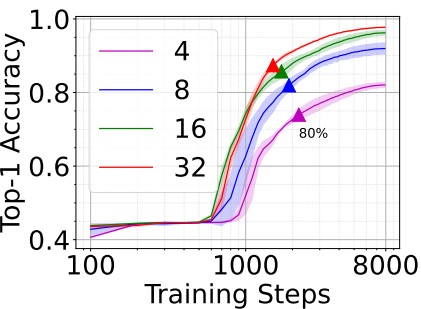

Figure 23: Comparison of BERT's training behavior on `cfg3b` with varying numbers of attention heads (where the embedding dimension scales linearly with the number of attention heads) over $8 \times 10^3$ training steps. The x-axis represents the number of training steps and is in log scale. Larger BERT models show an earlier and more pronounced drop in loss/increase in accuracy compared to smaller models. For reference, each training curve is annotated at the point where the model reaches $80\%$ of its performance at the final step.

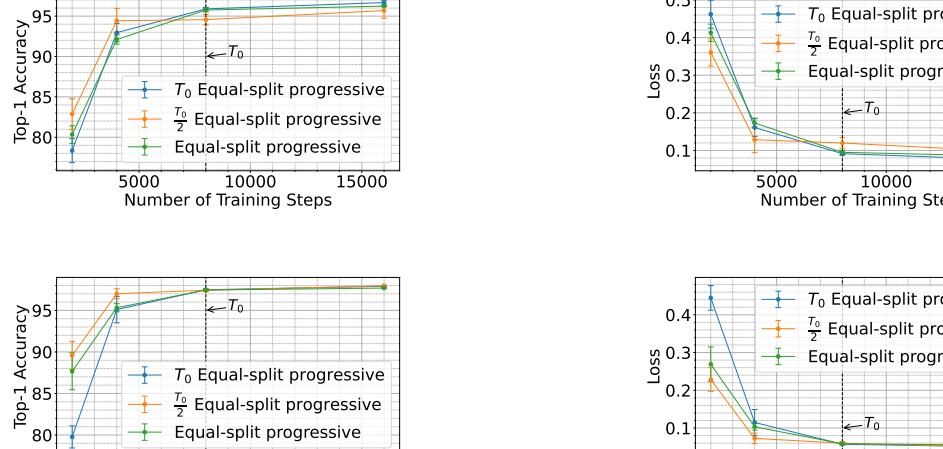

Figure 24: Experiments on BERT (Left to right/top to bottom): (a), (b) show the comparisons for an 8-attention head student, (c), (d) show the comparisons for a 16-attention head student. We observe differences between the different variants of progressive distillation at different training steps. For training steps lower than the teacher's (marked by $T_0$), $T_0/2$-Equal-split progressive distillation is better, implying that for shorter training, we shouldn't try to fit all the teacher's checkpoints. The trend reverses as the training sample budget approaches $T_0$ and beyond.

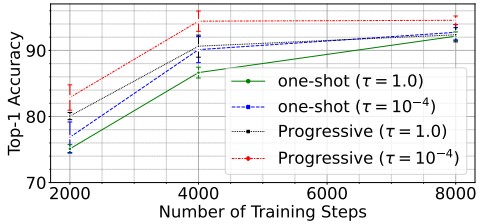 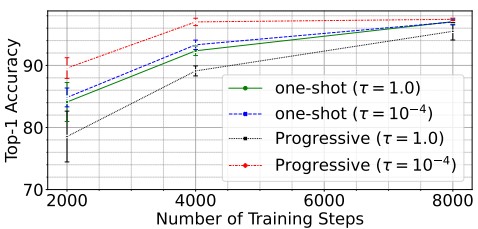

(a) Number of heads=8         (b) Number of heads=16

Figure 25: The experiments above compare progressive distillation and one-shot distillation at temperature $1$ and $10^{-4}$ (representing hard label supervision) for PCFGs `cfg3b` at masking rate $30\%$ using a BERT model with $8$ attention heads (left)/ $16$ attention heads (right), per head dimension $8$, and $4$ layers. We observe that progressive distillation with hard labels performs better than one-shot distillation at temperatures $1$ and $10^{-4}$. However, progressive distillation at temperature $1$ can perform worse than one-shot distillation for stronger student. We keep explorations on the effect of temperature on the algorithms as future work. Here, we use $\frac{T_0}{2}$-Equal-split progressive distillation as progressive distillation, where $T_0 = 8000$ is the total number steps used for teacher training.

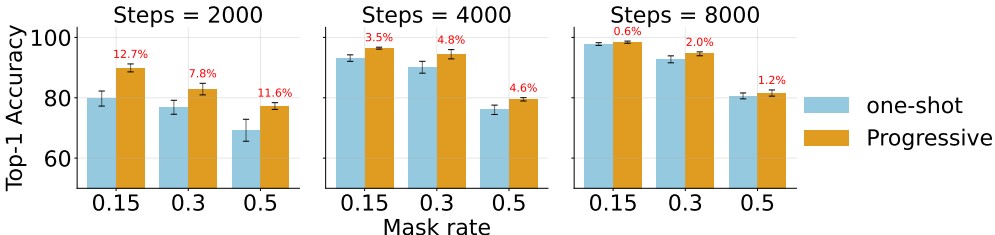

Figure 26: The experiments above compare progressive distillation and one-shot distillation for PCFGs `cfg3b` at different masking rates using a BERT model with $8$ attention heads, per head dimension $8$, and $4$ layers. The relative gap between the performance of progressive distillation and one-shot distillation have been reported on the bar plots. We observe that progressive distillation performs better than one-shot distillation at all masking rates, with the gap diminishing with the number of training steps. Here, we use $\frac{T_0}{2}$-Equal-split progressive distillation as progressive distillation, where $T_0 = 8000$ is the total number steps used for teacher training.

**Ablation with difficulty of PCFG** In Figure 27, we compare progressive distillation with one-shot distillation with increasing difficulty of the underlying PCFG. The benefit of progressive distillation over one-shot distillation is influenced by the model's capacity and the specific PCFG being trained.

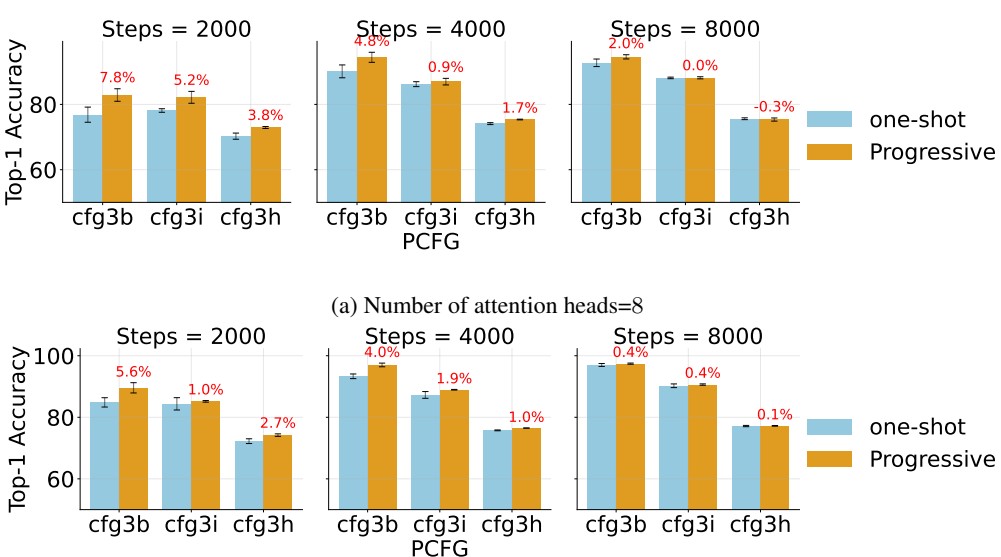

(a) Number of attention heads=8

(b) Number of attention heads=16

Figure 27: The experiments above compare progressive distillation and one-shot distillation for PCFGs `cfg3b`, `cfg3h`, and `cfg3i` at masking rate $30\%$ using BERT models with $8/16$ attention heads, per head dimension $8$, and $4$ layers. The relative gap between the performance of progressive distillation and one-shot distillation have been reported on the bar plots. The benefit of progressive distillation over one-shot distillation is influenced by the model's capacity and the specific PCFG being trained. For instance, on `cfg3i`, the student model can only achieve a top-1 accuracy of $75\%$. Progressive distillation reaches this within 2000 steps but fails to improve further, resulting in minimal gains over one-shot distillation when compared with `cfg3b`. The comparisons are at temperature $\tau = 10^{-4}$. Here, we use $\frac{T_0}{2}$-Equal-Split Progressive Distillation as Progressive Distillation, where $T_0 = 8000$ is the total number steps used for teacher training. Our teacher is a BERT model with 32 attention heads, per head dimension $8$, and $4$ layers, which doesn't train on `cfg3g` and `cfg3f`, hence we don't report the performance of the student on these PCFGs.

## E AUTOREGRESSIVE TRAINING WITH GPT2

**Setting:** Similar to experiments on BERT, we train GPT2 models of depth 4 with $\{8, 16, 32\}$ attention heads, while keeping the dimension per attention head fixed at $8$.

**A brief introduction into GPT models:** GPT models are trained with the auto-regressive loss. The teacher and student models operate on sequences of input domain $f_{\mathcal{T}} : \mathcal{X}^h \to \mathbb{R}^C$ and $f_{\mathcal{S}} : \mathcal{X}^h \to \mathbb{R}^C$, where the input sequence length $h$ can be arbitrary. Denote the length-$h$ input sequence as $\mathbf{x} := [x_1, \cdots, x_h]$, and denote $\mathbf{x}_{i:j}$ as the subsequence $[x_i, \cdots, x_j]$ (i.e. the indexing is inclusive on both ends). The cross entropy loss for next-token prediction training on $\mathbf{x}$ is given by

$$\frac{1}{h} \sum_{i=1}^{h} \text{KL}(\boldsymbol{e}_{x_i} \| p_{\mathcal{S}}(\mathbf{x}_{1:i-1}))),$$

where $\boldsymbol{e}_{x_i}$ denotes a one-hot vector with 1 in $x_i$th coordinate. We take a different approach, where we compare the algorithms at different difficult levels, by training on a subset of tokens in each sequence. The subsets that we consider are the boundary tokens at different levels of PCFG generation (recall Figure 3).

Formally, if $\mathcal{C}^{(\ell)}(\boldsymbol{x})$ represents the set of level-$\ell$ boundary tokens, then we define the cross entropy loss and the distillation loss corresponding to boundary tokens at any level $\ell$ of the PCFG as

$$\ell^{(\ell)}(\mathbf{x}; f_{\mathcal{S}}) = \frac{1}{\left|\mathcal{C}^{(\ell)}(\boldsymbol{x})\right|} \sum_{i:x_i \in \mathcal{C}^{(\ell)}(\boldsymbol{x})} \text{KL}(\boldsymbol{e}_{x_i} \| p_{\mathcal{S}}(\mathbf{x}_{1:i-1})); \tag{9}$$

$$\ell_{\text{DL}}^{(\ell)}(\mathbf{x}; f_{\mathcal{S}}, f_{\mathcal{T}}) = \frac{1}{\left|\mathcal{C}^{(\ell)}(\boldsymbol{x})\right|} \sum_{i:x_i \in \mathcal{C}^{(\ell)}(\boldsymbol{x})} \text{KL}(p_{\mathcal{T}}(\mathbf{x}_{1:i-1}; \tau) \| p_{\mathcal{S}}(\mathbf{x}_{1:i-1})). \tag{10}$$

There are a few remarks that need to be made about the above loss function. First, note that the subsets satisfy the condition $\mathcal{C}^{(\ell_1)}(\boldsymbol{x}) \subseteq \mathcal{C}^{(\ell_2)}(\boldsymbol{x})$ for all $\ell_1 \geq \ell_2$. Hence, the loss $L^{(\ell_2)}$ includes loss $L^{(\ell_1)}$ for all $\ell_1 \geq \ell_2$ and losses $L \in \{\ell, \ell_{\text{DL}}\}$. Second, $L^{(1)}$ will average the losses at all tokens, which is the standard auto-regressive loss used in practice to train large language models.

We focus on `cfg3f` that has 6 levels in the generation process, and we report the behavior of the models when trained with losses $L^{(2)}, L^{(3)}, L^{(4)}$, with $L \in \{\ell, \ell_{\text{DL}}\}$. We focus on $\frac{T_0}{2}$-Equal-split progressive distillation.

**Definitions for $M_{\text{robust}}$ and $M_{\text{close}}$.** Similar to our experiments on BERT, we track the change in the model's predictions with and without the $n$-gram context tokens. However, as the model is trained autoregressively, we need to change our definitions of $M_{\text{robust}}$ and $M_{\text{close}}$ from Equations (5) and (6), as well as the definition of $n$-grams.

For a $h$ length sentence $\boldsymbol{x} \in v^h$ and for $i \in [h]$, we define the $n$-gram neighboring context around the $i_{th}$ token as the set of tokens at positions within $n-1$ distance to the left from $i$, i.e. the set $\{x_j\}$ for $i - n < j < i$.

For $M_{\text{close}}$ on a teacher $f_{\mathcal{T}}$ and ngram length $n$, we measure the TV distance between the model's probability distributions of the model at any position $i$ when all the tokens at positions $1, 2, \cdots, i-1$ are available, and when only the tokens in the neighboring $n$-gram context window are available (i.e. at positions $i - n + 2, \cdots, i - 1$)[15]

$$M_{\text{close}}(f_{\mathcal{T}}, \mathbf{x}, i, n) = \text{TV}(p_{\mathcal{T}}(\mathbf{x}_{1:i-1}), p_{\mathcal{T}}(\mathbf{x}_{i-n+1:i-1})). \tag{11}$$

For $M_{\text{robust}}$ on a teacher $f_{\mathcal{T}}$ and an n-gram length $n$, we measure the total variation (TV) distance between the model's probability distributions at any position $i$, considering two scenarios: one where all tokens at positions $1, 2, \ldots, i-1$ are available, and another where the tokens within the $n$-gram context window are masked. However, since the attention mechanism in GPT requires a token at position $i - 1$ before it can predict $x_i$ and we don't have a special token to replace the masked tokens, we cannot remove that specific token from the context. Therefore, we keep the token at position $i - 1$ intact while masking the other tokens within the $n$-gram context window. We refer to this modified approach as "skip $n$-gram."

$$M_{\text{robust}}(f_{\mathcal{T}}, \mathbf{x}, i, n) = \text{TV}(p_{\mathcal{T}}(\mathbf{x}_{1:i}), p_{\mathcal{T}}(\mathbf{x}_{\{1, \cdots, i-n+1, i\}}))). \tag{12}$$

### E.1 Observations

**Teacher's behavior during training** Figure 29 shows the loss behavior of a teacher run. We observe 2 distinct phases of training: a rapid loss drop phase in the first $10\%$ of training, and a final phase of slow loss drop till end of training. In Figure 28, we compare the training accuracy behavior across models of different sizes. At log scale, we observe a very small dormant phase in the training behavior at the start of training. Larger models transition to the rapid loss drop phase faster than smaller models and also show a more prominent change in this phase.

**Teacher's checkpoint selection for progressive distillation** As outlined in the previous section, we select the first supervision checkpoint at roughly the middle of the first phase ($1/20_{\text{th}}$ fraction of training), and the other checkpoints are selected at $\{i/20\}_{i=2}^{20}$ fractions of training.

**Similar inflection points in loss as BERT and an implicit curriculum:**

---

[15]others are simply masked out during attention score computation to avoid shifts in position embeddings.

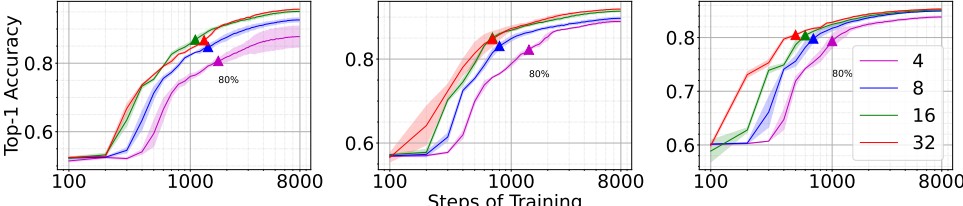

Figure 28: (left to right) Models are trained with the cross entropy loss $\ell^{(4)}, \ell^{(3)}, \ell^{(2)}$ respectively. Here, we compare GPT's training behavior with cross entropy loss on `cfg3f` with varying numbers of attention heads (where the embedding dimension scales linearly with the number of attention heads) over $8 \times 10^3$ training steps. Larger models show an earlier and more pronounced increase in performance compared to smaller models. For reference, each training curve is annotated at the point where the model reaches $80\%$ of its performance at the final step.

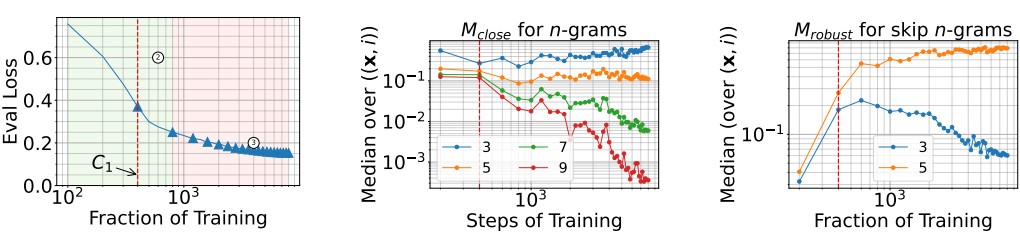

(a) Loss behavior of teacher model     (b) Median of $M_{\text{close}}(f_{\mathcal{T}}, \mathbf{x}, i, n)$    (c) Median of $M_{\text{robust}}(f_{\mathcal{T}}, \mathbf{x}, i, n)$ over $\mathbf{x}, i$ for different $n$    over $\mathbf{x}, i$ for different $n$

Figure 29: Experiments on GPT: Behavior of teacher model when trained on `cfg3f` with cross entropy loss: $\ell^{(3)}$. We observe two distinct phases; (2) a rapid drop in loss phase, and (3) slow drop in loss till end of training. The rapid loss drop phase signifies a transition phase for the model, similar to one we observed for hierarchical boolean data (Section 3). All selected checkpoints for progressive distillation are marked by triangles. The first teacher checkpoint is roughly picked at the center of the second phase. The rest of the checkpoints are picked at training steps that are multiples of the first one. (b) and (c) show inflection points in the teacher's predictions with full context and with/without $n$-gram contexts at the selected checkpoint.

We observe inflection points in the model's behaviors at the first selected checkpoint. Similar to our observations on BERT, we observe a curriculum on the reliance of the model's predictions on 3-gram predictions (Figure 29). Hence, we check whether progressive distillation can help train a smaller model faser.

**(R7) Progressive distillation helps train smaller model faster** In Figures 30 and 31, we compare one-shot distillation to $\frac{T_0}{2}$-Equal-split progressive distillation. We observe that progressive distillation help the student learn faster than one-shot distillation, and the gap diminishes as the students are trained for longer. However, the gap between progressive distillation and distillation decreases as more tokens are involved in the loss function i.e. the gap is smaller for loss $L_0^{(2)}$ compared to loss $L_0^{(4)}$. We conjecture that auto-regressive training with all tokens involved provides a strong curriculum for the model to learn the structure of the language. We keep a thorough study of this analysis to future work.

**Does longer training lead to degradation in performance for progressive distillation?**

One primary concern is whether progressive distillation can lead to degradation in the student model's performance in longer run. We extended all our GPT-2 experiments in Figures 30 and 31 to 16,000 steps. We do not observe any degradation in the performance of the progressively distilled models in Table 1, which shows that progressive distillation stays useful in the long run.

To further evaluate the learned representations, we measured the performance of a linear classifier when trained on top of the output embeddings on the non-terminal prediction task (Definition 4.3).

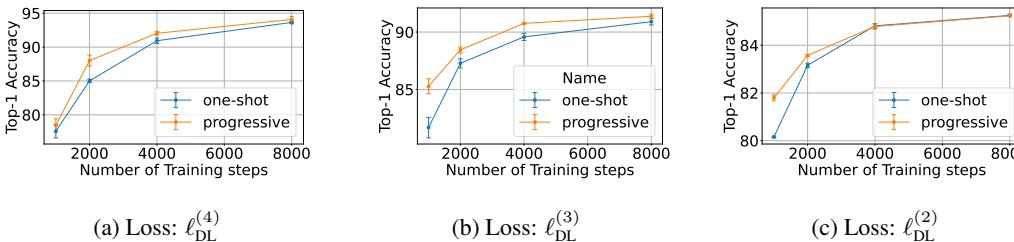

(a) Loss: $\ell_{\mathrm{DL}}^{(4)}$          (b) Loss: $\ell_{\mathrm{DL}}^{(3)}$          (c) Loss: $\ell_{\mathrm{DL}}^{(2)}$

Figure 30: Experiments on GPT (Left to right) for an 8- attention head model at different losses. Here, progressive distillation refers to $\frac{T_0}{2}$-Equal-split progressive distillation. We observe that progressive distillation outperforms one-shot distillation at all training sample budgets, with the gap diminishing with increasing training sample budget. The gap between progressive distillation and distillation decreases as the number of tokens involved in the loss function increases i.e. the gap is smaller for loss $L_0^{(2)}$ compared to loss $L_0^{(4)}$.

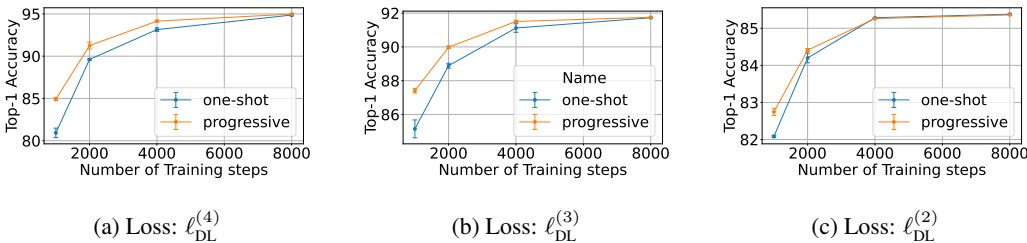

(a) Loss: $\ell_{\mathrm{DL}}^{(4)}$          (b) Loss: $\ell_{\mathrm{DL}}^{(3)}$          (c) Loss: $\ell_{\mathrm{DL}}^{(2)}$

Figure 31: Experiments on GPT (Left to right) for a 16-attention head model at different losses. Here, progressive distillation refers to $\frac{T_0}{2}$-Equal-split progressive distillation. We observe that progressive distillation outperforms one-shot distillation at all training sample budgets, with the gap diminishing with increasing training sample budget. The gap between progressive distillation and distillation decreases as the number of tokens involved in the loss function increases i.e. the gap is smaller for loss $L_0^{(2)}$ compared to loss $L_0^{(4)}$.

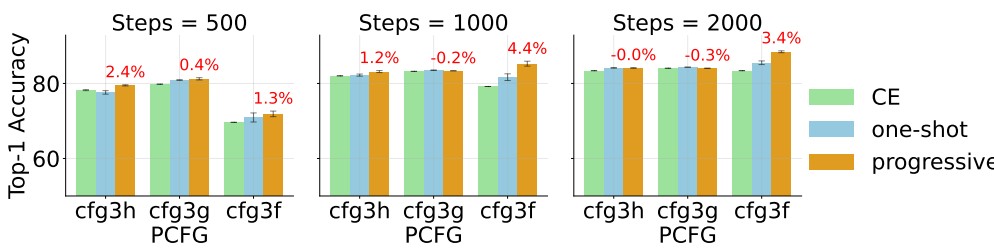

Figure 32: The experiments above compare progressive distillation and one-shot distillation for PCFGs `cfg3h`, `cfg3g`, and `cfg3f` on GPT models with 8 attention heads (each head having a dimension of 8) and 4 layers. The models were trained using the distillation loss $L_0^{(3)}$. The relative performance gap between progressive and one-shot distillation is presented in the bar plots. Notably, the advantage of progressive distillation over one-shot distillation depends on the specific PCFG being trained. For example, with `cfg3f`, the student model can achieve beyond $90\%$ top-1 accuracy, and progressive distillation allows it to reach this more quickly. In contrast, for `cfg3g`, the student model's top-1 accuracy plateaus at $84\%$, and after $500$ steps, progressive distillation shows only marginal gains over one-shot distillation. All comparisons were made at a temperature $\tau = 10^{-4}$. Here, progressive distillation refers to $\frac{T_0}{2}$-Equal-split progressive distillation, where $T_0 = 8000$ denotes the total number of steps for teacher training. The teacher model is a GPT with 32 attention heads, each with a dimension of 8, and 4 layers.

Table 1: **Progressive distillation remains beneficial during extended training.** We train GPT-2 student models with 8 and 16 attention heads for longer. We evaluate their performance using top-1 accuracy on the next token prediction task from their respective pre-training objectives. Our results show that progressively distilled models do not exhibit any performance degradation compared to those trained with one-shot distillation.

| Number of attention heads of student model | Training loss | Steps = 8000 | | Steps = 16000 | |
|---|---|---|---|---|---|
| | | Progressive | One-shot | Progressive | One-shot |
| 8 | $\ell_{\text{DL}}^{(3)}$ | $94.3_{(0.4)}$ | $93.4_{(0.4)}$ | $94.4_{(0.1)}$ | $93.1_{(0.8)}$ |
| | $\ell_{\text{DL}}^{(4)}$ | $90.5_{(0.5)}$ | $90.7_{(0.1)}$ | $91.0_{(0.1)}$ | $90.4_{(0.1)}$ |
| | $\ell_{\text{DL}}^{(5)}$ | $85.1_{(0.1)}$ | $85.1_{(0.2)}$ | $84.9_{(0.1)}$ | $84.8_{(0.1)}$ |
| 16 | $\ell_{\text{DL}}^{(3)}$ | $95.1_{(0.0)}$ | $95.0_{(0.2)}$ | $95.1_{(0.1)}$ | $94.9_{(0.2)}$ |
| | $\ell_{\text{DL}}^{(4)}$ | $91.7_{(0.1)}$ | $91.7_{(0.0)}$ | $91.8_{(0.0)}$ | $91.7_{(0.0)}$ |
| | $\ell_{\text{DL}}^{(5)}$ | $85.3_{(0.0)}$ | $85.3_{(0.0)}$ | $85.3_{(0.0)}$ | $85.3_{(0.0)}$ |

The classifier trained on embeddings from progressively distilled models performed competitively or exceeded the performance of classifiers trained on embeddings from one-shot distilled models (Table 2). These results confirm that progressive distillation does not compromise the long-term generalization of the learned embeddings.

Table 2: **Progressive distillation leads to favorable linear probe results on non-terminal prediction** (Definition 4.3), compared to one-shot distilled models.

| Number of attention heads of student model | Training loss | Steps = 8000 | | Steps = 16000 | |
|---|---|---|---|---|---|
| | | Progressive | One-shot | Progressive | One-shot |
| 8 | $\ell_{\text{DL}}^{(3)}$ | $79.4_{(0.4)}$ | $78.8_{(0.4)}$ | $79.5_{(0.3)}$ | $78.3_{(0.7)}$ |
| | $\ell_{\text{DL}}^{(4)}$ | $79.6_{(0.7)}$ | $79.6_{(0.2)}$ | $80.2_{(0.1)}$ | $79.3_{(0.2)}$ |
| | $\ell_{\text{DL}}^{(5)}$ | $80.8_{(0.4)}$ | $80.7_{(0.5)}$ | $80.2_{(0.2)}$ | $80.1_{(0.4)}$ |
| 16 | $\ell_{\text{DL}}^{(3)}$ | $81.0_{(0.1)}$ | $80.4_{(0.4)}$ | $80.9_{(0.2)}$ | $80.4_{(0.4)}$ |
| | $\ell_{\text{DL}}^{(4)}$ | $81.9_{(0.1)}$ | $81.8_{(0.2)}$ | $81.9_{(0.1)}$ | $81.7_{(0.1)}$ |
| | $\ell_{\text{DL}}^{(5)}$ | $82.1_{(0.2)}$ | $82.1_{(0.3)}$ | $82.2_{(0.1)}$ | $82.1_{(0.2)}$ |

Table 3: **Performance comparison on downstream tasks.** We pick the checkpoints from Figure 6 that have been trained with $120 \times 10^3$ training steps on each individual method and fine-tune them further with Adam optimization on different downstream tasks. Models trained with progressive distillation outperform others by an average of 0.9 points across all downstream tasks.

| Method | Sentiment | | NLI | | | Topic Classification | Average |
|---|---|---|---|---|---|---|---|
| | SST-2 | IMDB | QNLI | MNLI | SNLI | AG-NEWS | |
| Progressive | **86.9** (0.5) | **90.7** (0.2) | **84.4** (0.2) | **74.8** (0.1) | **86.0** (0.2) | **94.1** (0.1) | **86.2** |
| One-shot | 86.3 (0.6) | 90.4 (0.1) | 82.6 (0.1) | 72.8 (0.3) | 85.7 (0.1) | 93.9 (0.1) | 85.3 |
| CE | 85.5 (0.6) | 90.6 (0.0) | 83.1 (0.1) | 73.5 (0.0) | 85.3 (0.2) | 94.0 (0.1) | 85.3 |
| Teacher | 87.9 (0.6) | 91.4 (0.2) | 86.2 (0.3) | 78.4 (0.3) | 87.9 (0.2) | 94.4 (0.1) | 87.7 |

## F   DETAILS ON WIKIPEDIA + BOOKS EXPERIMENTS

We use the same hyperparameters for Adam training as our experiments on BERT and PCFG in Appendix D.6. However, we fix the peak learning rate to $10^{-4}$ (Devlin et al., 2018) in each case to minimize computation costs.

**Fine-tuning on downstream tasks**: We further report the performance of the trained student models (in Figure 6, we pick the checkpoints that have been trained with $120 \times 10^3$ training steps on each individual method) on various downstream tasks after fine-tuning in Table 3. We train the models with Adam optimizer with 3 epochs and cosine learning rate, and vary the peak learning rate in $\{1, 2, 5\} \times 10^{-5}$. Our results indicate that models trained with progressive distillation outperform others by an average of 0.9 points across all downstream tasks. This suggests that progressive distillation enables the student model to acquire generalizable features from the teacher during pre-training, which can also help improved generalization on different downstream tasks after fine-tuning.

