# OpenReview forum: "Progressive distillation induces an implicit curriculum"
_ICLR.cc/2025/Conference — ICLR 2025 Oral_

### Official Review · Reviewer_sKbn · 2024-10-29

**Soundness:** 4
**Presentation:** 4
**Contribution:** 3
**Rating:** 8
**Confidence:** 3

**Summary:**

Traditional knowledge distillation relies on a single, powerful teacher model to train a smaller student model. However, it has been observed that stronger teachers do not always yield better-performing students. To address this issue, the authors explore progressive distillation, where the student model is incrementally guided by a sequence of increasingly capable teacher checkpoints. Through theoretical analysis and extensive experiments on tasks such as sparse parity and probabilistic context-free grammars (PCFGs), the paper demonstrates that progressive distillation accelerates the optimization process, achieving improved performance with fewer training steps compared to one-shot distillation and direct learning from data.

**Strengths:**

Strengths
* **Intuitive Motivation and Theoretical Foundation:** The paper is well-motivated, addressing a significant challenge in the effective distillation from large to small models. The theoretical underpinnings are well grounded, with rigorous proofs using the sparse parity example demonstrating why progressive distillation is effective.
* **Empirical Validation Across Diverse Tasks:** The authors conduct comprehensive experiments on both synthetic tasks (sparse parity and PCFGs) and realistic settings (training BERT on Wikipedia and Books datasets). This breadth of evaluation underscores the generalizability of their findings.

**Weaknesses:**

**Weaknesses**

- **Scalability to Larger Models and Diverse Architectures**: While the experiments include models like BERT, the paper does not thoroughly investigate how progressive distillation scales with increasingly large models (in the main paper).

- **Effectiveness in GPT Models**: In Appendix E, the paper examines autoregressive training with GPT-2. Although progressive distillation accelerates learning speed, the performance plateaus around 8,000 training steps. It remains unclear why this convergence occurs and what the outcomes would be if training continued beyond 8,000 steps (e.g., up to 20,000 steps).

- **Potential for Degenerate Features**: While progressive distillation leverages an implicit curriculum to identify key patterns, there is a concern that this curriculum might lead to degenerate features that could hinder long-term generalization. This issue could be further investigated by extending the training duration in GPT-2 to ensure that no negative consequences arise from prolonged training.

**Questions:**

* **Teacher Checkpoint Selection Across Domains**: What guidelines or heuristics can assist in selecting intermediate teacher checkpoints effectively for diverse tasks such as visual classification, image generation, or reinforcement learning?
* **Joint Training of Student Models:** What are the effects of training small student models jointly with large teacher models, instead of using progressive distillation with intermediate checkpoints? Does this approach provide a similar implicit curriculum that benefits the student’s learning?

I am willing to improve my score if the authors demonstrate that progressive distillation remains beneficial compared to one-shot distillation in the long run for large models, such as GPT-2.

---

> ### Author Response · Authors · 2024-11-21
>
> We thank the reviewer for their positive comments and suggestions about our paper. Please find our responses to your questions below.
>
> **Q1: Could progressive distillation lead to degenerate features that hinder long-term generalization?**
>
> **Response:**
> We show, by two new sets of experiments, that features learned by progressive distillation stay useful in the long run.
>
> **Setting 1: Fine-tuning experiments on BERT (trained on Wikipedia+Books, Figure 6)**
>
> On our trained BERT models on Wikipedia and Books dataset (figure 6), we observe that progressive distilled models have $0.9$ percent better performance on downstream tasks after fine-tuning. This shows that the features learned by progressive distilled models prove to be more helpful for downstream performance generalization.
>
> | Method      | SST-2       | IMDB        | QNLI        | MNLI        | SNLI        | AG-NEWS    | Avg|
> |-------------|-------------|-------------|-------------|-------------|-------------|--------|--------|
> | Progressive | **86.9** (0.5)  | **90.7** (0.2)  | **84.4** (0.2)  | **74.8** (0.1)  | **86.0** (0.2)  | **94.1** (0.1)  | **86.2** |
> | One-shot    | 86.3 (0.6)  | 90.4 (0.1)  | 82.6 (0.1)  | 72.8 (0.3)  | 85.7 (0.1)  | 93.9 (0.1)  | 85.3 |
> | CE          | 85.5 (0.6)  | 90.6 (0.0)  | 83.1 (0.1)  | 73.5 (0.0)  | 85.3 (0.2)  | 94.0 (0.1)  | 85.3 |
> | Teacher     | 87.9 (0.6)  | 91.4 (0.2)  | 86.2 (0.3)  | 78.4 (0.3)  | 87.9 (0.2)  | 94.4 (0.1)  | 87.7 |
>
>
> **Setting 2: Longer training on GPT-2 (trained on CFGs, Appendix E)**
>
> In our GPT-2 experiments, both the student and the teacher had converged by 8000 training steps on the PCFGs that we used. We want to clarify that 8000 training steps use roughly 4M training samples. In all of our experiments (Figures 29-31), progressive distillation provides acceleration in training when compared at 2000 steps, which translates to a million training samples.
>
> On the request of the reviewer, we extended all our GPT-2 experiments to 16,000 steps. We do not observe any degradation in the performance of the progressively distilled models. Top-1 prediction accuracy:
>
> | Student configuration | Training loss | Steps =         8000             |Steps =         8000 | Steps =  16000     |      Steps =  16000                 |
> |-------------|---------------|--------------------|---------------------|--------------------|---------------------|
> |             |               | **Progressive**        |  **One-shot**          | **Progressive**        | **One-shot**            |
> |   **8 heads, 64 width**        | $ℓ^{(3)}_{DL} $    | 94.3 (0.4)         | 93.4 (0.4)          | 94.4 (0.1)         | 93.1 (0.8)          |
> |          |$ℓ^{(4)}_{DL}$     | 90.5 (0.5)         | 90.7 (0.1)          | 91.0 (0.1)         | 90.4 (0.1)          |
> |            | $ℓ^{(5)}_{DL}$      | 85.1 (0.1)         | 85.1 (0.2)          | 84.9 (0.1)         | 84.8 (0.1)          |
> |=============|===============|====================|=====================|====================|=====================|
> |  **16 heads, 128 width**           |$ℓ^{(3)}_{DL} $     | 95.1 (0.0)         | 95.0 (0.2)          | 95.1 (0.1)         | 94.9 (0.2)          |
> |         | $ℓ^{(4)}_{DL} $     | 91.7 (0.1)         | 91.7 (0.0)          | 91.8 (0.0)         | 91.7 (0.0)          |
> |         | $ℓ^{(5)}_{DL} $ | 85.3 (0.0)         | 85.3 (0.0)          | 85.3 (0.0)         | 85.3 (0.0)          |

---

> ### Author Response · Authors · 2024-11-21
>
> [Continued]
>
> **Non-terminal prediction task (Measure 3, line 424):**
>
> To further evaluate the learned representations, we measured the performance of a linear classifier when trained on top of the output embeddings on the non-terminal prediction task. The classifier trained on features derived from progressively distilled models performed competitively or exceeded the performance of classifiers trained on features from one-shot distilled models. These results confirm that progressive distillation does not compromise the long-term generalization of the learned features.
>
> Average Non-terminal prediction accuracy:
>
> | Student configuration | Training loss | Steps =         8000             |Steps =         8000 | Steps =  16000     |      Steps =  16000                 |
> |-------------|---------------|--------------------|---------------------|--------------------|---------------------|
> |             |               | **Progressive**        |  **One-shot**          | **Progressive**        | **One-shot**            |
> | **8 heads, 64 width**            |  $ℓ^{(3)}_{DL} $     | 79.4 (0.4)         | 78.8 (0.4)          | 79.5 (0.3)         | 78.3 (0.7)          |
> |          |  $ℓ^{(4)}_{DL} $       | 79.6 (0.7)         | 79.6 (0.2)          | 80.2 (0.1)         | 79.3 (0.2)          |
> |          |  $ℓ^{(5)}_{DL} $       | 80.8 (0.4)         | 80.7 (0.5)          | 80.2 (0.2)         | 80.1 (0.4)          |
> |=============|===============|====================|=====================|====================|=====================|
> | **16 heads, 128 width**           |  $ℓ^{(3)}_{DL} $      | 81.0 (0.1)         | 80.4 (0.4)          | 80.9 (0.2)         | 80.4 (0.4)          |
> |         |  $ℓ^{(4)}_{DL} $       | 81.9 (0.1)         | 81.8 (0.2)          | 81.9 (0.1)         | 81.7 (0.1)          |
> |         |  $ℓ^{(5)}_{DL} $   | 82.1 (0.2)         | 82.1 (0.3)          | 82.2 (0.1)         | 82.1 (0.2)          |

---

> ### Author Response · Authors · 2024-11-21
>
> **Q2: What guidelines can help select effective teacher checkpoints for tasks like visual classification, image generation, and reinforcement learning?**
>
> **Response**: The heuristics and guidelines for progressive distillation can vary significantly depending on the model's loss behavior during training. Our work specifically focuses on tasks where the loss exhibits dormant phases followed by fast phase transitions. We demonstrate that leveraging teacher checkpoints from fast phase transitions can substantially accelerate student training by enabling the model to bypass the slow, dormant phase and providing guidance on the intermediate features that are most beneficial for further learning.
>
> Extending our analysis to other diverse tasks will require building toy models that can simulate the input label relationship in these tasks. Additionally, our implicit curriculum perspective provides one possible framework through which we can measure the utility of intermediate teacher checkpoints. Expanding our framework to other perspectives and diverse domains through which larger models can facilitate more efficient and effective learning for smaller models stays an important research direction.
>
> **Q3: Does joint training of small student models with large teachers provide a similar implicit curriculum?**
>
> **Response**: Joint training can be considered equivalent to progressive distillation if the student model continually distills from the teacher checkpoints at every step, rather than at predefined intervals. Under this interpretation, our theory could be extended to such a setting, as the implicit curriculum would still be governed by the transitions in the teacher’s training phases.
>
> However, for the sake of simplicity in experimentation and to facilitate ablation studies, we focus on a setup where teacher was first trained to completion, and then the student model is trained using the logits from the teacher’s checkpoints sampled at regular intervals. This approach allows us to isolate and analyze the role of each teacher checkpoint in progressive distillation in a controlled manner.
>
> **Q4: Scalability to Larger Models and Diverse Architectures**
>
> **Response**: Our work isn’t proposing progressive distillation as a strategy to train smaller models, instead we want to understand how a larger model can help optimization in a smaller model. Given our interpretability analysis on intermediate checkpoints, scaling to larger architectures wasn’t feasible within the scope of our study.
>
> The algorithm of progressive distillation has been widely used in practice for efficient and improved knowledge distillation in different domains. Here are some references (many of which have been cited in our paper):
>
> - Vision models
>     - Harutyunyan et al. 2023. Supervision Complexity and its Role in Knowledge Distillation.
>     - Mirzadeh et al. 2019. Improved Knowledge Distillation via Teacher Assistant.
>     - Cho & Hariharan 2019. On the efficacy of knowledge distillation.
>     - Beyer et al.’2021. Knowledge distillation: A good teacher is patient and consistent
>     - Cao et al’23. Learning Lightweight Object Detectors via Multi-Teacher Progressive Distillation (Object detection and segmentation)
> - Language models
>     - Gemini Google Team. Gemini 1.5: Unlocking multimodal understanding across millions of tokens of context.
>         - Gemini Flash has been reported to be progressive (online) distilled from Gemini Pro.
>     - Rezagholizadeh et al. Pro-KD: Progressive distillation by following the footsteps of the teacher

---

> > ### Comment · Reviewer_sKbn · 2024-11-25
> >
> > Thank you for the detailed response. I’m satisfied with the clarification and have updated my score accordingly.

---

### Official Review · Reviewer_9DA8 · 2024-11-03

**Soundness:** 3
**Presentation:** 4
**Contribution:** 4
**Rating:** 8
**Confidence:** 4

**Summary:**

The paper investigates how progressive distillation can accelerate the training of neural networks by leveraging an implicit curriculum provided in the form of intermediate teacher checkpoints. The authors conduct experiments on tasks like sparse parity, probabilistic context-free grammars (PCFGs), and real-world datasets ("Wikipedia and Books") using models like MLPs, Transformers, and BERT. The main claimed findings are:

1. Progressive distillation accelerates student learning compared to one-shot distillation.
2. An implicit curriculum emerges from intermediate teacher checkpoints, guiding the student through increasingly complex subtasks.
3. This implicit curriculum results in an observable empirical acceleration, and stems from provable benefits.

**Strengths:**

I liked the spirit of this paper, which I found complete, well-written, carefully designed and executed. Overall, I enjoyed the following strengths:

- **Experimental completeness:** The paper is generous in providing extensive empirical evidence across synthetic tasks (sparse parity, PCFGs) and real-world datasets (Wikipedia, Books).

- **Theoretical Analysis:** Offers mathematical proofs demonstrating sample complexity benefits in specific cases like sparse parity learning.

- **Memorable Observations:** Beyond the idea that checkpointed learning leads to faster optimization, it identifies phase transition in the teacher's training where intermediate checkpoints can be most beneficial to the student. These correspond to acquring new skills of increasing complexity.

- **Impact and Practical and actionable implications**

**Weaknesses:**

- **Alternative intepretations, e.g. winning subnetworks and the Lottery Ticket Hypothesis:**

The first thing I thought about, as a possible intepretation of the empirical (and theoretical) findings of the paper is the lottery ticket hypothesis (LTH), which could alternatively explain the benefits observed in progressive distillation. The hypothesis posits that within a large neural network, there exist smaller subnetworks (winning tickets) that can learn the tasks efficiently. Searching for these subnetworks takes long, but once they are identified, training only on them makes the learning faster.

So while the paper frames the intermediate checkpoints as providing an implicit curriculum of increasing task complexity, this could be reinterpreted as the teacher progressively revealing parts of the winning subnetwork, narrowing the student's search space at each step. Hence a crucial question is: could the novelty in using intermediate checkpoints be just an operationalization of the lottery ticket hypothesis in a distillation context (i.e. Guided Search) rather than a fundamentally new concept (implicit curriculum)?
There, the student model is being guided to explore progressively smaller regions of the solution space, as per LTH. These correspond to learning features of increasing complexity as the paper points out, but that's because that's how networks progress in their learning of the best parameters to solve a problem. By emphasizing the curriculum aspect, the paper might divert the "reader" from other factors contributing to accelerated learning, such as the inherent properties of the optimization landscape or the effects of network pruning (implicit or explicit).

And both interpretations would be aligned with the empirical and theoretical findings.

First, the fact that not all checkpoints provide good performance could be  due to the model learning more and more complex skills as it discovers the winning tickets, and not because the tasks are more and more complex (curriculum). Matter of fact, the training data is not going from simple tasks to complex ones, what goes from simple to complex is what the model has learned, not what is was trained on. Said differently, the discovery of progressively complex features could be the result/consequence of a guided search to efficient parameter configurations.

Second, looking at the sample complexity improvement that the authors prove:
"the total sample complexity needed for the student to reach ϵ-loss using progressive distillation with 2 checkpoints is Õ(2^k d^2 ϵ^−2 + k^3). However, one-shot distillation requires at least Ω(d^k−1, ϵ^−2) samples"
could perhaps be reinterpreted as: The student benefits from getting direct signal about which features are important (winning tickets), rather than having to discover them from scratch like in one-shot distillation.
Same goes for the theorem about intermediate checkpoints having stronger correlations with degree-1 monomials: could be read as a curriculum of increasing task complexity, or about how the teacher naturally learns (simple correlations first) and why intermediate checkpoints help (they provide clearer signal about important features).

Seen this way, calling this an "implicit curriculum" might be misleading because the task complexity is constant, and what is changing is the model's internal learning progression.

-  **Other minors**

The specific temperature values, as well as the choice of dramatically different temperatures (10^-4 vs 10^-20) based on vocabulary, could benefit from more rigorous exploration. Ok, the authors still acknowledge this as a limitation they defer to future work.

**Questions:**

The only perhaps unanswered question in this paper is: are simpler features learned first because they're easier, or because they're sufficient for initial loss reduction? Does the progression represent increasing task difficulty or natural optimization paths as the student discovers incrementally winning subnetworks?

As discussed above, it would be interesting to at least provide an analysis of how progressive distillation differs from or extends the Lottery Ticket Hypothesis (and the variants of progressive stacking/learning subnetworks etc). Would be (may be) interesting to discuss whether there could be experiments to show if the observed acceleration is due to narrowing the search space via subnetworks and how this impacts the interpretation of an implicit curriculum.

**Future experiments to isolate effects/figure out the right interpretations**

Below are just some ideas, probably mostly for future work.

It would be interesting to evaluate (in the experiments of Figure2), the corresponding sizes of the winning subnetworks as per the lottery ticket hypothesis.

One could also analyze the sparsity patterns of successful networks at different checkpoints, compare with explicit lottery ticket pruning approaches, then try to isolate whether the benefits come from parameter space guidance or feature complexity.

**Quick check of the literature**
The following is (slightly) related from a quick search, mainly for your interest:
https://arxiv.org/pdf/2402.05913 too recent of course.
Older work on progressive stacking: https://proceedings.mlr.press/v202/j-reddi23a.html  also only slightly related
But more generally, it might be useful to reprobe the related work, with the lens of using winning subnetworks to accelearate learning, perhaps checking how they form and how they help accelerating learning (whether there's an increase in task complexity ...)

---

> ### Author Response · Authors · 2024-11-21
>
> We thank the reviewer for their positive feedback, constructive comments, and suggestions for future exploration!
> Please find our responses below.
>
> **Q1: Alternative interpretations, e.g. the Lottery Ticket Hypothesis and winning subnetworks.**
>
> Thank you for the very interesting and detailed comments! We would like to clarify on the agreements and differences between the comments and claims in the paper.
>
> - **Curriculum vs guided search**: First, we’d like to clarify that “implicit curriculum” and “guided search” are not meant as alternative interpretations. In fact, “guided search” was the mental picture we had in mind throughout the project. What the paper shows is that implicit curriculum is a *mechanism* through which the guidance is provided.
>     - We also want to clarify a potential confusion based on the comment: “*Said differently, the discovery of progressively complex features could be the result/consequence of a guided search to efficient parameter configurations.*” — The discovery of progressively complex features is in the teacher, whereas the one being guided is the student.
> - **Connection to LTH or network pruning**: Both LTH and network pruning are based on the hypothesis that there exists a sparse subnetwork that can compute a similar function to the full network, which is about the *learner network*. In contrast, the implicit curriculum in our paper is about the *target function*. The same target function can be learned using different learner networks. For instance, the sparse parity can be learned using MLP or Transformer, both of which provides a low-degree curriculum towards sparse parity (Figure 2, Figure 12).
>     - “*The student benefits from getting direct signal about which features are important  (winning tickets)*” — To clarify, the teacher only provides supervision through the logits. For sparse parity, the logits are 2-dimensional, while the student has an input dimension of 100 and a hidden width of 100 or 1000. Hence it is impossible for the teacher to directly provide supervision on which subnetwork is the winning one, but the teacher’s logits can encode information towards an alternative function, which is what we call an implicit curriculum (e.g. a lower-degree polynomial instead of the target degree-k monomial).
> - **Task complexity**: The reviewer commented that “calling this an ‘implicit curriculum' might be misleading because the task complexity is constant.“ We’d like to clarify that while the complexity of the target function is constant, the function induced by intermediate teacher checkpoints often does not have a constant complexity [Nakkiran et al. 2019]. As the reviewer correctly pointed out, “what is changing is the (teacher) model's internal learning progression”, and this learning progression is the source of the implicit curriculum.
>
> *References*: Nakkiran et al. 2019. SGD on Neural Networks Learns Functions of Increasing Complexity
>
> **Q2: Effect of temperature**
>
> Thank you for the comment! Please note that Figure 14 in Appendix C.2 provides additional results on different choices of $\tau$, which demonstrate that setting the temperature as  $\tau = 10^{-4}$ or $\tau = 10^{-20}$ indeed does not affect the conclusion.
>
> Thank you very much also for the references on progressive stacking; this is an interesting connection and we will add a discussion in the camera ready version.
>
> Please let us know if you have more suggestions or concerns, and we will be happy to discuss more. Thank you!

---

> > ### Comment · Reviewer_9DA8 · 2024-11-27
> >
> > Thank you for your answer. I have no further comments. The discussion about guided search versus (implicit) curriculum is interesting, but is perhaps a philosophical one. *The question that remains for me is whether the winning subnetworks always necessarily come with an associated "skill"  or not*.
> > A winning subnetwork represents an efficient parameter configuration, but does that automatically mean it has learned a meaningful new capability or "skill"? What if winning subnetworks could simply be a more computationally efficient way to implement the same skill, rather than representing a new capability?
> > In any case, thank you for this work, I enjoyed reading your paper.

---

### Official Review · Reviewer_qfzz · 2024-11-07

**Soundness:** 4
**Presentation:** 4
**Contribution:** 3
**Rating:** 8
**Confidence:** 4

**Summary:**

The paper demonstrates how progressive distillation helps in better training of the student models by inducing an implicit curriculum. The experimental results demonstrate that progressive distillation results in a higher sample efficiency in all cases as well as a higher performance in some cases, as compared to vanilla knowledge distillation (referred to as "one-shot distillation" in the paper). The results also show that progressive distillation induces an implicit curriculum wherein the intermediate checkpoints provide a stronger learning signal and act as "stepping stones" for the student models during the learning process. These results are validated by experimenting across two different model architectures and 3 different tasks. In the case of the sparse parity task, authors also provide a theoretical proof of how progressive distillation with 2 checkpoints (one intermediate and one final) leads to a better sample complexity as compared to one-shot distillation.

**Strengths:**

* The paper is well-written and easy to read.
* The paper includes results on tasks across different complexity levels - going from a toy setting of sparse parity to PCFGs and then to a non-synthetic task of natural language modeling.
* Authors also run experiments across multiple model architectures, name MLPs and transformers of different sizes.
* The induced curriculum is discussed from a human interpretability point of view (i.e. showing the correlation between degree 1 monomials and the logits of the intermediate teacher checkpoint in the sparse parity task, and drawing similar analogy in the PCFG task).
* The paper (more specifically; the appendix) includes further extensive experimentation on the settings discussed in the main paper.

**Weaknesses:**

* There is a typo in Definition 4.3: I believe it should be "boundary of span(n^{(i)})" instead of boundary of n^{(i)}
* Discussion about how the relative sizes of teacher and student models were decided is missing. It would be interesting to see a study of how the performance is affected w.r.t the size of the student models
* Empirical analysis on tasks in the vision domain and with other model architectures such as CNNs and recurrent networks would strengthen the paper significantly.

**Questions:**

* How were the sizes of student models decided? Can the authors show some results on one task (preferably the natural language modeling OR the PCFG tasks) for what happens when the sizes of student models are varied while the size of the teacher size is kept constant?
* Can the authors run similar experiments on a simple multiclass classification task in the visual domain? Even results on something simple such as multiclass classification in CIFAR-100 with an ImageNet pretrained ResNet-18 finetuned on CIFAR-100 as the teacher, and a smaller CNN as the student would be interesting to see.

---

> ### Author Response · Authors · 2024-11-21
>
> We thank the reviewer for their positive feedback and constructive comments! Please find our responses below.
>
>
> **Q1: Effect of model size; relative size of the teacher vs student**
>
> This is a great question. For the current paper, we’ve chosen the sizes to ensure that there is a significant performance gap (in terms of either the optimization speed or the final performance) for teachers and students trained directly on the data; the exact sizes are chosen somewhat arbitrarily as long as a gap is observed.
>
> Below we summarize the empirical comparisons on sizes.
> - For sparse parity learned with MLP, when training on the label only, there is an inverse relationship between the model width and the optimization speed, as shown in Edelman et al. 2023 and replicated in our experiments (Figure 7 in Appendix C.2). In contrast, when learning from a teacher, students of different sizes (i.e. width 100 or width 1000) are able to follow the teacher and hence replicate the teacher’s accuracy curves (Figure 2, Figure 8). We did not try further small width as we think a 500-time size difference (i.e. a width-100 student and a width-50000 teacher) is sufficiently big.
>     - Additionally, Figure 15 shows similar findings for a hierarchical generalization of sparse parity.
> - For sparse parity learned with Transformers, we find that increasing the number of heads or the MLP width both lead to faster training (Figure 10). The precise relation between the model size and the learning speed is an interesting direction for future exploration: the answer would be more nuanced than that of MLP, as Transformers are not permutation invariant over the positions.
> - Our PCFG experiments use 4-layer Transformers with 32-head teachers, and 8-head or 16-head students. We observe similar trends for both students (Figure 5, Figure 24, Figure 25, Figure 27).
> - Our natural language experiments use 12-layer Transformers with 12 heads for the teacher and 4 heads for the students. We did not experiment with other sizes due to limited compute.
>
> Note that in all experiments, the teacher and the student share the same number of layers and differ only in some notion of “width” (the MLP width, or the number of heads). This is because we want to focus on optimization difficulty which is typically related to width; please refer to “Benefit of width in optimization” in Appendix A.1 for more details.
>
> Understanding depth is an interesting future direction. For instance, there have been results on the depth’s effect on representation capacities [Levine et al. 2020, Merrill & Sabharwal 2022, Sanford et al. 2024] and inductive biases [Tay et al. 2021, Petty et al. 2023, Ye et al. 2024]. Moreover, we note that prior work on vision (see references for the next question) typically vary the depth across the teacher and the student.
>
> *References*:
> - Levine et al. 2020. The Depth-to-Width Interplay in Self-Attention.
> - Merrill & Sabharwal 2022. The Parallelism Tradeoff: Limitations of Log-Precision Transformers.
> - Petty et al. 2023. The Impact of Depth on Compositional Generalization in Transformer Language Models.
> - Sanford et al. 2024. One-layer transformers fail to solve the induction heads task.
> - Tay et al. 2021, Scale Efficiently: Insights from Pre-training and Fine-tuning Transformers.
> - Ye et al. 2024. Physics of Language Models: Part 2.1, Grade-School Math and the Hidden Reasoning Process.
>
>
> **Q2: Can the results be extended to the vision domain?**
>
> Previous works have demonstrated that progressive distillation outperforms traditional knowledge distillation in vision tasks (cited in lines 93-99), which we highlight here:
> - Harutyunyan et al. 2023. Supervision Complexity and its Role in Knowledge Distillation.
>     - This is most related to our setup, where multiple teacher checkpoints are used to supervise the student.
> - Mirzadeh et al. 2019. Improved Knowledge Distillation via Teacher Assistant.
> - Cho & Hariharan 2019. On the efficacy of knowledge distillation.
>
> Extending our analysis to vision domain is indeed an interesting question. In all our experiments, we have a theoretical model to simulate input label distribution (for our analysis in NLP domain, we built our theory of PCFGs). However, we do not currently have a well-suited theoretical model to map input-label functions for vision tasks, which is why we have not conducted experiments in this domain.
>
> Thank you also for catching the typo for us. We hope our responses have addressed your concerns, and we are happy to discuss further if you have more suggestions or if there's anything unclear. Thank you!

---

> > ### Comment · Reviewer_qfzz · 2024-11-27
> >
> > Thank you for addressing my concerns in detail. I am satisfied with the explanation about the relative sizes of the student models w.r.t the teacher models. Attempting to extend the proposed theory to the visual domain is an interesting future direction indeed.
> >
> > I have raised my confidence rating.

---

### Official Review · Reviewer_ktaG · 2024-11-08

**Soundness:** 4
**Presentation:** 3
**Contribution:** 3
**Rating:** 6
**Confidence:** 4

**Summary:**

This paper provides theoretical and empirical analysis on how progressive distillation can be helpful to speed up knowledge distillation. Specifically, the author studies a toy use case in sparse parity with synthetic data to show both mathematically and empirically how using intermediate teacher checkpoints can assist student models to learn faster. Then they run progressive distillation experiments in PCFGs and BERT masked token prediction using Wikipedia and Books data to further verify their findings.

**Strengths:**

1. The paper is well written, and despite the complexity of the narrative, it is generally easy to follow, and I enjoyed the reading.
2. Though only applied to a simple use case, the mathematical analysis does provide useful insight about sample efficiency of progressive distillation.
3. The metrics selected in the analysis such as $\mathcal{M}_{robust}$ is quite useful to understand the feature learning aspect of the method.
4. The authors run experiments in various settings including three datasets, MLPs, BERT and GPT-2 (in the appendix) to show the gains can be generalized.

**Weaknesses:**

Despite the strength, I think the paper can be improved.
1. I understand the necessity to use a toy use case (sparse parity) to show rigorous mathematical analysis, but the following experiments can be more practical in order to provide stronger empirical evidence of the effectiveness of progressive distillation.
- Instead of masked token prediction, can run experiments in challenging NLP tasks such as QA, summarization and long-form generation.
- Can also experiment with more recent LLMs - GPT-2 in the appendix is a good start, but it is still a very outdated model.
2. Some writing (particularly Theorem 3.2) can be made even clearer. For example, L291-292, I didn't quite follow where higher degree odd polynomials come from. L296-297 "This gap allows .. to grow quickly with only ... samples." This statement isn't clear if I don't read the entire proof in the appendix. Please consider writing it in a more intuitive way in the main text.
3. Probably unfair as this is more of an analysis paper, but the overall contribution appears to be limited considering the scope of its experiments.

**Questions:**

1. For experiments in Figure 2, how do you explain the results that the best correlation (Middle) with in-support variable does not lead best accuracy (Left)?
2. L406, what's the formula for total variation?
3. How do we pick the most appropriate teacher checkpoints in other more complicated tasks?

---

> ### Author Response · Authors · 2024-11-21
>
> We thank the reviewer for their positive comments and valuable suggestions about our paper. Please find our responses to your questions below.
>
> **Q1: Instead of masked token prediction, can the authors run experiments in challenging NLP tasks such as QA, summarization and long-form generation?**
>
> We address the question from two aspects, NLP tasks after fine-tuning, and reasoning as a type of long-form generation.
>
> **Fine-tuning on NLP tasks:** We take the checkpoints trained with different methods on Wikipedia and Books dataset for 120k steps (figure 6) and fine-tune them on different GLUE tasks. We report the top-1 accuracy after fine-tuning. For each task, we append a prompt [Gao et al. 2021] to the end of each example (e.g. for sentiment task, we append ‘The sentiment is <mask>’)) and train a classification head on the <mask> token.  We observe that progressively distilled models perform better than other models across all tasks.
>
> | Method      | SST-2       | IMDB        | QNLI        | MNLI        | SNLI        | AG-NEWS    | Avg|
> |-------------|-------------|-------------|-------------|-------------|-------------|--------|--------|
> | Progressive | **86.9** (0.5)  | **90.7** (0.2)  | **84.4** (0.2)  | **74.8** (0.1)  | **86.0** (0.2)  | **94.1** (0.1)  | **86.2** |
> | One-shot    | 86.3 (0.6)  | 90.4 (0.1)  | 82.6 (0.1)  | 72.8 (0.3)  | 85.7 (0.1)  | 93.9 (0.1)  | 85.3 |
> | CE          | 85.5 (0.6)  | 90.6 (0.0)  | 83.1 (0.1)  | 73.5 (0.0)  | 85.3 (0.2)  | 94.0 (0.1)  | 85.3 |
> | Teacher     | 87.9 (0.6)  | 91.4 (0.2)  | 86.2 (0.3)  | 78.4 (0.3)  | 87.9 (0.2)  | 94.4 (0.1)  | 87.7 |
>
> **Reasoning via long-form generation**
>
> Here, we provide preliminary experiments on training different GPT models from scratch on TinyGSM [Liu et al. 2023]. We measure the accuracy, where “correctness” requires the model to generate a python program that is both executable and returns the correct numeric solution  for a grade-school math problem.
>
> - The trained models are evaluated by pass rate, defined as the number of problems for which at least one of the python programs returned by the model is correct. In particular, we take pass@1 to be the accuracy for a single greedy generation (i.e. temperature 0) per problem, and pass@10 takes the best solution over 10 candidates generated at temperature 0.8.
>
> Our results show that progressive distillation has better pass rates than cross entropy and one-shot distillation. This suggests that progressive distillation can improve performance of models on more difficult reasoning tasks. However, we would like to note that these experiments deserve a thorough study on its own and are not covered by our current theory. We keep a more extensive exploration to future work.
>
> | Student Configuration       | Metric                | CE    | Progressive | One-shot |
> |---------------------|-----------------------|-------|-------------|----------|
> | **Width 128, layers 4** | Eval loss            | 0.48  | 0.47        | 0.47     |
> |                     | Pass at 1, greedy     | 11.07 | **12.58**   | 11.37    |
> |                     | Pass at 10, 0.8 temp. | 24.79 | **28.89**   | 27.14    |
> | **Width 256, layers 4** | Eval loss            | 0.38  | 0.38        | 0.38     |
> |                     | Pass at 1, greedy     | 25.32 | **29.64**   | 29.11    |
> |                     | Pass at 10, 0.8 temp. | 46.70 | **49.66**   | 48.90    |
>
>
> Here, the teacher was a 4 layer 768 width model, that achieved $38\%$ pass at 1 accuracy.
>
> References:
> 1. Liu et al. 2023. TinyGSM: achieving >80% on GSM8k with small language models.
> 2. Gao et al 2021. Making Pre-trained Language Models Better Few-shot Learners.

---

> > ### Author Response · Authors · 2024-11-21
> >
> > **Q2: Can the authors also experiment with more recent LLMs?**
> >
> > **Response**: Given the goal of studying the underlying mechanism of progressive distillation (and more broadly, how a large model can accelerate training of a small model), our experiments focus on MLP, Bert, and GPT-2 architectures, as these models are more suitable for controlled experiments and interpretability analyses given our compute budget. However, we’d like to point out that there has been evidence that progressive distillation can be useful at more recent, larger-scaled models. For example, as mentioned in line 40-42, Gemini-1.5 Flash has been progressively (online) distilled from Gemini-1.5 Pro (Reid et al., 2024).
> >
> > References: Reid et al. 2024. Gemini 1.5: Unlocking multimodal understanding across millions of tokens of context.
> >
> > **Q3: How do we pick the most appropriate teacher checkpoints in other more complicated tasks?**
> >
> > **Response**: The heuristics and guidelines for progressive distillation can vary significantly depending on the model's loss behavior during training. Our work specifically focuses on tasks where the loss exhibits dormant phases followed by fast phase transitions. We demonstrate that leveraging teacher checkpoints from fast phase transitions can substantially accelerate student training by enabling the model to bypass the slow, dormant phase and providing guidance on the intermediate features that are most beneficial for further learning.
> >
> > Extending our analysis to other diverse tasks will require building toy models that can simulate the input label relationship in these tasks. Additionally, our implicit curriculum perspective provides one possible framework through which we can measure the utility of intermediate teacher checkpoints. Expanding our framework to other perspectives and diverse domains through which larger models can facilitate more efficient and effective learning for smaller models stays an important research direction.

---

> > ### Comment · Reviewer_ktaG · 2024-11-22
> > **Reply to the authors**
> >
> > Thanks for your effort showing additional experiments as requested, which boosted my confidence of the quality of this paper.
> >
> > I increased my soundness and confidence rating of the paper accordingly.

---

> > > ### Author Response · Authors · 2024-11-22
> > >
> > > Thank you for for taking the time to review our additional experiments! Please let us know if you have other comments that could help improve our paper, and we'd love to incorporate your suggestions.

---

> ### Author Response · Authors · 2024-11-21
>
> **Q4: ..more intuitive explanation for the statements in the proof of Theorem 3.2**
>
> **Response**: Thank you for pointing this out.
>
> **Lines 291–292**: This step asserts that the intermediate teacher’s output after first stage of training can be expressed as a polynomial consisting of degree-1 monomials involving the support variables and higher odd-degree polynomials involving the input variables. The coefficients corresponding to degree-1 monomials involving the non-support variables, as well as all even-degree polynomials, are close to zero. Additionally, the higher odd-degree polynomials do not contribute to the gradients for the student model in the subsequent steps.
>
> **Lines 293–294**: Due to the structured output of the intermediate teacher checkpoint, the gradients for the student’s first-layer weights exhibit higher magnitudes for the support coordinates.
>
> **Lines 295–296**: As a result of the larger gradients directed toward the support coordinates, these coordinates experience faster growth in their corresponding weight parameters. To ensure that the noise in the gradients, arising from random sampling, does not outweigh the difference in gradient magnitudes between the support and non-support coordinates, a minimum number of samples is required.
>
> **Q5: For experiments in Figure 2, how do the authors explain the results that the best correlation (Middle) with in-support variable does not lead to best accuracy (Left)?**
>
> **Response**: It is important to note that the true label is determined by the product of the support variables. Consequently, the degree-1 monomials involving the in-support variables are the primary contributors to the error in the teacher’s performance. Therefore, a key takeaway from our work is that the noise present in the intermediate teacher’s predictions, when it has not yet achieved 100% accuracy, is not merely random when averaged across all samples. Instead, this structured noise can be effectively leveraged by progressive distillation to accelerate the training of a smaller student model.
>
> **Q6: L406, what's the formula for total variation?**
>
> **Response:** Total variation distance between two probability vectors p and q is given by $\sum_i |p_i- q_i|.$

---

### Official Review · Reviewer_oRqc · 2024-11-11

**Soundness:** 3
**Presentation:** 3
**Contribution:** 3
**Rating:** 8
**Confidence:** 3

**Summary:**

Knowledge distillation is a widely used technique for training smaller "student" models by leveraging the knowledge captured by larger, pre-trained "teacher" models. This paper focuses on progressive distillation, where a student model is trained on a series of intermediate checkpoints from the teacher's training process, as opposed to one-shot distillation, which relies solely on the final teacher checkpoint.

The authors provide both empirical and theoretical evidence that progressive distillation accelerates student learning and leads to better generalization compared to one-shot distillation. They attribute this improvement to an "implicit curriculum" embedded within the intermediate teacher checkpoints. This curriculum emphasizes easier-to-learn aspects of the task, facilitating the student's learning process.
Two main distillation strategies are compared: one-shot distillation and progressive distillation.  One-shot distillation involves training the student with a fixed, converged teacher model.  In contrast, progressive distillation utilizes multiple checkpoints from the teacher's training trajectory.  The authors also explore a variant of progressive distillation where only a single, carefully selected intermediate checkpoint is used.

The authors delve deeper into the mechanisms behind progressive distillation by utilizing the sparse parity task, a well-established benchmark for studying feature learning in neural networks.  They employ teacher and student models with identical architectures but varying sizes.  For MLPs, the model width is adjusted, while for Transformers, the number of attention heads is modified.  Increasing either parameter effectively increases the number of "parallel search queries" the model can perform during training.

A key finding is that the intermediate teacher checkpoints implicitly provide a "degree curriculum."  This means that the checkpoints guide the student model to learn features in a progressive manner, starting with simpler, lower-degree features. Remarkably, progressive distillation using just a single, well-chosen intermediate checkpoint can surpass the performance of one-shot distillation.  Furthermore, the authors demonstrate that progressive distillation reduces the sample complexity of learning the sparse parity task.


The authors conclude by highlighting the effectiveness of progressive distillation in improving feature learning through the implicit curriculum present in intermediate teacher checkpoints.  They discuss the importance of teacher temperature as a hyperparameter in knowledge distillation and acknowledge limitations in their exploration of this aspect.  Further investigation into the precise role of temperature, particularly its impact on optimization, is proposed as a promising direction for future research.

Another avenue for future work is extending progressive distillation to the setting where student models are trained on synthetic data generated by teacher models.  This approach has shown significant potential in recent studies. The authors identify key differences between their current work and generation-based methods, primarily in the type and quantity of supervision provided.  Bridging this gap and developing a unified framework for progressive distillation in various training scenarios is an important challenge for the field.

**Strengths:**

Originality - The paper presents a novel perspective on progressive distillation by identifying and formalizing the concept of an "implicit curriculum." While prior work has explored progressive distillation empirically, this study delves deeper into the underlying mechanisms and provides theoretical grounding for its efficacy. The connection between intermediate teacher checkpoints and an implicit curriculum is a fresh insight that contributes to a better understanding of knowledge distillation.

Quality - The research is technically sound, employing rigorous methodology and comprehensive experiments. The authors combine theoretical analysis with empirical validation, drawing on diverse tasks like sparse parity, PCFGs, and natural language modeling. The use of multiple progress measures to quantify the implicit curriculum further strengthens the quality of their analysis. The study is well-designed, and the results are convincing.

Clarity -The paper is clearly written and well-organized. The authors present their ideas in a logical progression, starting with a clear motivation and gradually building up their analysis. The concepts are well-explained, and the figures effectively illustrate key findings. The paper is accessible to readers with a background in knowledge distillation and deep learning.


Significance - The findings have significant implications for the field of knowledge distillation. By elucidating the role of an implicit curriculum in progressive distillation, the study provides valuable insights for designing more effective distillation strategies. The theoretical results on sample complexity offer a deeper understanding of the optimization benefits of progressive distillation. The practical implications of this work are substantial, particularly for training efficient and capable models in resource-constrained settings.

**Weaknesses:**

Some possible improvements I can see are the following :-

a) more investigation of impact of temeprature on knowledge distillation. this seems to be a bit missing in the main sections of the paper
b) analysis of how implicit curriculum learning varies across model layers, across datasets, trainign objecives etc

c) exploring more tasks and architectures

d ) explore interaction of optimization algorithms, batch size etc with curriculum learning

**Questions:**

No questions beyond suggestions above:

---

> ### Author Response · Authors · 2024-11-21
>
> We thank the reviewer for their helpful comments about the quality and clarity of our paper.
> Please find our responses below.
>
> **Q1: Impact of temperature**
>
> Thank you for the comment! The temperature is indeed an important hyperparameter for knowledge distillation. We consider the default of $\tau=1$ and the extreme of $\tau \approx 0$. Our one-shot distillation results are reported by taking the better of the two, while the progressive distillation results are reported on $\tau \approx 0$. In other words, we made sure to provide more advantage to one-shot distillation when comparing one-shot and progressive distillation. For progressive distillation alone, Figure 14 in Appendix C.2 provides additional results on different choices of $\tau$.
>
> As mentioned in the paper, we consider $\tau \approx 0$ to remove potential regularization effects induced by soft logits, which are believed to help improve generalization by prior work [Yuan et al. 2020, Menon et al. 2021]. To the best of our knowledge, precise understanding on temperature impact’s on optimization, which is orthogonal to the main claim of our current work and is an interesting future direction.
>
> **Q2: Implicit curriculum across setups**
>
> We answer the question from two aspects:
> - *Across model layers*: We’d appreciate it if the reviewer could clarify what this means. Here are some explanations based on our interpretations:
>   - If the question is about how the implicit curriculum arises in intermediate layers of the teacher, this question could be interpreted as regarding interpretability, which we provide preliminary analyses in Figure 11 and 12 in Appendix C.2 on Transformers.
>   - If the question is about how the implicit curriculum affects intermediate layers of the student, please note that 1) the teacher and the student have different architectures, and 2) the teacher only provides supervision through logits (i.e. final layer outputs), so the intermediate layers are not guaranteed to learn similar representations even if the teacher and the student had the same architecture.
>
> - *Across datasets and training objectives*: Our results consider both discriminative and generative objectives, on both synthetic (i.e. sparse parity and PCFG) and real-world (i.e. Wikipedia + Books) datasets. We identify the low-degree curriculum for sparse parity, and the n-gram curriculum for both PCFG and natural languages.
>
> **Q3: Exploring more tasks and architectures**
>
> We would appreciate it if the reviewer could clarify the specific tasks and architectures of interest. Please also refer to our response to Q2 of reviewer **sKbn.**
>
> **Q4: Interaction of optimization algorithms and batch size**
>
> We use SGD for MLP training and Adam for Transformer training, following common recommendations in practice. Regarding batch sizes, our theory currently requires a sufficiently large batch size, but our experiments do not require so. Empirically, we do not find qualitative difference with different batch sizes. Therefore, we use batch size = 1 (i.e. online SGD) for MLPs, and use a larger batch size for Transformers to speed up training; in particular, we use batch size = 32 for Transformer on parity and batch size = 512 for Transformer on PCFG and natural languages.
>
> We hope our response has helped clarify the questions and concerns. Please let us know if there are further comments, we would love to learn from your insights. Thank you!

---

> > ### Comment · Reviewer_oRqc · 2024-11-26
> > **thank you for clarifications**
> >
> > thanks for the clarifications - i have updated my rating.

---

### Meta-Review · Area_Chair_fyBQ · 2024-12-17

**Metareview:**

The paper studies progressive distillation, where the student learns from a series of intermediate teacher checkpoints, as opposed to learning from a single, fully trained teacher. The authors propose that the improvement from progressive distillation stems from an "implicit curriculum" embedded within these intermediate teacher checkpoints, accelerating the optimization process of the student. This curriculum starts with easier-to-learn aspects of the task and gradually increases complexity as the student progresses through the checkpoints.Interestingly, a better teacher model does not always translate to a better student model. Progressive distillation, with its use of intermediate checkpoints, offers a way to mitigate this challenge.

Strengths: The paper moves beyond the traditional view of knowledge transfer and identifies the implicit curriculum for progressive distillation. The paper is generally well-written and easy to follow. The authors demonstrate the effectiveness of progressive distillation across different tasks and architectures, including sparse parity with MLPs and Transformers, PCFGs, and real-world pre-training datasets. This suggests the generalizability of their findings.

Weaknesses include lack of experiments on models larger than GPT-2 scale, particularly scaling the size of the student and teacher, raising questions about how these findings generalize to modern LLMs of much larger scale. Furthermore, the scope of contribution is limited due to analyzing an existing technique that heavily relies on heuristics to select intermediate teacher checkpoints.

**Additional Comments On Reviewer Discussion:**

Reviewers pointed out several weaknesses and suggestions, and the authors incorporated several of them  including detailed analysis of temperature, evaluations and fine-tuning distilled checkpoints on downstream tasks,  and running prolonged training. As such, several reviewers updated their rating to accept while others updated their confidence in the paper.

---

### Decision · Program_Chairs · 2025-01-22

Accept (Oral)